



# Intercomparison of MAX-DOAS vertical profile retrieval algorithms: studies on field data from the CINDI-2 campaign

Jan-Lukas Tirpitz[1], Udo Frieß[1], François Hendrick[2], Carlos Alberti[3,a], Marc Allaart[4], Arnoud Apituley[4], Alkis Bais[5], Steffen Beirle[6], Stijn Berkhout[7], Kristof Bognar[8], Tim Bösch[9], Ilya Bruchkouski[10], Alexander Cede[11,12], Ka Lok Chan[3,b], Mirjam den Hoed[4], Sebastian Donner[6], Theano Drosoglou[5], Caroline Fayt[2], Martina M. Friedrich[2], Arnoud Frumau[13], Lou Gast[7], Clio Gielen[2,c], Laura Gomez-Martín[14], Nan Hao[15], Arjan Hensen[13], Bas Henzing[13], Christian Hermans[2], Junli Jin[16], Karin Kreher[18], Jonas Kuhn[1,6], Johannes Lampel[1,19], Ang Li[20], Cheng Liu[21], Haoran Liu[21], Jianzhong Ma[17], Alexis Merlaud[2], Enno Peters[9,d], Gaia Pinardi[2], Ankie Piters[4], Ulrich Platt[1,6], Olga Puentedura[14], Andreas Richter[9], Stefan Schmitt[1], Elena Spinei[12,e], Deborah Stein Zweers[4], Kimberly Strong[8], Daan Swart[7], Frederik Tack[2], Martin Tiefengraber[11,22], René van der Hoff[7], Michel van Roozendael[2], Tim Vlemmix[4], Jan Vonk[7], Thomas Wagner[6], Yang Wang[6], Zhuoru Wang[15], Mark Wenig[3], Matthias Wiegner[3], Folkard Wittrock[9], Pinhua Xie[20], Chengzhi Xing[21], Jin Xu[20], Margarita Yela[14], Chengxin Zhang[21], and Xiaoyi Zhao[8,f]

[1]Institute of Environmental Physics, University of Heidelberg, Heidelberg, Germany
[2]Royal Belgian Institute for Space Aeronomy, Brussels, Belgium
[3]Meteorological Institute, Ludwig-Maximilians-Universität München, Munich, Germany
[4]Royal Netherlands Meteorological Institute (KNMI), De Bilt, The Netherlands
[5]Laboratory of Atmospheric Physics, Aristotle University of Thessaloniki, Thessaloniki, Greece
[6]Max Planck Institute for Chemistry, Mainz, Germany
[7]National Institute for Public Health and the Environment (RIVM), Bilthoven, The Netherlands
[8]Department of Physics, University of Toronto, Toronto, Canada
[9]Institute for Environmental Physics, University of Bremen, Bremen, Germany
[10]Belarusian State University, Minsk, Belarus
[11]LuftBlick Earth Observation Technologies, Mutters, Austria
[12]NASA-Goddard Space Flight Center, USA
[13]Netherlands Organisation for Applied Scientific Research (TNO), Utrecht, The Netherlands
[14]National Institute of Aerospatial Technology (INTA), Madrid, Spain
[15]Remote Sensing Technology Institute, German Aerospace Center (DLR), Oberpfaffenhofen, Germany
[16]Meteorological Observation Centre, China Meteorological Administration, Beijing, China
[17]Chinese Academy of Meteorology Science, China Meteorological Administration, Beijing, China
[18]BK Scientific GmbH, Mainz, Germany
[19]Airyx GmbH, Justus-von-Liebig-Straße 14, 69214 Eppelheim, Germany
[20]Anhui Institute of Optics and Fine Mechanics, Chinese Academy of Sciences, Hefei, China
[21]School of Earth and Space Sciences, University of Science and Technology of China, 230026, Hefei, China
[22]Department of Atmospheric and Cryospheric Sciences, University of Innsbruck, Innsbruck, Austria
[a]now at Institute of Meteorology and Climate Research (IMK-ASF), Karlsruhe Institute of Technology (KIT), Karlsruhe, Germany
[b]now at Remote Sensing Technology Institute (IMF), German Aerospace Center (DLR), Oberpfaffenhofen, Germany
[c]now at Institute for Astronomy, KU Leuven, Belgium
[d]now at Institute for Protection of Maritime Infrastructures, Bremerhaven, Germany
[e]now at Virginia Polytechnic Institute and State University, Blacksburg, VA, USA



[f]now at Air Quality Research Division, Environment and Climate Change Canada, Canada

**Correspondence:** Jan-Lukas Tirpitz (jan-lukas.tirpitz@iup.uni-heidelberg.de)

**Abstract.** Multi-AXis Differential Optical Absorption Spectroscopy (MAX-DOAS) is a well-established ground-based measurement technique for the detection of aerosols and trace gases particularly in the boundary layer and the lower troposphere: ultraviolet- and visible radiation spectra of skylight are analysed to obtain information on different atmospheric parameters, integrated over the light path from space to the instrument. An appropriate set of spectra recorded under different viewing

geometries ("Multi-Axis") allows retrieval of tropospheric aerosol and trace gas vertical distributions by applying numerical inversion methods.

     The second Cabauw Intercomparison of Nitrogen Dioxide measuring Instruments (CINDI-2) took place in Cabauw (The Netherlands) in September 2016 with the aim of assessing the consistency of MAX-DOAS measurements of tropospheric species ($NO_2$, HCHO, $O_3$, HONO, CHOCHO and $O_4$). This was achieved through the coordinated operation of 36 spectrom-

eters operated by 24 groups from all over the world, together with a wide range of supporting reference observations (in situ analysers, balloon sondes, lidars, Long-Path DOAS, sun photometer and others).

     In the presented study, the retrieved CINDI-2 MAX-DOAS trace gas ($NO_2$, HCHO) and aerosol vertical profiles of 15 participating groups using different inversion algorithms are compared and validated against the colocated supporting observations. The profiles were found to be in good qualitative agreement: most participants obtained the same features in the retrieved

vertical trace gas and aerosol distributions, however sometimes at different altitudes and of different intensity. Under clear sky conditions, the root-mean-square differences of aerosol optical thicknesses, trace gas ($NO_2$, HCHO) vertical columns and surface concentrations among the results of individual participants vary between $0.01 - 0.1$, $(1.5 - 15) \times 10^{14} \, \mathrm{molec \, cm^{-2}}$ and $(0.3 - 8) \times 10^{10} \, \mathrm{molec \, cm^{-3}}$, respectively. For the comparison against supporting observations, these values increase to $0.02 - 0.2$, $(11 - 55) \times 10^{14} \, \mathrm{molec \, cm^{-2}}$ and $(0.8 - 9) \times 10^{10} \, \mathrm{molec \, cm^{-3}}$. It is likely that a large part of this increase is caused

by imperfect spatio-temporal overlap of the different observations.

     In contrast to what is often assumed, the MAX-DOAS vertically integrated extinction profiles and the sun photometer total aerosol optical thickness were found to not necessarily being comparable quantities, unless information on the real aerosol vertical distribution is available to account for the low sensitivity of MAX-DOAS observations at higher altitudes.

# 1   Introduction

The planetary boundary layer (PBL) is the lowest part of the atmosphere, whose behaviour is directly influenced by its contact with the Earth's surface. Its chemical composition and aerosol load is determined by gas and particulate matter exchange with the surface and also driven by homogeneous and heterogeneous chemical reactions. Monitoring of both, trace gases and



aerosols, preferably simultaneous, is crucial for the understanding of the spatio-temporal evolution of the PBL composition and the chemical and physical processes.

Multi-AXis Differential Optical Absorption Spectroscopy (MAX-DOAS) (e.g. Hönninger and Platt, 2002; Hönninger et al., 2004; Wagner et al., 2004; Heckel et al., 2005; Frieß et al., 2006; Platt and Stutz, 2008; Irie et al., 2008; Clémer et al., 2010;

Wagner et al., 2011; Vlemmix et al., 2015b) is a well-established ground-based measurement technique for the detection of aerosols and trace gases particularly in the PBL and the lower free troposphere: ultraviolet (UV)- and visible (Vis) radiation spectra of skylight are analysed to obtain information on different atmospheric parameters, integrated along the light path (in fact a superposition of a multitude of light paths) from the top of the atmosphere (TOA) to the instrument. The amount of atmospheric trace gases along the light path is inferred by identifying and analysing their characteristic narrow spectral

absorption features, applying differential optical absorption spectroscopy (DOAS, Platt and Stutz, 2008). Detectable gases are nitrogen dioxide ($NO_2$), formaldehyde (HCHO), nitrous acid (HONO), water vapour ($H_2O$), sulfur dioxide ($SO_2$), ozone ($O_3$), glyoxal (CHOCHO) and halogen oxides (e.g. BrO, OClO). The oxygen collision complex $O_4$ can be used to infer information on aerosols: since the $O_4$ concentration is proportional to the square of the $O_2$ concentration, its vertical distribution is well known. The $O_4$ absorption signal can therefore be utilized as a proxy for the light path with the latter being strongly dependent

on the atmosphere's aerosol content. An appropriate set of spectra recorded under a narrow field of view (FOV, full aperture angle around $10\,mrad$) and different viewing elevations ("Multi-Axis") provides information on the trace gas and aerosol vertical distributions. Profiles can be retrieved from this information by applying numerical inversion algorithms, typically incorporating radiative transport models. These profile retrieval algorithms are the subject of this comparison study.

Today, there are numerous such algorithms in regular use within the MAX-DOAS community which rely on different

mathematical inversion approaches. This study involves nine of these algorithms (listed in Table 2), of which six use the optimal estimation method (OEM), two use a parametrized approach (PAR) and one algorithm relies on simplified radiative transport assumptions and analytical calculations (ANA). The main objective of this study is to assess their consistency with respect to different conditions and to review strengths and weaknesses of the individual algorithms and techniques. Note that this study is strongly linked to the report by Frieß et al. (2019), who performed similar investigations on nearly the same set of profiling

algorithms with synthetic data, whereas the underlying data here was recorded during the second "Cabauw Intercomparison for Nitrogen Dioxide measuring Instruments" (CINDI-2, Arnoud et al., 2019 in prep.). The CINDI-2 campaign took place from 25 August to 7 October 2016 on the Cabauw Experimental Site for Atmospheric Research (CESAR, $51.9676\,°\,N$, $4.9295\,°\,E$) in the Netherlands, which is operated by the Royal Netherlands Meteorological Institute (KNMI). 36 spectrometers of 24 participating groups from all over the world were synchronously measuring together with a wide range of supporting observations (in situ

analysers, balloon sondes, lidars, Long-Path DOAS, sun photometer and others) for validation. This study compares MAX-DOAS profiles of $NO_2$, HCHO and aerosol extinction ($O_4$) from 15 of the 24 groups. For HONO and $O_3$ profiling results please refer to Wang et al. (2019 in prep.) and Wang et al. (2018), respectively. The results are compared with each other and validated against CINDI-2 supporting observations. In a recent publication by Bösch et al. (2018), CINDI-2 MAX-DOAS profiles retrieved with the BOREAS algorithm were already compared against supporting observations but regarding a few

days only. Finally it shall be mentioned that already in the course of the precedent CINDI-1 campaign in 2009, there were





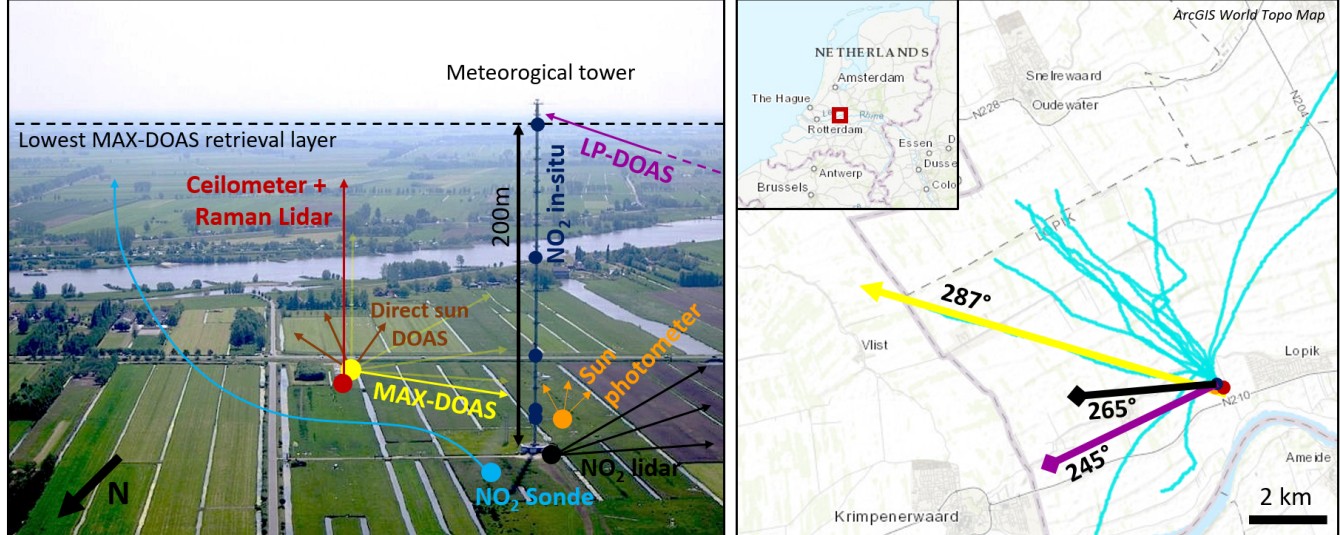

**Figure 1.** Left: Image of the CESAR site with position and approximate viewing directions of the MAX-DOAS instruments and supporting observations of relevance for this study. Right: Map (Esri et al., 2018) with instrument locations, viewing geometries and sonde flight paths indicated.

comparisons of MAX-DOAS aerosol extinction profiles e.g. by Frieß et al. (2016) and Zieger et al. (2011), however also over shorter periods and a smaller group of participants.

The paper is organized as follows: Sect. 2 introduces the campaign setup, the MAX-DOAS dataset with the participating groups and algorithms (Sect. 2.1), the available supporting observations for validation (Sect. 2.2) and the general comparison
5   strategy (Sect. 2.3). The comparison results are shown in Sect. 3. A compact summarizing plot and the conclusions appear in Sect. 4.

## 2   Instrumentation and methodology

Figure 1 shows an overview of the CINDI-2 campaign setup, including the supporting observations relevant for this study. Instrument locations, pointing (remote sensing instruments) and flight paths (radiosondes) are indicated on the map. Details on
10   the instruments and their data products can be found in the following subsections. For further information refer to Kreher et al. (2019) and Arnoud et al. (2019 in prep.).



Atmospheric
Measurement
Techniques

Discussions

## 2.1 MAX-DOAS dataset

### 2.1.1 Underlying dSCD dataset

Deriving vertical gas concentration/aerosol extinction profiles from scattered skylight spectra can be regarded as a two-step process: the 1[st] step is the DOAS spectral analysis, where the magnitude of characteristic absorption patterns of different gas species in the recorded spectra is quantified to derive the so called "differential slant column densities" (dSCDs, definition in the following paragraph). These provide information on integrated gas concentrations along the lines of sight. The 2[nd] step is the actual profile retrieval, where inversion algorithms incorporating atmospheric radiative transfer models (RTM) are applied to retrieve concentration profiles from the dSCDs derived in the 1[st] step.

The very initial data in the MAX-DOAS processing chain are spectra of scattered skylight $I_\lambda(\alpha)$ recorded under different viewing elevation angles $\alpha$ (the telescope's FOV is usually negligible compared to the elevation angle resolution). Along the light path $l$ from the top of the atmosphere (TOA) to the instrument on the ground, each atmospheric gas species $i$ imprints its unique spectral absorption pattern (given by the absorption cross section $\sigma_{i,\lambda}$) onto the TOA spectrum $I_{\lambda,TOA}$ with the optical thickness

$$\tau_\lambda(\alpha) = \log\left(\frac{I_{\lambda,TOA}}{I_\lambda(\alpha)}\right) = \sigma_{i,\lambda} S_i(\alpha) + C \tag{1}$$

$S_i(\alpha)$ is the slant column density (SCD), which is the trace gas concentration integrated along $l$. $C$ represents further terms accounting for other effects than trace gas absorption that will not be further discussed in this context. $S_i(\alpha)$ is inferred by spectrally fitting literature values of $\sigma_{i,\lambda}$ to the observed $\tau_\lambda(\alpha)$. Since normally $I_{\lambda,TOA}$ is not available for the respective instrument, optical thicknesses are instead assessed with respect to the spectrum recorded in zenith viewing direction to obtain

$$\tau_\lambda(\alpha) = \log\left(\frac{I_\lambda(\alpha = 90°)}{I_\lambda(\alpha)}\right) \tag{2}$$

Then the spectral fit yields the so called differential slant column densities (dSCDs)

$$\Delta S(\alpha) = S(\alpha) - S(\alpha = 90°) \tag{3}$$

which are the typical output of the DOAS spectral analysis when applied to MAX-DOAS data. For further details on the DOAS method refer to Platt and Stutz (2008).

During the CINDI-2 campaign, each participant measured spectra with an own instrument and derived dSCDs applying their preferred DOAS spectral analysis software. The pointings (azimuthal and elevation) of all MAX-DOAS instruments were aligned to a common direction (Donner et al., 2019) and all participants had to comply with a strict measurement protocol, assuring synchronous pointing and spectra acquisition under highly comparable conditions (Apituley et al., 2009). A detailed comparison and validation of the dSCD results was conducted by Kreher et al. (2019). In the course of their study, Kreher et al. identified the most reliable instruments to derive a "best" median dSCD dataset. This dataset - in the following referred to as the "median dSCDs" - was distributed among the participants. All participants used the median dSCDs as the input data for their retrieval algorithms and retrieved the profiles that are compared in this study. The "median dSCD" approach was chosen



for the following reasons: i) it enables to compare the profiling algorithms independently from differences in the input dSCDs, which is necessary to assess the individual algorithm performances. ii) it makes this study directly comparable to the report by Frieß et al. (2019). Among others, this allows to assess to what extent MAX-DOAS profiling studies on synthetic data (with lower effort) can be used to substitute studies on real data. iii) two decoupled studies are obtained (Kreher et al. and this

study), each confined to a single step in the MAX-DOAS processing chain (the DOAS spectral analysis to obtain dSCDs and the actual profile inversion). A disadvantage of the median dSCD approach is, that the reliability of a typical MAX-DOAS observation undergoing the whole spectra acquisition and processing chain cannot be assessed. Therefore, a comparison of profiles retrieved with the participant's own dSCDs was also conducted, but is not a substantial part of this study. However, these results and a corresponding short discussion can be found in Supplement S10 and Sect. 3.8, respectively. The median

dSCDs cover the campaign core period from 12 to 28 September 2016, considering only data from the first 10 minutes of each hour between 7:00 and 16:00 UT, where the CINDI-2 MAX-DOAS measurement protocol scheduled an elevation scan in the nominal 287° azimuth viewing direction with respect to the north. Hence, the total number of processed elevation scans was 170. An elevation scan consisted of ten successively recorded spectra at viewing elevation angles $\alpha$ of 1, 2, 3, 4, 5, 6, 8, 15, 30 and 90°, at an acquisition time of 1 minute each. DSCDs were provided for five species, namely $O_4$ UV, $O_4$ Vis, HCHO,

$NO_2$ UV and $NO_2$ Vis, where "UV" and "Vis" indicate different DOAS spectral fitting ranges in the ultraviolet and the visible spectral region, respectively (see Table 1). From the median dSCDs, the participants retrieved profiles for the species listed in Table 1. Not all participants retrieved all species and therefore do not necessarily appear in all plots.

**Table 1.** List of the retrieved species and fitting ranges. For further details on the spectral analysis, please refer to Kreher et al. (2019).

| Species | Retrieved quantity | Retrieved from dSCDs of | spectral fitting window [nm] |
|---|---|---|---|
| Aerosol UV | Extinction [km$^{-1}$] | $O_4$ UV | 338 - 370 |
| Aerosol Vis | Extinction [km$^{-1}$] | $O_4$ Vis | 425 - 490 |
| $NO_2$ UV | Number concentration [molec cm$^{-3}$] | $NO_2$ UV | 336.5 - 359 |
| $NO_2$ Vis | Number concentration [molec cm$^{-3}$] | $NO_2$ Vis | 425 - 490 |
| HCHO | Number concentration [molec cm$^{-3}$] | HCHO | 336.5 - 359 |

### 2.1.2 Participating groups and algorithms

Table 2 lists the compared algorithms including the underlying method (OEM, PAR or ANA) and the participating groups

with corresponding labels and plotting symbols as they are used throughout the comparison. OEM and PAR algorithms rely on the same idea: a layered horizontally homogeneous atmosphere is set up in a radiative transfer model (RTM) with distinct parameters (aerosol extinction, trace gas amounts, temperature, pressure, aerosol microphysical properties, ...) attributed to each layer. This model atmosphere is then used to simulate MAX-DOAS dSCDs under consideration of the viewing geometries. To retrieve a profile from the measured dSCDs, the model parameters are optimized to obtain maximum agreement between

the simulated and measured dSCDs by minimising a pre-defined cost function. Typically only $p = 2$ to $4$ degrees of freedom


for signal (DOFS) can be retrieved from MAX-DOAS observations, such that general profile retrieval problems with more than $p$ layers are underconstrained and *a priori* information has to be assimilated to obtain unambiguous solutions. For OEM algorithms, this is provided in the form of an *a priori* profile (Rodgers, 2000) "filling" the lack of information which is most prominent at higher altitudes (see Sect. 3.1). Parametrized approaches achieve this by only allowing predefined profile shapes

which can be described by a few parameters. For OEM algorithms, the radiative transport simulations are typically performed online in the course of the retrieval whereas the PAR algorithms in this study rely on look-up tables, which are pre-calculated for the parameter ranges of interest. Therefore, PAR algorithms are typically faster than OEM algorithms but require more memory. The ANA approach by NASA was developed as a quick look algorithm and assumes a simplified radiative transport, based on trigonometric considerations. Since the model equations can be solved analytically for the parameters of interest, neither

radiative transport simulation nor the calculation of look-up tables is necessary and an outstanding computational performance is achieved compared to other algorithms (factor of $\approx 10^3$ in processing time, see Frieß et al., 2019). For further descriptions of the methods and the individual algorithms, please refer to Frieß et al. (2019). The $M^3$ algorithm by LMU appears as an additional algorithm in our study. Its description can be found in Supplement S1. For details, refer to the references given in Table 2. Note that two versions of aerosol results from the MAPA algorithm with different $O_4$ scaling factors ($SF$) are

discussed within this paper, referred to as mp-0.8 (retrieved with $SF = 0.8$) and mp-1.0 ($SF = 1.0$), respectively. The scaling factor is applied to the measured $O_4$ dSCDs prior to the retrieval and was initially motivated by previous MAX-DOAS studies which reported a significant yet debated mismatch between measured and simulated dSCDs (Wagner et al., 2019; Ortega et al., 2016, and references therein). Also for MAPA during CINDI-2, a scaling factor of 0.8 was found to improve the dSCD agreement, to enhance the number of valid profiles and to significantly improve the agreement with the sun photometer aerosol

optical thickness (Beirle et al., 2019). However, in the course of this study it was found that for OEM algorithms there are no clear indications that a $SF$ is necessary if smoothing effects, in particular the low sensitivity of MAX-DOAS observations to higher altitudes, are taken into account (see Sect. 2.3.2 and Supplement S3).

### 2.1.3   Retrieval settings

To reduce possible sources of discrepancies, all profiles shown in this study were retrieved according to predefined settings

similar to those of the intercomparison study by Frieß et al. (2019): pressure, temperature, total air density, and $O_3$ vertical profiles were averaged from $O_3$ sonde measurements performed in De Bilt by KNMI during September months of the years 2013-2015. The surface albedo was fixed to 0.06, according to Koelemeijer et al. (2003). A fixed altitude grid was used for the retrieval, consisting of 20 layers between 0 and 4 km altitude, each with a height of $\Delta h = 200\,\mathrm{m}$. The results of the parametrized approaches and OEM algorithms where the exact grid could not be directly implemented, were interpolated/ averaged to this

grid to simplify the comparison. Surface and instruments' altitudes were fixed to 0 m, which is close to the real conditions: the CESAR site and most of the surrounding area lie at 0.7 metres b.s.l., whereas the instruments were installed at 0 to 6 m above sea level. The model wavelengths were fixed according to Table 3. In the case of the HCHO retrieval, the aerosol profiles retrieved at 360 nm were interpolated to 343 nm using the mean Ångström exponent for the 440-675 nm wavelength range derived from sun photometer measurements (see Sect. 2.2.1) on 14 September 2016 in Cabauw. For the aerosol parameters, the





single scattering albedo was fixed to 0.92 and the asymmetry factor to 0.68 for both 360 and 477 nm. These are mean values for 14/09/2016 derived from AERONET measurements at 440 nm in Cabauw. The standard CINDI-2 trace gas absorption cross-sections were applied (see Kreher et al., 2019). A scaling of the measured $O_4$ dSCDs prior to the retrieval was not applied. An exception is the parametrized MAPA algorithm for which two datasets, one without and one with a scaling ($SF = 0.8$)

were included in this study. The OEM *a priori* profiles for both aerosol and trace gas retrievals were exponentially-decreasing profiles with a scale height of 1 km and aerosol optical thicknesses (AOTs) and vertical column densities (VCDs) as given in Table 3. For the AOTs the mean value at 477 nm for the first days of September 2016 derived from AERONET measurements are used. Trace gas VCDs are mean values derived from OMI observations in September 2006-2015. *A priori* variance and correlation length were set to $50\%$ and $200\,\mathrm{m}$, respectively.

### 2.1.4 Requested dataset

All participants were requested to submit the following results of their retrieval: (1) Profiles and profile errors, optionally with errors separated into contributions from propagated measurement noise and smoothing effects. (2) Modelled dSCDs as calculated by the RTM for the retrieved atmospheric state. (3) Averaging Kernels (AVKs) for assessment of information content and vertical resolution (only available for OEM approaches). (4) Optional flags, giving participants the opportunity to

mark profiles as invalid. The flagging must be based on inherent quality indicators, which typically are the root-mean-square difference between measured and modelled dSCDs or the general plausibility of the retrieved profiles. Note that only four institutes submitted flags (INTA, BIRA/ bePRO, KNMI and MPIC/ MAPA). It is assumed that an accurate aerosol retrieval is necessary to infer light path geometries, thus trace gas profiles are generally considered invalid if the underlying aerosol retrieval is invalid. A detailed description of the flagging criteria and flagging statistics can be found in Supplement S4.

## 2.2 Supporting observations

This section introduces the supporting observations, that were used for comparison and validation of the MAX-DOAS retrieved profiles. It shall be pointed out that a general challenge here was to find compromises between i) using only accurate and representative data with good spatio-temporal overlap and ii) keeping as many supporting data as possible to have a large comparison dataset. Considerations and investigations on this issue (e.g. comparisons between the supporting observations,

spatio-temporal variability and overlap, ...) which lead to the decisions finally taken are mentioned in the following subsections and described in more detail in the supplementary material they refer to.

### 2.2.1 Aerosol optical thickness (AOT)

Independent aerosol optical thickness measurements $\tau_{aer}$ were performed with a sun photometer (CE318-T by Cimel) located close to the meteorological tower of the CESAR site (see Fig. 1), which is part of the Aerosol Robotic Network (AERONET,

see Holben et al., 1998). AOTs were derived from direct-sun radiometric measurements in $\approx$ 15 minute intervals at 1020, 870, 675 and 440 nm wavelength. The AERONET level 2.0 data was used, which is cloud screened, recalibrated and quality filtered

(according to Smirnov et al., 2000). For the extrapolation of $\tau_{aer}$ to the DOAS retrieval wavelengths of 360 and 477 nm, a dependency of $\tau_{aer}$ on the wavelength $\lambda$ according to

$$\ln \tau_s(\lambda) = \alpha_0 \, + \, \alpha_1 \cdot \ln \lambda \, + \, \alpha_2 \cdot (\ln \lambda)^2 \tag{4}$$

was assumed, following Kaskaoutis and Kambezidis (2006). The parameters $\alpha_i$ were retrieved by fitting Eq. (4) to the available
data points. Note, that $\alpha_1$ corresponds to the Ångström exponent when only the first two (linear) terms on the right hand side are used. The last quadratic term enables to additionally account for a change of the Ångström exponent with wavelength. For the linear temporal interpolation to the MAX-DOAS profile timestamps, the maximum interpolated data gap was set to 30 min, resulting in a data coverage of about $30\%$. Smirnov et al. (2000) propose a sun photometer total accuracy in $\tau_s$ of 0.02. In this study, an enhanced uncertainty of 0.04 is assumed due to temporal and spectral inter-/extrapolation.

**2.2.2   Aerosol extinction profiles**

Information on the true aerosol extinction (AE) profiles was obtained by combining the sun photometer AOT with data from a ceilometer (Lufft CHM15k Nimbus). The latter continuously provided vertically resolved information on the atmospheric aerosol content by measuring the intensity of elastically backscattered light from a pulsed laser beam (1064 nm) propagating in zenith direction (see e.g. Wiegner and Geiß, 2012). The raw data are attenuated backscatter coefficient profiles over an altitude
range from $180\,\mathrm{m}$ to $15\,\mathrm{km}$, with a temporal and vertical resolution of 12 s and 10 m, respectively. These were converted to extinction coefficient profiles (in the following referred to by "extinction profiles") by scaling with simultaneously measured sun photometer or MAX-DOAS AOTs. This is described in detail in Supplement S5.1. Note that the approach described there presumes a constant extinction coefficient for altitudes $\leq 180\,\mathrm{m}$ and that the aerosol properties like size distribution, single scattering albedo and shape remain constant with altitude. To check plausibility, Supplement S5.1 compares the resulting
profiles at 360 nm to a few available extinction coefficient profiles, measured by a Raman lidar at 355 nm (the CESAR Water Vapor, Aerosol and Cloud lidar "CAELI", operated within the European Aerosol Research lidar Network (EARLINET, Bösenberg et al., 2003; Pappalardo et al., 2014) and described in detail in Apituley et al., 2009). The average RMSD between scaled ceilometer and Raman lidar profiles is $\approx 0.03$. However since there are only few Raman lidar validation profiles available and only for altitudes > 1 km, the ceilometer aerosol extinction profiles should be consulted for qualitative comparison only.

**2.2.3   NO$_2$ profiles**

NO$_2$ profiles were recorded sporadically by two measurement systems: radiosondes (described in Sluis et al., 2010) and an NO$_2$ lidar (Berkhout et al., 2006). Radiosondes were launched at the CESAR measurement site during the campaign. For this study, only data from sonde ascents through the lowest 4 km (which is the MAX-DOAS profiling retrieval altitude range) were used. A sonde profile was considered temporally coincident to a MAX-DOAS profile, when the middle timestamps of
MAX-DOAS elevation scan and sonde flight were less than 30 minutes apart. The horizontal sonde flight paths are indicated in Fig. 1. Typical flight times (lowest 4 km) were of the order of 10 - 15 minutes. Data was recorded at a rate of 1 Hz, typically resulting in a vertical resolution of approximately 10 m at an approximate measurement uncertainty in NO$_2$ concentration of



$5 \times 10^{10}\,\mathrm{molec\,cm^{-3}}$. The horizontal travel distances varied strongly between 4 and 18 km. A detailed overview on the flights is given in Supplement S5.2.

The NO$_2$ lidar is a mobile instrument setup inside a lorry which was located close to the CESAR meteorological tower. It combines lidar observations at different viewing elevation angles to enhance vertical resolution and to obtain sensitivity close to

the ground, despite the limited range of overlap between sending and receiving telescope (see also Sect. 2.2.2). The instrument is sensitive along its line of sight from 300 to 2500 m distance to the instrument. The azimuthal pointing was $265°$ with respect to the north and the operational wavelength is 413.5 nm. Typical specified uncertainties in the retrieved concentrations are around $2.5 \times 10^{10}\,\mathrm{molec\,cm^{-3}}$. Profiles were provided at a temporal resolution of 28 minutes, each profile consisting of a series of (occasionally overlapping) altitude intervals with constant gas concentration. For an exemplary profile and details

on its conversion to the MAX-DOAS retrieval altitude grid, please refer to Supplement S5.3. A lidar profile was considered temporally coincident to a MAX-DOAS profile, when the middle timestamps of MAX-DOAS elevation scan and lidar profile were less than 30 minutes apart. Example profiles of both radiosonde and NO$_2$ lidar are shown in the course of a comparison between the two observations in Supplement S5.5.

### 2.2.4   Trace gas vertical column densities (VCD)

Tropospheric trace gas VCDs were derived from direct-sun DOAS (DS-DOAS) observations, which were performed between minutes 40 and 45 of each hour. NO$_2$ VCDs were retrieved from combined datasets of two Pandora DOAS instruments (instrument numbers 31 & 32) and calculated based on the Spinei et al. (2014) approach. The reference spectrum was created from the spectra with lowest radiometric error over the whole campaign and the residual NO$_2$ signal was determined by applying the so-called Minimum Langley Extrapolation (Herman et al., 2009). The temperature dependence of the NO$_2$ cross sections

was used to separate the tropospheric from the stratospheric column.

HCHO VCDs were retrieved from data of the BIRA DOAS instrument (number 4). A fixed reference spectrum acquired on 18 September 2016 at 9:41 UTC and $55.6°$ SZA was used. DOAS fitting settings were identical to those used for the CINDI-2 HCHO dSCD intercomparison (Kreher et al., 2019). The residual amount of HCHO in the reference spectrum of $(8.8 \pm 1.6) \times 10^{15}\,\mathrm{molec\,cm^{-2}}$ was estimated using a MAX-DOAS profile retrieved on the same day and a geometrical AMF

corresponding to $55.6°$ SZA. Because of that, the HCHO VCDs cannot be considered as a fully independent dataset. VCDs were calculated from total HCHO SCDs using a geometrical AMF including a simple correction for the earth sphericity. Only spectra with DOAS fit residuals $< 5 \times 10^{-4}$ were considered as valid direct-sun data. As for AOTs, these observations can only be performed when the sun is clearly visible, hence the coverage for cloudy scenarios is scarce.

### 2.2.5   Trace gas surface concentrations

Note that in the following, "surface concentration" will not refer to measurements in the very proximity to the ground but to the average concentration in the lowest 200 m of the atmosphere, as retrieved for the MAX-DOAS first profile layer. Trace gas surface concentrations of HCHO and NO$_2$ were provided by a long path DOAS system operated by IUP-Heidelberg (LP-DOAS, see Pöhler et al., 2010; Merten et al., 2011; Nasse et al., 2019). The LP-DOAS system consists of a light-sending





and receiving telescope unit located at 3.8 km horizontal distance to a retro reflecting mirror mounted at the top (207 m altitude) of the meteorological tower (see Supplement S5.4). Light from a UV-Vis light source is sent by the telescope to the retroreflector and the reflected light is again received by the telescope unit and spectrally analysed applying the DOAS method. The fundamental difference to the MAX-DOAS instruments is the well-defined light path which enables very accurate

determination of trace gas mixing ratios, averaged along the line of sight. Accordingly, with the retroreflector mounted at 207 m altitude, one obtains average mixing ratios over the lowest MAX-DOAS retrieval layer, as indicated in Fig. 1. Considering DOAS fitting errors and uncertainties in the applied literature cross-sections (Vandaele et al., 1998; Meller and Moortgat, 2000; Pinardi et al., 2013) yields an average accuracy of the LP-DOAS of $\pm 1.5 \times 10^9 \, \mathrm{molec \, cm^{-3}} \pm 3\%$ ($\pm 5 \times 10^9 \, \mathrm{molec \, cm^{-3}} \pm 9\%$) for $NO_2$ (HCHO), respectively. Given the high accuracy, the total vertical coverage of the surface layer and a near-continuous

dataset over the campaign period, the LP-DOAS provides the most reliable dataset for the validation of CINDI-2 MAX-DOAS trace gas profiling results.

Further observations for qualitative validation are the surface values of the $NO_2$ lidar and the radiosondes and also in-situ monitors in the CESAR meteorological tower. Teledyne in situ $NO_2$ monitors (Teledyne API, model M200E) were located in the tower basement and were subsequently connected to different inlets located at 20, 60, 120 and 200 m altitude (switching

intervals approx. 5 minutes). Further, a CAPS (type AS32M, based on attenuated phase shift spectroscopy, Kebabian et al., 2005) and a CE-DOAS (cavity enhanced DOAS, Platt et al., 2009 and Horbanski et al., 2019) were continuously measuring at 27 m altitude. All the in situ measurements at the tower were combined to obtain another set of surface concentration measurements, more representative for concentrations close to the site. The data were combined by linearly interpolating over altitude between the instruments and subsequently averaging the resulting profile over the retrieval surface layer (0 - 200 m

altitude).

### 2.2.6 Meteorology

Meteorological data for the surface layer (pressure, temperature and wind information) routinely measured at the CESAR site were taken from the CESAR database (CESAR, 2018) at a temporal resolution of 10 minutes. Cloud conditions were retrieved from MAX-DOAS data of instruments 4 and 28 according to the cloud classification algorithm developed by MPIC

(Wagner et al., 2014; Wang et al., 2015). Basically only two cloud condition states are distinguished in the statistical evaluation: "clear-sky" (green) and "presence of clouds" (red). Only in the overview- and correlation plots, "presence of clouds" is further subdivided into "optically thin clouds" (orange) and "optically thick clouds" (red). According to this classification 72 (98) of the 170 profiles were measured under clear-sky (cloudy) conditions. Over the whole campaign, there was only one rain event (precipitation > 0.01 mm) coinciding with the measurements on 25 September 2016 between 15:00 and 17:00 h UT. At

forenoon on 16 September, a heavy fog event strongly limited the visibility (see also Supplement S6).



## 2.3 Comparison strategy

### 2.3.1 General approach

Different MAX-DOAS retrieval algorithms were extensively compared in Frieß et al. (2019) using synthetic data. The crucial differences of the presented study are: i) The underlying spectra are not synthetic, but were recorded with real instruments, meaning that real noise and instrument artefacts propagate into the results. ii) Independent information on the real profile can only be inferred from supporting observations with their own uncertainties and an imperfect spatio-temporal overlap with the MAX-DOAS measurements. iii) The real conditions encountered can exceed the model's scope because horizontal inhomogeneities or the fact that many of the fixed forward model input parameters (such as aerosol properties, surface albedo, T/P – profiles, ...) are averaged quantities of former observations which might be inaccurate for specific days and conditions. iv) In some cases, different participants used the same retrieval algorithms; this allows assessment of the impact of different settings in the remaining parameters, which were not prescribed (see Sect. 2.1.3). The approaches chosen here are therefore limited to the examination of i) the consistency among the participants, ii) the consistency of the results with available supporting observations and iii) inherent quality proxies of the retrieval (described in the next paragraph). Table 4 summarizes the quantities which are compared, together with the corresponding supporting observations if available.

In this study, agreement between different observations are statistically assessed by correlation analysis (weighted least-squares regression) and weighted root-mean-square differences (RMSD). Discussions and summary are focussed on RMSDs as in contrast to correlation coefficient, slope and offset from the regression analysis, RMSD is representative for both, statistical and systematic deviations. For two time series $x_{p,t}$ and $x_{ref,t}$ (each consisting of $N_T$ data points, $t$ and $p$ indicating time and participant, respectively) with associated uncertainties $\sigma_{p,t}$ and $\sigma_{ref,t}$ the RMSD is given by

$$\text{RMSD:} \qquad \sigma_{rms,p} = \sqrt{\frac{1}{N_T} \cdot \frac{1}{\sum_t w_t} \cdot \sum_t w_t \left(x_{p,t} - x_{ref,t}\right)^2} \qquad (5)$$

For both, RMSD calculation and least square regression, contributing data points are weighted by the reciprocal of the quadratic sum of their uncertainties:

$$w = \frac{1}{\sigma_{p,t}^2 + \sigma_{ref,t}^2} \qquad (6)$$

Sometimes the term "average RMSD" is used, which refers to the average over the RMSD values of the individual participants, hence

$$\text{Average RMSD:} \qquad \sigma_{arms,p} = 1/N_P \sum_p \sigma_{rms,p} \qquad (7)$$

with $N_P$ being the number of included participants. When referring to "relative RMSDs", the underlying RMSD value was divided by the average of the investigated quantity, hence:

$$\text{Relative RMSD:} \qquad \sigma_{rrms,p} = \frac{N_T \sigma_{rms,p}}{\sum_t x_{ref,t}} \qquad (8)$$





The consistency among the participants is assessed by comparing the results of individual participants with the median result over the valid profiles of all participants. The median is used instead of the mean value, since it is less sensitive to (sometimes unphysical) outliers. This comparison shows how far the choice of the retrieval algorithm/ technique affects the results but it does not reveal general systematic MAX-DOAS retrieval errors. Outliers observed for distinct participants and algorithms are

therefore not necessarily an indicator for poor performance.

The consistency with supporting observations is a better indicator for the real retrieval performance. However, uncertainties of supporting instruments (see Supplement S5.5), smoothing effects (see Sect. 2.3.2) and imperfect spatial and temporal overlap of the different observations (see Sect. 2.3.3) complicate the interpretation.

Inherent quality indicators for retrieval algorithms are the consistency of modelled and measured dSCDs and the consistency

of $NO_2$ results retrieved in different wavelength ranges. During the inversion, the goal is to minimize the deviation between the RTM simulated dSCDs and the actually measured ones. If strong deviations remain after the final iteration in the minimisation process, this indicates failure of the retrieval. The consistency of retrieval results of $NO_2$ in the UV and the Vis spectral ranges is another indicator for an algorithm's reliability since they should ideally yield the same results.

In a few cases (e.g. Section 3.2) the scatter among several participants $p$ (of number $N_P$) and potentially several retrieval

layers $h$ (of number $N_H$) is of interest. For this purpose, we define the "average standard deviation" (ASDev) which is the standard deviation observed among the participants for individual profiles averaged over retrieval layers and time, hence:

$$\text{ASDev:} \qquad \sigma_{asdev} = \frac{1}{N_T} \sum_t \frac{1}{N_H} \sum_h \sqrt{\frac{1}{N_P - 1} \sum_p \left( x_{p,h,t} - \bar{x}_{h,t} \right)^2} \tag{9}$$

with $\bar{x}_{t,h}$ being the average (over participants) MAX-DOAS retrieved concentration for a given time $t$ and layer $h$. If not stated otherwise, ASDev values of profiles are calculated considering the lowest five retrieval layers (up to $1\,\text{km}$ altitude).

In the statistical evaluations, clear-sky and cloudy conditions as well as unfiltered and filtered data are distinguished. The distinction between cloud conditions is of major importance, as particularly in the case of aerosol retrievals under broken clouds, the quality of the results is typically strongly degraded. A consequence of regarding these data subsets is that the number of contributing data points not only depends on the number of submitted profiles and the number of coincident data points from supporting observations but further on the filter settings. Any regression or RMSD with less than five contributing

data points are considered to be statistically unrepresentative and are omitted. If not stated otherwise, numbers given in the text were calculated considering valid data only.

### 2.3.2   Smoothing effects

As shown in Sect. 3.1 below, in particular in the UV range, the sensitivity of ground-based MAX-DOAS observations decreases rapidly with altitude, meaning that species above $\approx 1\,\text{km}$ typically cannot be reliably detected. At higher altitudes, OEM

retrieval results are drawn towards the *a priori* profile, while the results of parametrized and analytical approaches are driven by the chosen parametrization and their implementation. Further, the vertical resolution is limited (from 100 to several hundred meters, increasing with altitude), which affects the profile shape and - of most importance in this study - the retrieved surface concentration. Both effects cause deviations from the true profile that are in the following referred to as "smoothing effects". For



a meaningful quantitative comparison, they should be considered. This is possible for OEM retrievals, where the information on the vertical resolution and sensitivity is given by the averaging kernel matrix (AVK, see Sect. 3.1 for details). For a meaningful quantitative comparison of an OEM retrieved profile and a validation profile $x$ (assumed here to perfectly represent the true state of the atmosphere), the validation profile resolution and information content has to be degraded by "smoothing" it with

the corresponding MAX-DOAS AVK matrix $\mathbf{A}$ according to the following equation (Rodgers and Connor, 2003; Rodgers, 2000):

$$\widetilde{x} = \mathbf{A}x + (1 - \mathbf{A})x_a \tag{10}$$

Here, $x_a$ is the *a priori* profile and $\widetilde{x}$ represents the profile that a MAX-DOAS OEM retrieval (with the resolution and sensitivity described by $\mathbf{A}$) would yield in the respective scenario. For layers with high (low) gain in information, $\widetilde{x}$ is drawn

towards $x$ ($x_a$), while vertical resolution is degraded if $\mathbf{A}$ has significant off-diagonal entries (compare to Sect. 3.1). In this study, this has implications not only for the comparison of profiles, but also the comparison of the total columns (AOTs and VCDs, which are derived simply by vertical integration of the corresponding profiles) and surface trace gas concentrations. For total columns, the dominant issue is the lack of information at higher altitudes. In contrast, there is reasonable information on the surface concentration, however smoothing can have severe impact here in the case of strong concentration gradients

close to the surface. The impact on the individual observations is discussed in the corresponding sections below. A particularly important consequence of smoothing effects is the "partial AOT correction" (PAC), which is introduced in Sect. 3.4.

### 2.3.3   Spatio-temporal variability

It is obvious already from Fig. 1 and Sect. 2.2 that the MAX-DOAS instruments and the various supporting observations sample different air volumes at different times. In addition, the MAX-DOAS horizontal viewing distance (derived in Supplement S6)

is highly variable, changing between 2 and 30 km during the campaign for the lowest viewing elevation angles. Similar investigations were already performed by Irie et al. (2011) using CINDI-1 data, however using a different definition of the viewing distance. Hence, strong spatio-temporal variations of the observed quantities are expected to induce large discrepancies among the observations, independent of the data quality. It shall further be noted, that under strong spatial variability the horizontal homogeneity assumed by the retrieval forward models is not given. It was not possible to derive a reliable quantitative estimate

of the impact on the comparison, but investigations on the $NO_2$ surface concentration in Supplement S7 and investigations by Peters et al. (2019) indicate, that it significantly contributes to the residual RMSD observed between different observations.

## 3   Comparison results

### 3.1   Information content

In the case of OEM retrievals, the gain in information on the atmospheric state can be quantified according to Rodgers (2000).

Essentially speaking, this is done by comparing the knowledge before (represented by the *a priori* profile and its uncertainties) and after the profile retrieval. The gain in information for each individual vertical profile can be represented by the averaging





kernel matrix (AVK, denoted by $\mathbf{A}$). Each of its elements $A_{ij}$ describes the sensitivity of the concentration in the $i^{\text{th}}$ layer to changes in the real concentration in the $j^{\text{th}}$ layer. Each row $A_i$ can thus be plotted over altitude providing the following information: (1) the value in the layer $i$ itself (the diagonal element $A_{ii}$ with a value between 0 and 1) gives the gain in information while $1 - A_{ii}$ represents the amount of *a priori* knowledge which had to be assimilated to obtain a well defined

concentration value. (2) The values in the other layers (off-diagonal elements of $\mathbf{A}$) indicate the cross sensitivity of layer $i$ to layer $j$. Typically, the cross sensitivity decreases with the distance to the layer $i$. A reasonably defined characteristic length of this decay (note, that $i$ can be converted to the corresponding altitude by multiplication with the retrieval layer thickness $\Delta h$) can serve as a measure for the vertical resolution of the retrieval. Here, the so called "spread" $s(i)$ was chosen as the characteristic length, as defined according to equation 3.23 in Rodgers (2000):

$$s(i) = 12 \cdot \Delta h \cdot \frac{\sum_j (i-j)^2 A_{ij}}{\left(\sum_j A_{ij}\right)^2} \tag{11}$$

The trace of $\mathbf{A}$ equals the degrees of freedom of signal (DOFS), hence the total number of independent pieces of information gained from the measurements compared to the *a priori* knowledge. Figure 2 visualizes the average AVK matrices for all five species studied in this work. Note, that the AVKs do not necessarily represent the real/ total sensitivity and information content of MAX-DOAS observations as they only consider the gain of information with respect to the *a priori* knowledge. Hence, for

stricter *a priori* constraints less gain in information will be indicated by the AVKs.

For all species, the sensitivity is limited to about the lowest 1.5 km of the atmosphere. More information is obtained on the Vis species, as the light path increases with wavelength resulting in higher sensitivity. The obtained DOFS are generally a bit lower as observed in former studies. This is related to the rather small *a priori* covariance (50 %, see Sect. 2.1.3), which implies a good knowledge on the atmospheric state prior to the retrieval and finally leads to less gain in information from the

measurements. Figures S36, S37, S38, S39 and S40 in Supplement S8.1 show the average AVKs of the individual participants and reveals, that there are significant differences (up to 1 DOFS) between the participants even when using the same algorithm (up to 0.5 DOFS in the case of PRIAM). This indicates that the information content is not assessed consistently. BOREAS for instance states a very low gain in information especially for Aerosol Vis. This is related to an additional Tikhonov term used as a smoother which was also applied during AVK assessment. Furthermore, all BOREAS results were retrieved on another

grid and interpolated onto the submission grid, which leads to a decrease in all AVKs and therefore the DOFS. On average, the dependence of the total amount of information on the cloud conditions is small (typically decrease of 0.1 DOFS). Examination of the AVKs of individual profiles (not shown here), indicated that there are two competing effects: (1) the presence of clouds can increase the sensitivity to higher layers due to multiple scattering and thus light path enhancement in the clouds whereas (2) a decrease in the horizontal viewing distance (e.g. due to fog, rain or high aerosol loads) reduces the information content,

since the light paths are shorter and their geometry depends less on the viewing elevation.





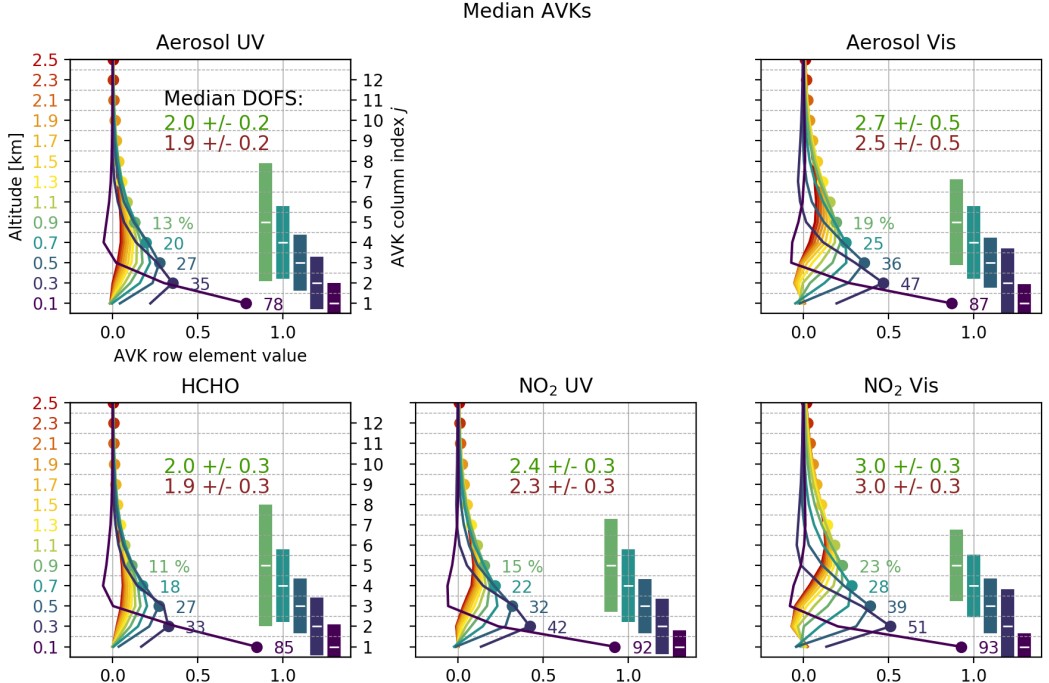

**Figure 2.** Mean AVKs for the retrieved species (median over participants, mean over time). Their meaning is described in detail in the text. Each altitude and corresponding AVK line $A_i$ are associated with a colour, which is defined by the colour of the corresponding altitude-axis label. The dots mark the AVK diagonal elements. The number next to the dots show the exact value in percent, which corresponds to the amount of retrieved information on the respective layer. In each panel, the numbers indicate the DOFS (median among institutes, average over time) for clear-sky (green) and cloudy conditions (red). The vertical bars indicate the vertical resolution (the "spread", defined according to Eq. 11) for the five lowest layers.

## 3.2 Overview plots

Figures 3 to 7 show the retrieved profiles of all participants over the whole semi-blind period. They serve as the basis for a general qualitative comparison. For the trace gases, the altitude ranges (full range is 4 km) were reduced to $0 - 2.5\,\mathrm{km}$ for better visibility, considering the MAX-DOAS sensitivity range and the occurrence altitude of the respective species.

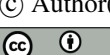



**Table 2.** Groups who retrieved and provided profiling results for this study.

| Algorithm | Method/ Model | Literature | Participants | Acronym | Sym |
|---|---|---|---|---|---|
| bePRO | OEM$^o$/ LIDORT | Clémer et al. (2010) | Aristotle University of Thessaloniki, Thessaloniki, Greece | AUTH | ● |
| | | Hendrick et al. (2014) | Royal Belgian Institute for Space Aeronomy, Brussels, Belgium | BIRA | ▶ |
| | | | National Institute of Aerospatial Technology, Madrid, Spain | INTA | ◆ |
| PRIAM | OEM$^l$/ SCIATRAN | Wang et al. (2013b, 2013a, 2017) | Anhui Institute Of Optics and Fine Mechanics, Anhui, China | AIOF | ● |
| | | | Belarusian State University, Minsk, Belarus | BSU | ▶ |
| | | | China Meteorological Administration, Beijing, China | CMA | ■ |
| | | | Max-Planck Institute for Chemistry, Mainz, Germany | MPIC | ◆ |
| HEIPRO$^x$ | OEM$^l$/ SCIATRAN | Yilmaz (2012) | Institute of Environmental Physics, University of Heidelberg, Germany | IUPHD | ● |
| | | | Department of Physics, University of Toronto, Toronto, Canada | UTOR | ▶ |
| BOREAS | OEM$^l$/ SCIATRAN | Bösch et al. (2018) | Institute of Environmental Physics, University of Bremen, Germany | IUPB | ● |
| M$^3$ | OEM$^l$/ LibRadTran | Chan et al. (2019, 2017) | Ludwig-Maximilians-University, Munich, Germany | LMU | ● |
| MMF | OEM$^l$/ VLIDORT | Friedrich et al. (2019) | Royal Belgian Institute for Space Aeronomy, Brussels, Belgium | BIRA | ● |
| Realtime | ANA$^a$/ - | | NASA-Goddard, Greenbelt, Maryland | NASA | ● |
| MARK | PAR$^p$/ DAK | Vlemmix et al. (2011, 2015a) | Royal Netherlands Meteorological Institute, De Bilt, The Netherlands | KNMI | ● |
| MAPA | PAR/ McArtim | Beirle et al. (2019) | Max-Planck Institute for Chemistry, Mainz, Germany | MPIC$^y$ | ● |
| | | | Max-Planck Institute for Chemistry, Mainz, Germany | MPIC$^y$ | ▶ |

$^o$ OEM: Optimal estimation
$^a$ ANA: Analytical approach without radiative transport model
$^p$ PAR: Parametrized approach
$^x$ IUPHD and UTOR used different versions of HEIPRO (1.2 and 1.5/1.4, respectively)
$^y$ Two versions of MAPA (labelled mp-10 and mp08) with different $O_4$ scaling factors (0.8 and 1.0) are included in the comparison.
$^l$ Aerosol extinction is retrieved in logarithmic space. This removes negative values and allows larger values.



**Table 3.** Prescribed settings for the radiative transport simulation wavelengths and *a priori* total columns (OEM algorithms only).

| Species | RTM wavelength [nm] | *A priori* VCD/ AOT |
|---|---|---|
| Aerosol UV | 360 | 0.18 |
| Aerosol Vis | 477 | 0.18 |
| NO$_2$ UV | 360 | $9 \cdot 10^{15}$ molec cm$^{-2}$ |
| NO$_2$ Vis | 460 | $9 \cdot 10^{15}$ molec cm$^{-2}$ |
| HCHO | 343 | $8 \cdot 10^{15}$ molec cm$^{-2}$ |

**Table 4.** Overview on compared quantities and available supporting data.

| Species | Quantity | Supporting observations | Result section |
|---|---|---|---|
| Aerosol UV | Profiles | Ceilometer[a] (Sec. 2.2.2) | 3.2 & Suppl. S8.2 |
| | Aerosol optical thickness (AOT) | Sun photometer (Sec. 2.2.1) | 3.4 |
| Aerosol Vis | Profiles | Ceilometer[a] | 3.2 & Suppl. S8.2 |
| | Aerosol optical thickness (AOT) | Sun photometer | 3.4 |
| HCHO | Profiles | N.A. | 3.2 & Suppl. S8.2 |
| | Vertical column (VCD) | Direct-sun DOAS (Sec. 2.2.4) | 3.5 |
| | Surface concentration | Long-path DOAS | 3.6 |
| NO$_2$ UV/Vis | Profiles | NO$_2$-Lidar & radiosonde[b] | 3.2 & Suppl. S8.2 |
| | Vertical column (VCD) | Direct-sun DOAS | 3.5 |
| | Surface concentration | Long-path DOAS | 3.6 |
| | UV vs. Vis retrieval | N.A.[c] | 3.7 |
| All species | Modelled vs. measured dSCDs | N.A.[c] | 3.3 |

[a] Elastic backscatter profiles scaled with sun photometer or MAX-DOAS AOT.

[b] Scarce data coverage.

[c] Inherent quality proxy.





**Figure 3.** Aerosol UV extinction profiles. For MAX-DOAS profiles (plots above the wind roses), red triangles at the top of the corresponding profile indicate invalid data. The lowest row shows AOT scaled ceilometer backscatter profiles, calculated as described in Sect. 2.2.2 (unsmoothed). Backscatter profiles, which were scaled from MAX-DOAS AOTs (and which are therefore not fully independent) are marked by red triangles. Maximum extinction values reach 20km$^{-1}$, exceeding the colour scale. Index letters behind the participant labels indicate whether an OEM (o) or parametrized (p) approach was used and whether aerosol was retrieved in the logarithmic space (l). The wind roses in the lower part of the panel show wind direction (azimuth), wind speed (see colour bar on the right) and occurrences (amplitude). The line close to the panel bottom marked with "CC" indicates the cloud conditions, as described in Sect. 2.2.6.





**Figure 4.** Aerosol Vis extinction profiles. Caption of Fig. 3 applies.

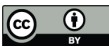

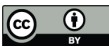

**Figure 5.** HCHO concentration profiles. The plot is similar to Fig. 3. Open red triangles at the top of the MAX-DOAS profiles indicate that the underlying aerosol retrieval failed, whereas the trace gas profile retrieval itself was considered successful. The "Surf"-row shows LP-DOAS surface concentrations.





**Figure 6.** NO$_2$ UV concentration profiles. The lowest row shows a combined dataset of NO$_2$ lidar, radiosonde, LP-DOAS and tower in-situ data. Redundant surface concentration measurements were averaged.



**Figure 7.** NO$_2$ Vis concentration profiles. The lowest row shows a combined dataset of NO$_2$ lidar, radiosonde, LP-DOAS and tower in-situ data. Redundant measurements were averaged here.





Considering valid data only, all algorithms detect similar features in the vertical profiles, but smoothed to different amounts and sometimes detected at different altitudes. For clear sky condition, the observed ASDevs are $3.5 \times 10^{-2}\,\mathrm{km^{-1}}$, $4.0 \times 10^{-2}\,\mathrm{km^{-1}}$, $1.2 \times 10^{10}\,\mathrm{molec\,cm^{-3}}$, $2.4 \times 10^{10}\,\mathrm{molec\,cm^{-3}}$ and $4.4 \times 10^{10}\,\mathrm{molec\,cm^{-3}}$ for Aerosol UV, Aerosol Vis, HCHO, $NO_2$ UV and $NO_2$ Vis, respectively. When regarding participants using the same algorithm, these values are reduced only by

about $50\,\%$, indicating that significant discrepancies are caused by differences in retrieval settings that were not prescribed (e.g. number of iteration in the inversion, accuracy criteria for the RTMs, update interval of the jacobians, ...). Larger discrepancies appear for the species measured in the Vis spectral range than in the UV. For $NO_2$ (aerosol) the ASDev increases in the Vis by $50\,\%$ ($90\,\%$). In the case of OEM algorithms, a reason might be that there is lower information content in the UV, meaning that the retrievals are drawn closer to the collectively used *a priori* profile. Further, the larger viewing distance of the Vis

retrievals (see Supplement S6) might be problematic, since the exact treatment of the viewing geometries (Earth curvature, treatment of instrument field of view, ...) gain influence. Horizontal inhomogeneities are an unlikely reason because the worse performance in the Vis was also apparent in the study by Frieß et al. (2019) with synthetic data, where horizontal gradients were non-existent. The presence of clouds affects ASDevs very differently for different species: for Aerosol UV and Vis it is degraded by a factor of 3 and 4, respectively, which is expected since clouds mostly feature high optical depths > 1 and are

detected to very different extent by the individual participants. For HCHO the ASDev decreases by $38\,\%$ which can be well explained by the systematically lower ($-36\,\%$) HCHO concentrations observed under cloudy conditions. ASDevs for $NO_2$ increase by about $20\,\%$, while the observed concentrations remain similar (increase < $10\,\%$). Considering valid data only, the parametrized approaches are mostly in good agreement with the other algorithms. For MAPA, unrealistic results are reliably identified and flagged as invalid, whereas in the case of MARK some valid profiles do not look plausible e.g. for Aerosol

Vis on 22 September 2016. For both algorithms a large fraction (30 to $70\,\%$) of the profiles are discarded as invalid or look unrealistic if the retrieval conditions are not ideal (see also flagging statistics in Sect. 4). Gaps in the MARK data appear where no optimum solution could be found at all.

For aerosol, OEM algorithms often see elevated layers in the Vis even in clear-sky scenarios that cannot be observed in the UV or the ceilometer profiles. On cloudy days, MMF is capable of detecting clouds as very defined features with a good

qualitative agreement with the ceilometer data. In the Vis, even high clouds are detected, e.g. on 17 September and 22 September 2016, which indeed coincide with high-altitude clouds above the retrieval altitude range of $4\,\mathrm{km}$. An example for large discrepancies between participants using the same algorithm is AUTH aerosol in the UV, where in contrast to other bePRO users oscillations seem to appear. We suspect this to originate from technical problems which could not yet been identified. The discrepancies between IUPHD and UTOR (both using HEIPRO) were found to mainly be caused by differences in the

number of applied iteration steps in the Levenberg-Marquardt optimization scheme during aerosol retrieval. IUPHD (UTOR) applied 20 (5) iterations. The consequences are evident throughout the comparison. Compared to the parametrized approaches, OEMs and the Realtime algorithm yield realistic profiles also under less favourable measurement conditions (e.g. clouds); in particular the OEM results are in qualitative agreement with the ceilometer profiles for many cases.

Regarding HCHO, the agreement of the profiles is exceptionally good considering the particularly low information content of

the measurements (due to higher uncertainties in the dSCD data). Probably because observed spatial and temporal concentration





gradients are much smaller than for $NO_2$, which might partly be related to enhanced smoothing by the retrieval, but is also well possible to be real, since HCHO sources (mainly the photolysis of volatile organic compounds) are less localized. High HCHO concentrations coincide with clear-sky conditions and with wind from the continent, which is what would be expected from the current knowledge on the origin and chemistry of atmospheric HCHO. As in the case of aerosol, there are significant

discrepancies among the bePRO participants, this time with INTA standing out of the group with slight overestimation.

For $NO_2$ very shallow layers and large vertical and horizontal gradients might complicate the retrievals. Nevertheless, good ASDev is achieved in the UV. Week-days and weekends (17, 18, 24 and 25 September) can clearly be distinguished. The lowest concentrations are observed on 18 September, where a Sunday coincides with northerly winds from the sea.

The agreement with the supporting observations will be discussed in detail in the following sections.

## 3.3 Modelled and measured dSCDs

An intrinsic indicator for a successful profile retrieval is a good agreement between the measured and the modelled dSCDs, the latter being the dSCDs obtained from the RTM model for the finally retrieved aerosol and trace gas profiles. Poor agreement might indicate that only a local minimum of the cost function was found (OEM approaches), that inappropriate retrieval settings were chosen (e.g. too small number of iterations in the minimisation) or that the RTM is inaccurate for other reasons,

for instance because it cannot describe horizontal inhomogeneities. Figures 8 to 12 show the correlation of measured and modelled dSCDs for all profiles and elevations of each participant. The NASA/ Realtime algorithm is not included since it does not use an RTM and therefore does not provide simulated dSCDs.

For clear-sky conditions, good agreement is achieved by most participants. Only IUPB, AUTH, BSU, KNMI exceed relative RMSDs of $10\%$ and only for $O_4$ and $NO_2$ Vis dSCDs. MMF achieves the best overall performance, being the only algorithm

with relative RMSDs $< 5\%$ for all species. Regarding HEIPRO, UTOR yields larger RMSD values than IUPHD, which is very likely related to the aforementioned smaller number of iterations applied by UTOR. For the trace gases, small relative RMSD values between $8\%$ and $8\%$ are achieved for all cloud conditions.

Regarding aerosol, PRIAM and BOREAS feature slightly too low slopes in the UV (approx. 0.9) and more pronounced in the Vis (0.8 to 0.85) interestingly almost exclusively caused by data recorded on the 23 and 27 September where the atmospheric

aerosol load is particularly low. RMSDs increase for cloudy scenarios by $10\%$ (HCHO), $30\%$ ($NO_2$ UV) and $50\%$ ($NO_2$ Vis, $O_4$), most likely because the horizontal inhomogeneity cannot be adequately reproduced by the 1D models. This is supported by the comparison results from synthetic data by Frieß et al. (2019), where horizontal homogeneity is inherently assured and the scatter remains similar for all cloud scenarios. KNMI has problems to reproduce $O_4$ dSCDs (relative RMSD $> 30\%$), while for trace gases the performance is comparable to the other algorithms. Regarding Vis species, $M^3$ shows outliers under cloudy

conditions (while performing excellently in the UV) and bePRO seems to have convergence problems, which was also evident in the synthetic data (Frieß et al., 2019). This problem is overcome by flagging of approx. $10\%$ of the data, reducing the RMSD by $> 50\%$. PRIAM (except MPIC) shows outliers, in particular for $NO_2$ Vis. The $O_4$ scaling factor of 0.8 for MAPA improves $O_4$ dSCD agreement in the UV by about $35\%$ (for clear sky and valid data), but not in the Vis spectral range (see also Supplement S3).





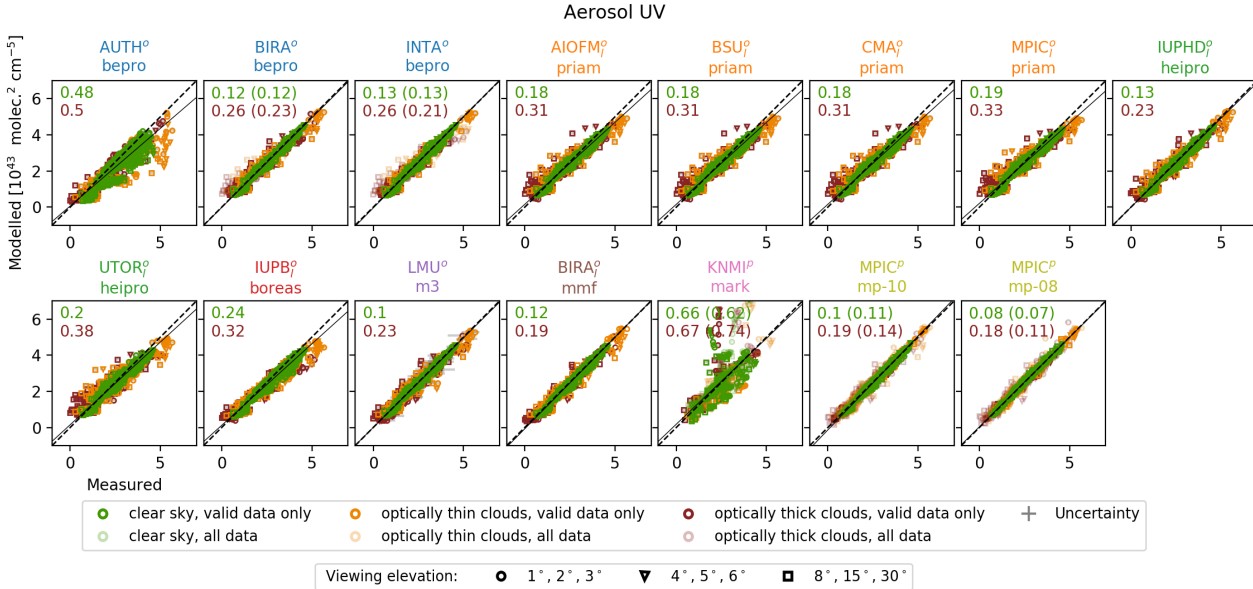

**Figure 8.** O$_4$ UV dSCD correlation. Marker colours and marker shapes indicate the cloud conditions and viewing elevation angles, respectively, as indicated in the legend. Numbers represent the measurement-error-weighted RMSD between measured and modelled dSCDs in units of $10^{43}$ molec$^2$ cm$^{-5}$ for clear sky (green) and cloudy (red) conditions. Values in brackets were calculated only considering valid data.

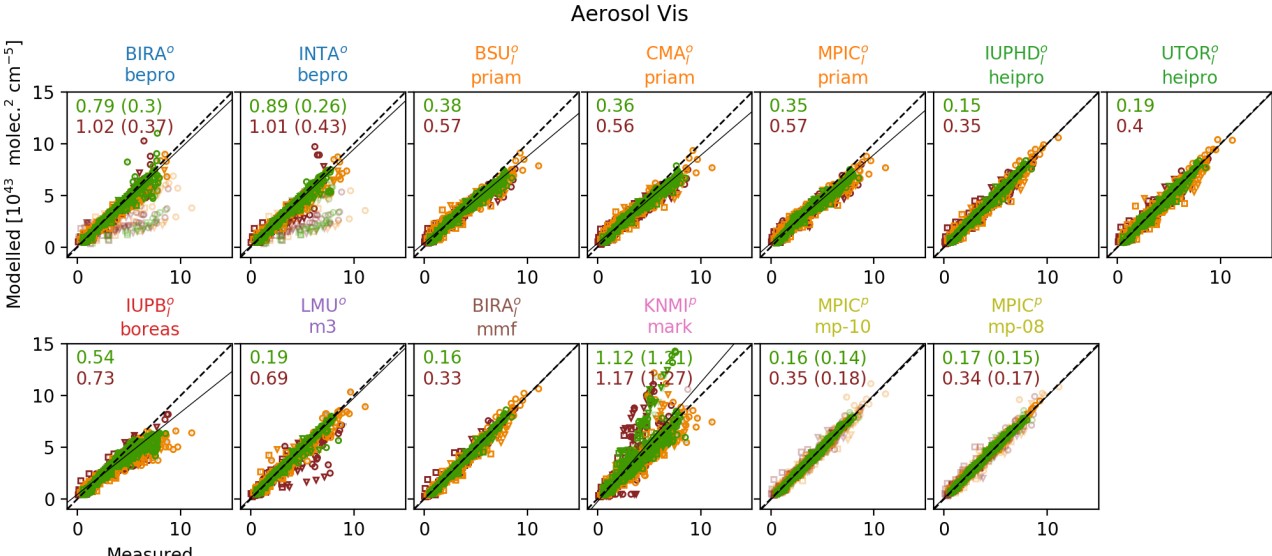

**Figure 9.** O$_4$ Vis dSCD correlation. Legends and description of Fig. 8 apply.

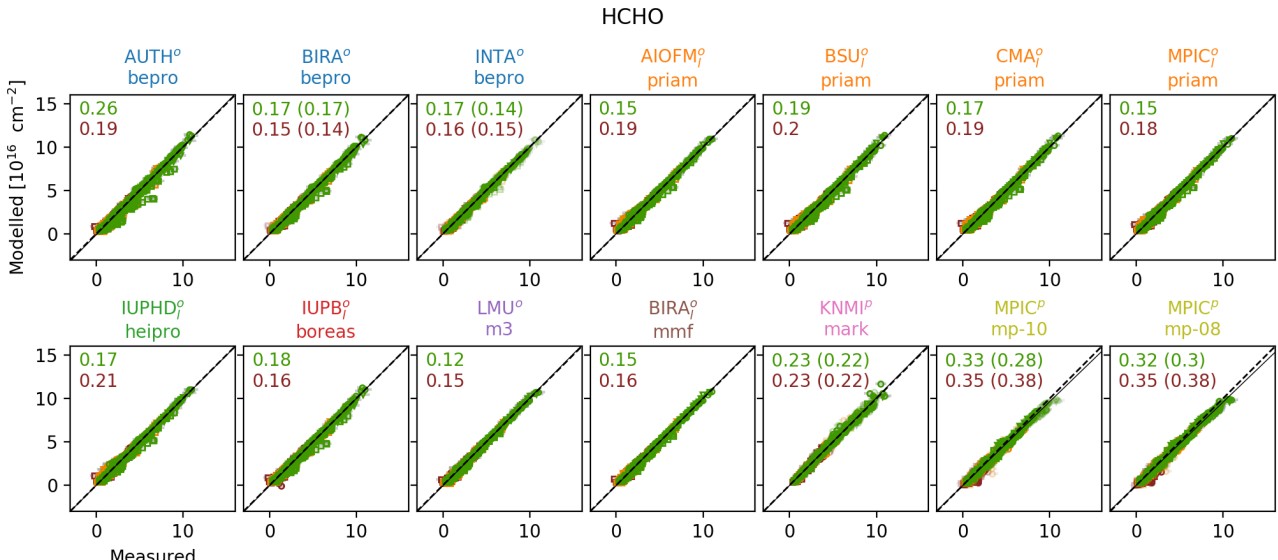

**Figure 10.** HCHO dSCD correlation. RMSD between measured and modelled dSCDs in units of $10^{16}\,\mathrm{molec\,cm^{-2}}$. Legends and description of Fig. 8 apply.

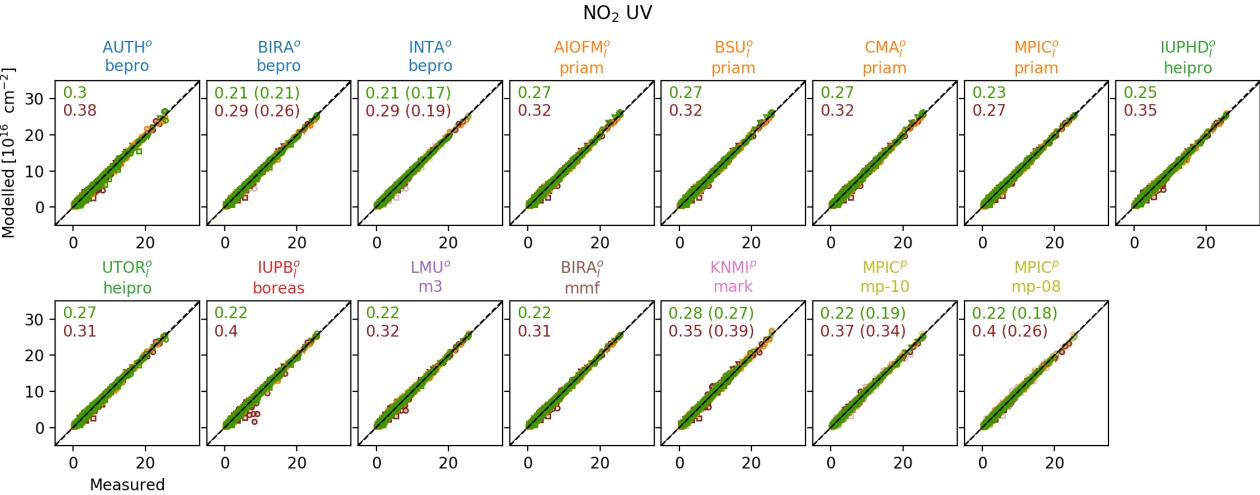

**Figure 11.** $NO_2$ UV dSCD correlation. RMSD between measured and modelled dSCDs in units of $10^{16}\,\mathrm{molec\,cm^{-2}}$. Legends and description of Fig. 8 apply.

## 3.4 Aerosol optical thickness (AOT)

This section compares vertically integrated MAX-DOAS aerosol extinction profiles with the AOTs observed by the nearby sun photometer. As discussed in Sect. 2.3.2 these two quantities are not necessarily comparable. As shown in Sect. 3.1 the

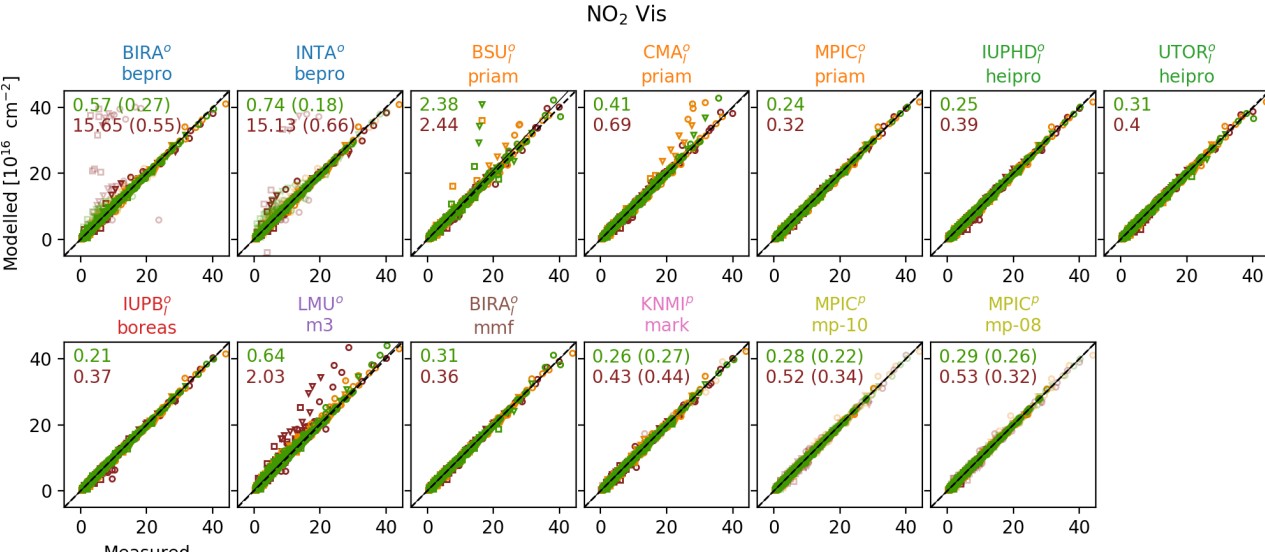

**Figure 12.** NO$_2$ Vis dSCD correlation. RMSD between measured and modelled dSCDs in units of $10^{16}$ molec cm$^{-2}$. Legends and description of Fig. 8 apply.

sensitivity of MAX-DOAS observations decreases rapidly with altitude. Even though the sensitivity to elevated layers was observed to be increased by the presence of optically thick aerosol layers at the corresponding altitudes (Frieß et al., 2006 and Sect. 3.1 of this study), high-altitude abundances of trace gases and aerosol typically cannot be reliably detected by ground-based MAX-DOAS observations. Thus they can only provide "partial AOTs" which basically only consider low-altitude aerosol

and which are additionally biased by *a priori* assumptions on the aerosol extinctions at higher altitudes (for OEM algorithms defined by the *a priori* profile, for PAR algorithms partly in the form of prescribed profile shapes). Therefore, a comparison between MAX-DOAS vertically integrated extinction profiles and sun photometer AOTs $\tau_s$ is not necessarily meaningful. However, for OEM approaches, information on the true aerosol extinction profile $x$ (which are available as described in Sect. 2.2.2) and the AVKs $\mathbf{A}$ can be used to account for this effect: inserting $x$ and $\mathbf{A}$ into Eq. (10) yields a smoothed profile $\widetilde{x}$ that

can be used to estimate which fraction $f_\tau$ of the aerosol column is expected to be detected by the OEM retrievals:

$$f_\tau = \frac{\tau_s'}{\tau_s} = \frac{\sum_i \widetilde{x}_i}{\sum_j x_j} \tag{12}$$

with $\tau_s'$ being the actually detectable "partial AOT". Average values over the whole campaign for $f_\tau$ are $0.81 \pm 0.16$ for Aerosol UV and $0.90 \pm 0.13$ for Aerosol Vis (using the median AVKs of all OEM retrievals). Multiplying the AOT observed by the sun photometer with $f_\tau$ significantly improves the agreement between MAX-DOAS and sun photometer observations in particular

in the UV (see Supplement S2 for details). In the following, this correction is referred to as "partial AOT correction" (PAC). Parametrized and analytical approaches typically do not quantify the sensitivity, the effective resolution or the amount of assimilated *a priori* knowledge. For these algorithms, the correction could not be performed and the total sun photometer AOT





$\tau_s$ had to be used for the comparison in this section. However, the comparison results in this section and further investigations in Supplement S3 indicate that a scaling of the measured $O_4$ dSCDs prior to the retrieval with $SF \approx f_\tau$ might be used to at least partly account for the PAC for MAPA and probably other PAR and ANA algorithms (see Supplement S3), even though the physical reason for PAC and $SF$ are different.

Figure 13 shows time series of the MAX-DOAS retrieved AOTs in comparison to their median and the sun photometer data. For the sun photometer, both the total AOT $\tau_s$ and the partial AOT $\tau_s'$ are shown. For the calculation of $\tau_s'$ in Fig. 13, the median AVKs of all OEM participants were used for the smoothing according to Eq. (10). In the correlation analysis (Fig. 14), AVKs of the individual participants and the individual profiles were applied. Keep in mind that the non-OEM approaches (NASA, KNMI and MPIC/ MAPA) are correlated against $\tau_s$ and might therefore be underprivileged. For correlations of OEM
algorithms against $\tau_s$ please refer to Supplement S8.3. Correlation parameters and RMSD values were derived as described in Sect. 2.3.

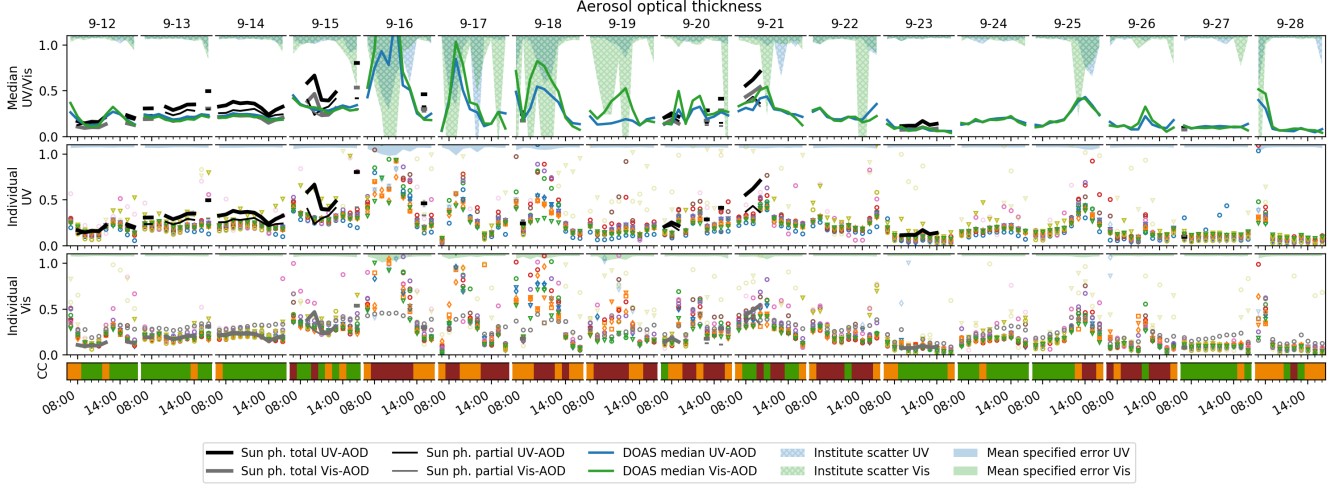

**Figure 13.** MAX-DOAS retrieved AOTs in comparison to sun photometer data. Symbol and symbol colours are chosen according to Table 2. Open symbols indicate data flagged as invalid. Top row: MAX-DOAS median results vs. the available supporting observations, according to the legend below the plot. The "institute scatter" hatched areas (sharing the AOT's y-axis scaling but starting at the top of the plot) show the scattering among the participants in terms of standard deviation with valid data considered only. Two lower rows: Comparison of the individual participants for the two spectral retrieval ranges. Here the coloured area is the average retrieval error, as specified by the participants.

Under clear sky conditions, average RMSD values against the MAX-DOAS median are 0.028 (0.032) for Aerosol UV (Vis). In the presence of clouds they increase by about 30 % (80 %), which is to a large part caused by the periods of particularly large scatter between 16 and 19 September 2016. As already shown in Sect. 3.2, different algorithms detect clouds to very different
extent. Especially in the presence of optically thick clouds (AOT > 10), this easily induces discrepancies of several orders of magnitudes. The observed average RMSDs are similar to the specified uncertainties (average is 0.025) that are derived from

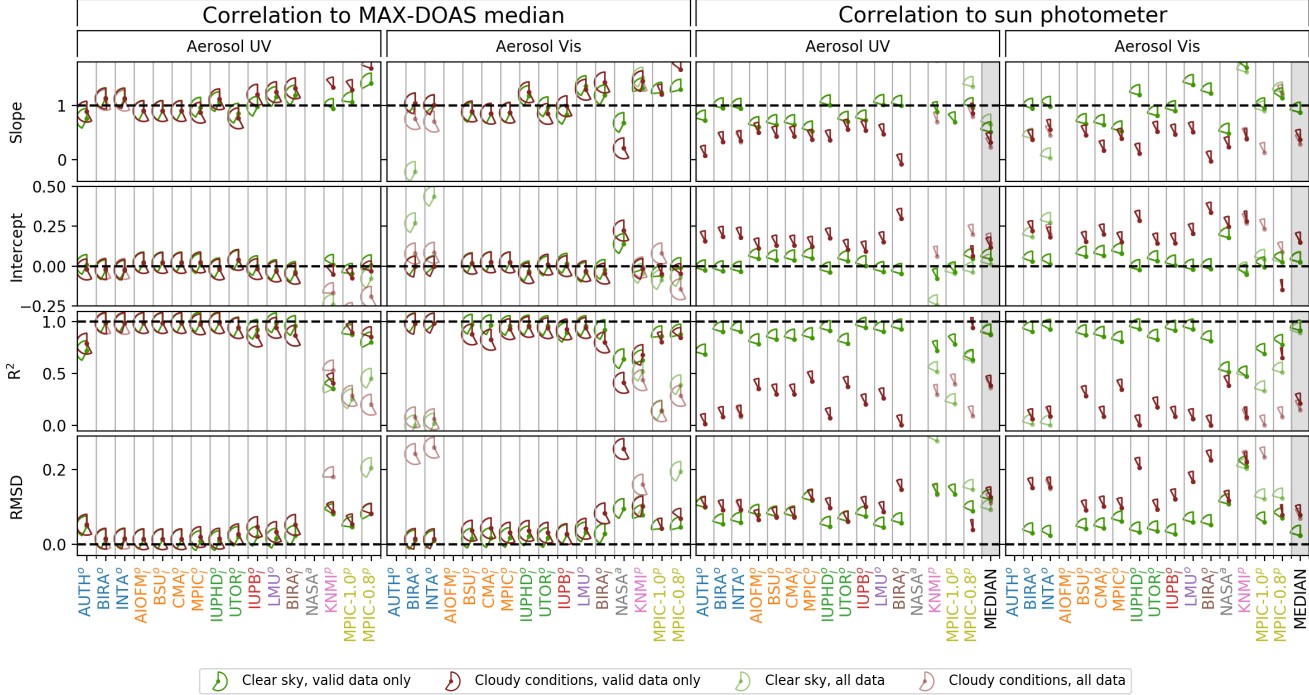

**Figure 14.** Correlation statistics for AOTs. The two left columns give an impression on the agreement among the institutes, as they show the correlation of the individual participant's retrieved AOT (ordinate of the underlying correlation plot) against the median (abscissa). The two right columns show the correlation against the sun photometer AOT (partial AOT in the case of OEM retrievals) instead of the median. Green and red symbols represent cloud-free and cloudy conditions, respectively. Light symbols represent values for all submitted data, opaque symbols only consider data points flagged as valid. The pies indicate, which fraction of the total number of profiles (170) contributed to the respective value. On the right also the correlation between the MAX-DOAS median results and supporting observations are included (grey shaded columns). The correlation plots are shown in Supplement S8.3.

propagated measurement noise and smoothing effects. Keeping in mind that the retrievals were performed on a common dSCD dataset, this indicates that the choice of the retrieval algorithm and the remaining free settings have severe impact on the results.

For the comparison to the sun photometer, it shall be noted that the PAC induces further uncertainties, as it incorporates the extinction profiles derived from the ceilometer and the algorithms' AVKs, both being error-prone. Further, the comparison to
5   sun photometer data under cloudy conditions might not be very meaningful as (1) there are only 13 measurements available in the presence of clouds and (2) as it is very likely that these measurements were made by looking through very local cloud holes, such that they will not be representative for the MAX-DOAS retrieved AOTs with a typical horizontal sensitivity range of several kilometres (see Supplement S6). The following discussion of the sun photometer comparison therefore refers to clear-sky conditions and valid data only. In general, there is reasonable agreement of the MAX-DOAS retrieved AOT with
10  the sun photometer, with average observed RMSDs of 0.08 (0.06) for Aerosol UV (Vis). Good performance is observed for





bePRO (except AUTH), HEIPRO (IUPHD), $M^3$ and MMF with RMSDs around 0.05 (0.03). For other OEM algorithms, larger underestimations of the partial AOT (0.5 < slope < 1) are observed in the UV, which are most evident in the case of PRIAM ($\approx 0.5$). Interestingly, the AVKs at higher layers derived from PRIAM are systematically higher than most other algorithms (see Sect. S8.1), which reduces the impact of the PAC and results in a larger partial AOD $\tau'_s$ than for most other datasets.

Therefore, the lower slopes of PRIAM might rather be owed to its assessment of information content than to the retrieval algorithm itself. For Aerosol Vis bePRO suffers the aforementioned convergence problems during inversion (see Sect. 3.3) but the affected results are reliably flagged. KNMI/ MARK and NASA/ Realtime feature the highest RMSDs around 0.1. A particular case is KNMI/ Aerosol Vis with RMSD> 0.2, with and without flagging being applied.

As described in Supplement S3, the PAC and the application of SF $\approx f_\tau$ have very similar impact on the AOT correlation.

Consequently, the application of an $O_4$ dSCD scaling factor of $SF = 0.8$ significantly improves the agreement to the sun photometer total AOT in the UV ($f_\tau \approx 0.8$) whereas in the Vis ($f_\tau \approx 0.9$) it leads to an overcompensation with slope > 1 and intercept > 0.

### 3.5   Trace gas vertical column densities

This section compares the VCDs of HCHO and $NO_2$. Independent observations of VCDs are the direct-sun DOAS obser-

vations ($NO_2$ and HCHO), but also integrated columns of radiosonde and lidar profiles ($NO_2$). Time series comparisons of all observations are shown in Fig. 15 and 16. For the statistical evaluation in Fig. 17, from the supporting observations only direct-sun observations were considered, as they provide the most complete dataset.

As for AOTs, smoothing effects (in particular the low sensitivity of MAX-DOAS observations for higher altitudes) potentially affects the comparability of MAX-DOAS and direct-sun observations. In contrast to aerosol, only scarce ($NO_2$) or no

(HCHO) information on the true profile is available and a correction similar to the PAC cannot be performed. However for $NO_2$ the available radiosonde profiles could be used for an impact estimate. Ignoring an outlier on 09-27 07:00:00, where $NO_2$ concentration was close to the radiosonde detection limit, correction factors of $1.06 \pm 0.05$ and $1.03 \pm 0.03$ in the UV and Vis are obtained, respectively, indicating that the MAX-DOAS retrieved tropospheric $NO_2$ VCD is affected by smoothing effects to only a few percent. This is expected since $NO_2$ mostly appears close to the ground. Also in Fig. 6 and 7, $NO_2$ appears to be

confined to the lowermost retrieval layers with concentrations dropping to around zero already at altitudes where MAX-DOAS sensitivity is still significant. Profiles from the $NO_2$ lidar were not used in this investigation as they often suffer from artefacts at higher altitudes. Regarding HCHO, the MAX-DOAS profiling results on some days show large concentrations over the whole altitude range where the information content of the measurements is significant (compare Fig. 2 and 5), indicating that there might be "invisible" HCHO at even higher altitudes. This is supported by Fig. 15, where MAX-DOAS observations tend to

yield smaller VCDs than the direct-sun observations in particular in scenarios with high HCHO abundance.

Under clear sky conditions, average RMSD values against the MAX-DOAS median are $5 \times 10^{14} \, \mathrm{molec \, cm^{-2}}$ and $7 \times 10^{14} \, \mathrm{molec \, cm^{-2}}$) for HCHO ($NO_2$, both UV and Vis). In contrast to AOTs, these values do not increase significantly ($< 15\%$) in the presence of clouds. For HCHO it is even reduced by $25\%$ for the same reasons as discussed already in Section 3.2.

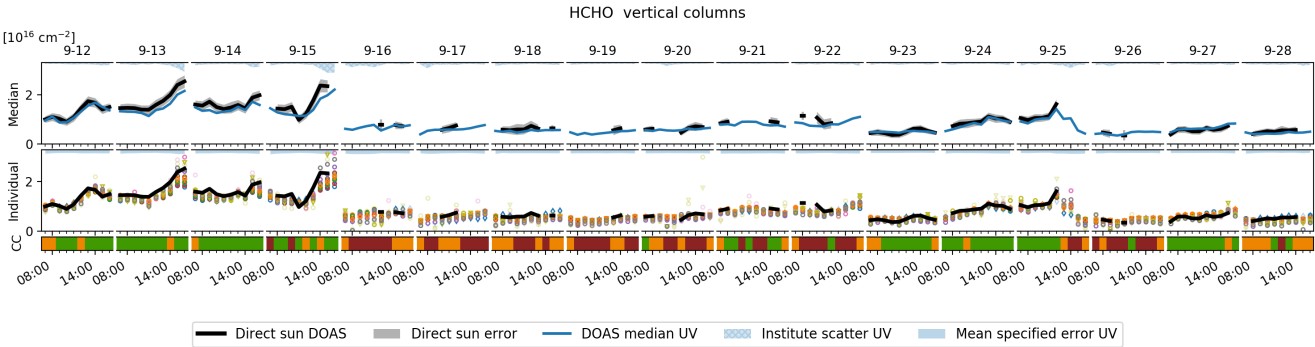

**Figure 15.** Comparison of MAX-DOAS retrieved HCHO VCDs vs. direct-sun DOAS. Basic descriptions of Fig. 13 apply.

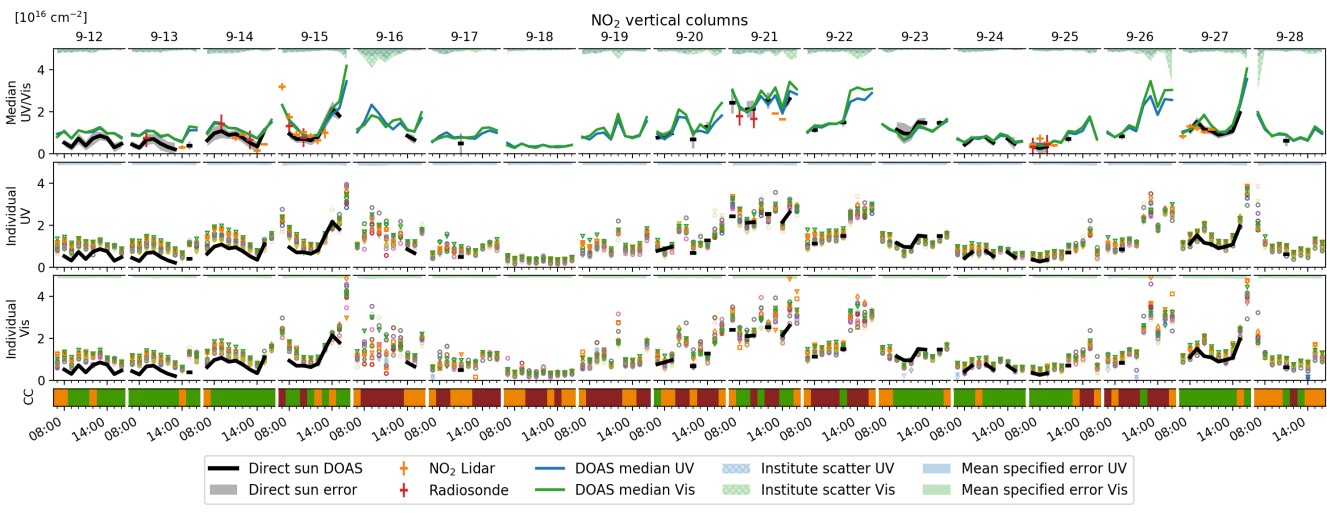

**Figure 16.** Comparison of MAX-DOAS retrieved $NO_2$ VCDs vs. direct-sun DOAS, $NO_2$ lidar and radiosonde. Basic descriptions of Fig. 13 apply.

For HCHO, the comparison against the direct-sun DOAS observations yields an average RMSD of $1.4 \times 10^{15}\,\mathrm{molec\,cm^{-2}}$. Note however that the two observations are not fully independent, as for the direct-sun data, the residual HCHO amount in the reference spectrum was adapted from the MAX-DOAS VCD (see Sect. 2.2.4).

For $NO_2$ UV (Vis) the comparison to the direct-sun DOAS yields an average RMSD of $3.7 \times 10^{15}\,\mathrm{molec\,cm^{-2}}$ ($3.8 \times 10^{15}\,\mathrm{molec\,cm^{-2}}$), which is about five times the average RMSD of the MAX-DOAS median comparison. Between 12 and 14 September the direct sun VCDs but also most radiosonde and lidar observation are systematically lower than the MAX-DOAS VCDs. The reason could not yet be identified. A candidate are the different sampling volumes: while radiosonde, lidar and direct-sun DOAS typically sample air at maximum distances of a few kilometres to the site, the MAX-DOAS instruments have





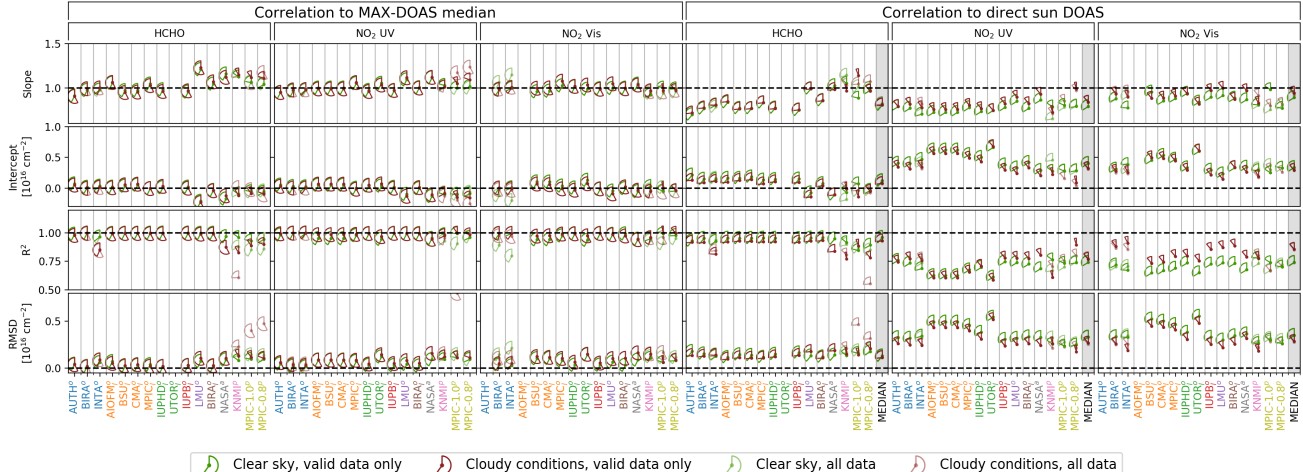

**Figure 17.** Correlation statistics of trace gas VCDs. The plot is similar to Fig. 14. In the underlying correlation plots, ordinates are MAX-DOAS VCDs of individual participants and abscissas are the MAX-DOAS median and direct-sun VCDs, respectively. The correlation plots are shown in Supplement S8.3.

a much larger horizontal sensitivity range (see Supplement S6), even extending to The Hague on some days, which is $> 40\,\mathrm{km}$ away. Indeed the agreement improves with decreasing visibility.

In contrast to the AOTs, the RMSDs against the MAX-DOAS median here are smaller than the specified retrieval errors, which are $1.3 \times 10^{15}\,\mathrm{molec\,cm^{-2}}$, $1.3 \times 10^{15}\,\mathrm{molec\,cm^{-2}}$ and $1.2 \times 10^{15}\,\mathrm{molec\,cm^{-2}}$ for HCHO, NO$_2$ UV and NO$_2$ Vis,
respectively. On the other hand NO$_2$ RMSDs against the direct-sun observations are about three times larger. For the less abundant HCHO, the signal-to-noise ratio (SNR) of the measured dSCDs is smaller, such that the specified uncertainties derived from the dSCD noise are more representative for the actual retrieval accuracy.

## 3.6 Trace gas surface concentrations

This section compares the number concentration of NO$_2$ and HCHO observed at the surface. Note that in this paper "surface
concentration" refers to the average concentration in the lowest MAX-DOAS retrieval layer extending from 0 to 200 m altitude. Independent observations are the LP-DOAS (NO$_2$ and HCHO), and the surface values of radiosonde and lidar profiles (NO$_2$), as well as integrated values of in situ measurements in the tower (described in Sect. 2.2.5). Comparisons of all observations are shown in Fig. 18 and 19. For the statistical evaluation (Fig. 20) only LP-DOAS data were considered since they provides a very accurate, representative and complete dataset. The impact of profile smoothing during the retrieval on the retrieved surface
concentration was estimated for NO$_2$ in Supplement S9 from available radiosonde and lidar NO$_2$ profiles and was found to be around $5.5 \times 10^9\,\mathrm{molec\,cm^{-3}}$ ($4 \times 10^9\,\mathrm{molec\,cm^{-3}}$) in the UV (Vis). Typical RMSD values in the comparison with the LP-DOAS are about one order of magnitude larger, indicating that the impact of smoothing on the NO$_2$ surface concentration is negligible in this study.



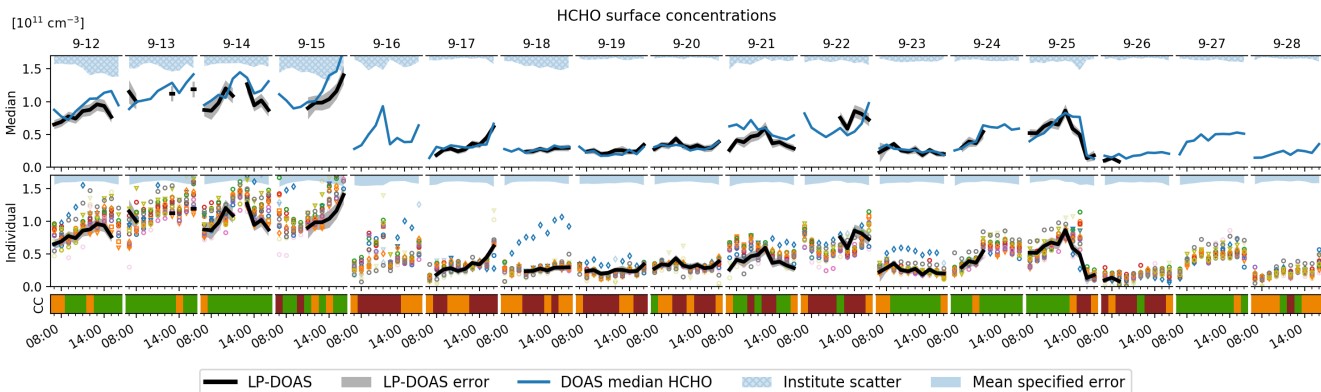

**Figure 18.** Comparison of MAX-DOAS retrieved HCHO surface concentrations. Basic descriptions of Fig. 13 apply.

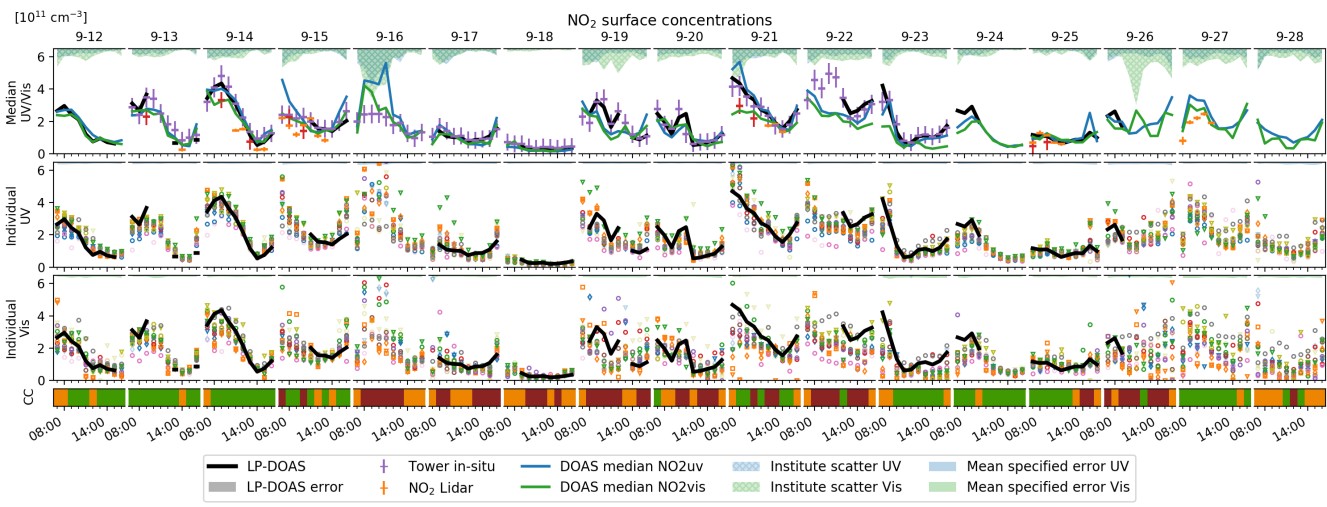

**Figure 19.** Comparison of MAX-DOAS retrieved $NO_2$ surface concentrations. Basic descriptions of Fig. 13 apply.

The comparisons of surface concentrations are particularly useful, because the largest set of validation data is available here and because in contrast to the comparison of AOT and VCDs, the surface concentration comparison also reflects the MAX-DOAS' ability to actually resolve vertical profiles, as it requires an isolation of the surface layer from the layers above.

Figures 18 and 19 show good qualitative agreement between all observations most of the time, even in the presence of clouds.
5 Apparent exceptions for $NO_2$ are the fog event on 16 September (strong scatter among the participants) and at forenoon on 22 September (MAX-DOAS median shows large deviations compared to the tower measurements probably due to a very local $NO_2$ emission event close to the tower).

Under clear sky conditions average RMSDs observed for the comparison to the MAX-DOAS median results are $8.8 \times 10^9 \, \mathrm{molec\,cm^{-3}}$, $1.8 \times 10^{10} \, \mathrm{molec\,cm^{-3}}$ and $2.7 \times 10^{10} \, \mathrm{molec\,cm^{-3}}$ for HCHO, $NO_2$ UV and $NO_2$ Vis, respectively. For the



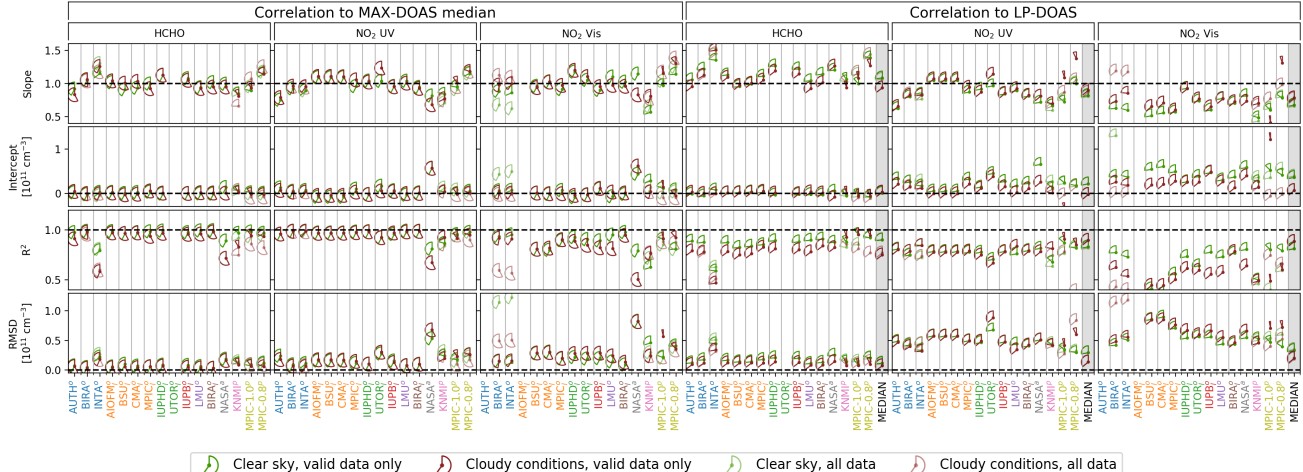

**Figure 20.** Correlation statistics of trace gas surface concentrations. The plot is similar to Fig. 14. In the underlying correlation plots, ordinates are MAX-DOAS surface concentrations of individual participants and abscissas are the MAX-DOAS median and direct-sun VCDs, respectively. The correlation plots are shown in Supplement S8.3.

comparison to the LP-DOAS, they increase to $1.8 \times 10^{10}\,\mathrm{molec\,cm^{-3}}$, $4.7 \times 10^{10}\,\mathrm{molec\,cm^{-3}}$ and $5.6 \times 10^{10}\,\mathrm{molec\,cm^{-3}}$. Clouds have very different impact on these results: the average RMSD to the median increases by 15, 26 and $38\,\%$, whereas the average RMSD to the LP-DOAS is even reduced by 4, 15 and $17\,\%$. A large fraction of the scatter in the comparison to the LP-DOAS might be related to the spatio-temporal variability of the gas concentrations, in particular in the Vis spectral

range, where the MAX-DOAS viewing distance is large. The good agreement of the surface concentrations with the supporting observations during the first days is opposite to the VCD comparison, which at least for $NO_2$ points to a problem with the direct-sun data. For $NO_2$ Vis, the agreement is generally worse than for $NO_2$ UV. Convergence problems of bePRO appear again in the form of outliers (see in particular the RMSD values), which are efficiently removed by flagging. INTA shows strong systematic outliers over whole days (e.g. on 18 September), which are not observed for other bePRO users and are very

likely produced by technical problems. Again the RMSDs to the MAX-DOAS median even for clear-sky conditions are similar or larger than the specified errors (factors of about 1, 2 and 3 for HCHO, $NO_2$ UV, $NO_2$ Vis, respectively).

### 3.7 NO$_2$ UV-Vis comparison

Another intrinsic consistency check for the algorithms, besides the comparison of modelled and measured dSCDs in Sect. 3.3, is the comparison of the $NO_2$ retrieval results in the two different spectral ranges (UV and Vis). These should ideally yield

equal results at least when assuming a horizontally homogeneous atmosphere. Figures 21 and 22 show the correlation of VCDs and surface concentrations.

For the VCDs, the average RMSD is $1.5 \times 10^{15}\,\mathrm{cm^{-2}}$, which increase by $70\,\%$ in the presence of clouds. For clear sky conditions very good agreement (less than $10\,\%$ relative RMSD) is observed for MAPA, M³, MARK, NASA/ Realtime and

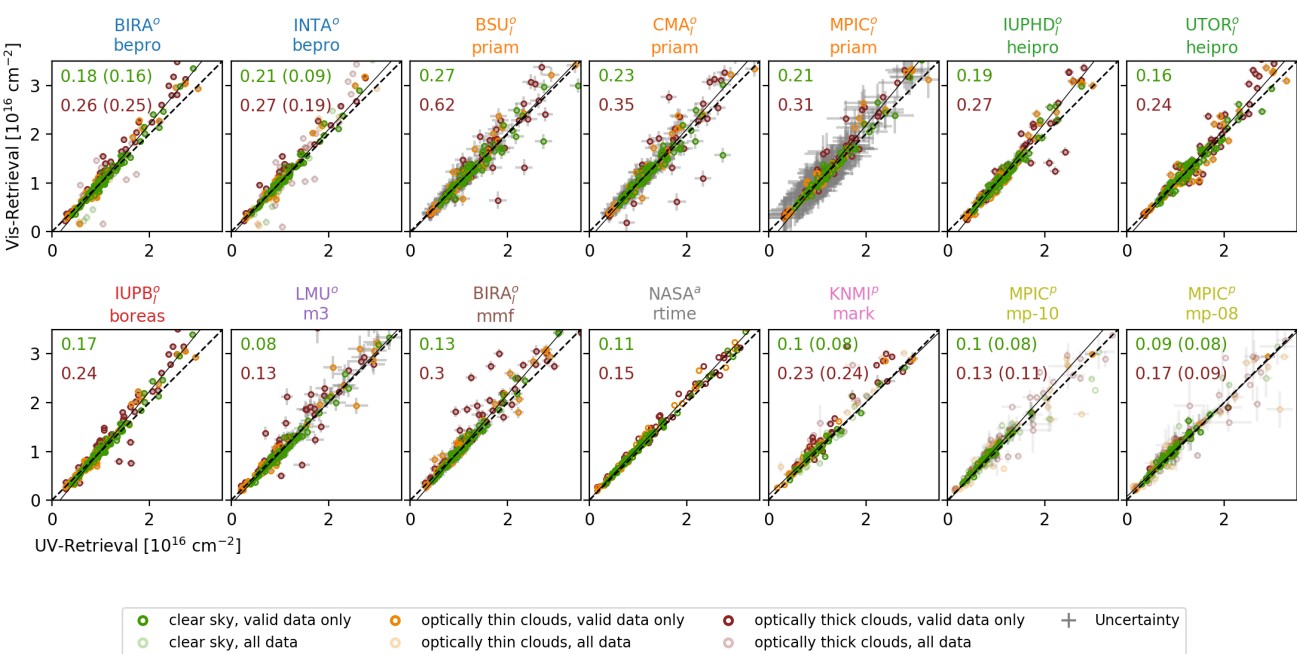

**Figure 21.** Correlation of MAX-DOAS retrieved $NO_2$ VCDs in the UV and the Vis spectral ranges. Marker colours and transparency indicate the cloud conditions and flagging, respectively, according to the legend.

bePRO/ INTA. There is a tendency for Vis VCDs to be larger than UV VCDs (by $6\%$ regarding the campaign averages) which might be caused by the different sensitivity in particular in the retrieval layers between $200\,\mathrm{m}$ and $1\,\mathrm{km}$ altitude. The extended horizontal viewing distance is an unlikely reason since in contrast to the VCDs, surface concentrations in the Vis are smaller than in the UV.

5    For the surface concentrations, the results are very different for the individual algorithms and participants: The average RMSD is $6.0 \times 10^{10}\,\mathrm{cm}^{-3}$ and increases by $25\%$ in the presence of clouds. Best agreement with $10\% < $ relative RMSD $ < 20\%$ are achieved by MAPA, Heipro/IUP-HD, MMF and NASA. Both bePRO users show a similar pattern with systematically smaller values in the Vis retrieval. bePRO suffers from a few strong outliers (even exceeding the plotting range), which are however in most cases removed by flagging. For PRIAM, there is large scatter for all the participants. For HEIPRO, there are

10   large discrepancies between the two participants: while IUPHD achieves very good results here, UTOR shows large scatter (approx. factor of 4) similar to PRIAM users which is once more likely to be explained by the different number of applied iteration steps during the aerosol inversion. The remaining algorithms perform reasonable (relative RMSD $ < 30\%$), apart from few outliers that usually occur under cloudy conditions. Particularly good correlation for both, VCDs and surface concentrations, are only achieved by NASA/ Realtime and MPIC/ MAPA .



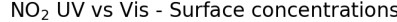

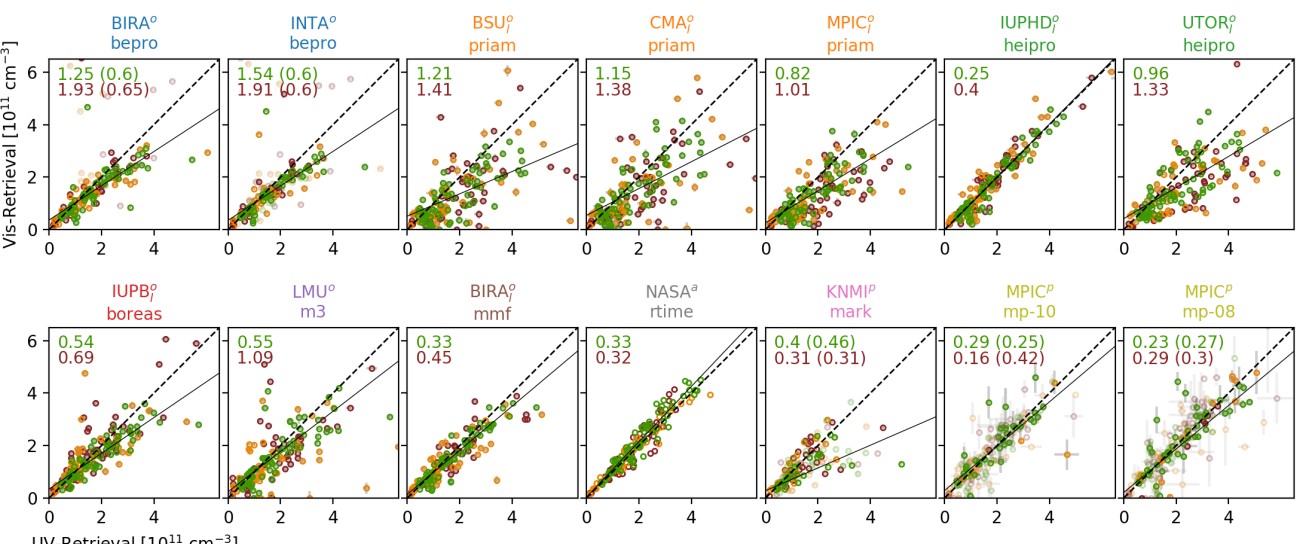

**Figure 22.** Correlation of MAX-DOAS retrieved $NO_2$ surface concentrations in the UV and the Vis spectral ranges. Legends and description of Fig. 21 apply.

## 3.8 Retrieval from dSCDs of individual participants

As described in Sect. 2.1.1, the results compared so far were retrieved from a common set of median dSCDs. Thus, the results only illustrate the performance of the different retrieval techniques. However, it is also interesting to compare collocated MAX-DOAS measurements which are fully independent, to obtain an estimate of the reliability of a typical MAX-DOAS profile

measurement undergoing the whole spectra acquisition and data processing chain. Therefore, the study above was once more conducted with each participant using their own measured dSCDs (see Kreher et al., 2019, for dataset details). The complete results are shown in Supplement S10. A summary is given in Table 5 which shows the increase in average RMSD for the most important comparisons (as described in the precedent subsections for the median dSCDs) when participants use their own instead of the median dSCDs. Only valid data of participants appearing in both studies were considered and BIRA/ bePRO and

KNMI were excluded because in contrast to the median dSCD study BIRA/ bePRO and KNMI did not submit flags for the own dSCD study, which heavily impacted the results.

Regarding only the increase in RMSD in the MAX-DOAS median comparison (hence, the degradation of consistency among the participants) is qualitatively consistent with what one would expect from the findings by Kreher et al. (2019) on the CINDI-2 dSCD consistency: for $NO_2$, almost all participating instruments were able to deliver good quality dSCDs suitable for

profile inversion, while for HCHO the quality was much more variable, resulting in the stronger degradation given in Table 5. Kreher et al. (2019) identified instrumental characterisation (e.g. detector non-linearity and stray-light in the spectrometer) and





**Table 5.** Increase in average RMSD when participants retrieve profiles from their own dSCDs instead from the median dSCDs. Values are given for clear sky and cloudy conditions separately. Further the comparisons among the participants (to the MAX-DOAS median) and the comparisons to the supporting observations are distinguished.

| | | Clear sky | | Cloudy | |
| | | To median [%] | To supp. obs. [%] | To median [%] | To supp. obs. [%] |
| Observation | Species | | | | |
|---|---|---|---|---|---|
| AOT | Aerosol UV | 29 | -10 | 32 | 45 |
| | Aerosol Vis | 29 | 18 | 26 | 21 |
| VCD | HCHO | 175 | 66 | 152 | 46 |
| | NO2 UV | 45 | -8 | 45 | -8 |
| | NO2 Vis | 43 | 6 | 27 | 3 |
| Surface | HCHO | 87 | 16 | 120 | 37 |
| | NO2 UV | 28 | 10 | 25 | 1 |
| | NO2 Vis | 13 | 6 | -9 | -13 |

pointing issues as the main sources of discrepancy between the participant's own dSCD datasets. The degradation is smaller for the surface concentrations than for the trace gas VCDs and is very similar for different cloud conditions.

For the comparison to the supporting observations, the increase in average RMSD is smaller (second and fourth column of Table 5). This means, that even though using the own dSCDs induces differences among the participants, the average quality of the dSCDs is basically maintained. Interestingly, the RMSD for the UV AOT and $NO_2$ VCD even decreases, indicating that the median dSCDs suffer from systematic biases. Under clear sky conditions, low impact ($\leq 10\%$) was found for Aerosol UV AOTs under clear sky conditions and $NO_2$ data products. Particularly large impact is observed for HCHO VCDs ($66\%$). Under cloudy conditions, the impact on $NO_2$ products remains small (again $< 10\%$), whereas for all other products, the increase in average RMSD exceeds $20\%$.

## 4 Conclusions

Within this study, 15 participants used 9 different profiling algorithms with 3 different technical approaches to retrieve aerosol and trace gas ($NO_2$, HCHO) vertical profiles from a common set of dSCDs which was recorded during the CINDI-2 campaign. The results were compared and validated against colocated supporting observations with the aim to assess performance and reliability of individual algorithms but also of the MAX-DOAS profiling technique in general.

Figure 23 shows an overview of RMSD values for the inherent quality indicators (correlations between measured and modelled dSCDs as well as between $NO_2$ UV and Vis results) and the comparisons to available supporting observations (AOT, VCD and surface concentration). General strengths and weaknesses of different algorithms become particularly apparent here. Very good overall performance without the need for validity flagging is achieved by the MMF and the $M^3$ algorithm. Note,





that the results for aerosol are of very similar quality, even though in contrast to $M^3$, MMF retrieves aerosol in the logarithmic space. For valid data (about $20\%$ discarded) INTA also shows good overall performance apart from the outliers in the HCHO surface concentration, which are very likely related to technical problems. Very good performance for aerosol is observed for IUPHD over the full dataset. For $NO_2$, best performance is achieved by MAPA. The AOT comparison looks generally worse

for parametrized approaches which is expected since no partial AOT correction can be performed and thus - with the MAX-DOAS integrated extinction profile and the sun photometer total AOT - basically two different quantities are compared. Finally, the Realtime algorithm by NASA shall be pointed out: despite its simplified radiative transport and the associated outstanding computational performance it provides reasonable results for trace gases (RMSD/ Average RMSD around unity).

Parametrized approaches appear to be less stable in the sense that for less favourable conditions no convergence is achieved

or inconsistent results are returned. For MAPA, these cases are reliably identified and flagged as invalid such that the remaining results achieve very good RMSD values. In contrast for MARK, even some profiles considered valid do not look plausible. The instability of parametrized algorithms is likely related to the approach: in reality, a vertical profile can be described by an arbitrarily large set of parameters and the information on those contained in a MAX-DOAS measurement depends on the atmospheric conditions, hence the profiles themselves. For parametrized approaches, the number of retrieved parameters is

reduced to the number of typically observed DOFs by describing the profile by a few prescribed (not necessarily orthogonal) parameters. Lack of information in those due to particular atmospheric conditions (also if information is available but only on parameters not covered by the chosen parametrization) leads to an under-determined problem with ambiguous solution and the inversion fails. For OEM approaches, the information can be dynamically distributed to a larger number of parameters (20 in this study, namely the species abundances in the retrieval layers) while any lack of information is filled by *a priori* knowledge.

This is why OEM inversions converge under a broader range of atmospheric conditions even when information from the measurement is reduced or shifted between retrieved parameters. On the other hand, this means that OEM algorithms even provide plausibly looking profiles (basically the a priori profile) when few/no information is contained in the measurements. Even though such cases can be identified by examining the AVKs, this makes OEM retrievals prone to misinterpretations particularly by inexperienced users.

Regarding full profiles, the overview plots in Sect. 3.2 show a good qualitative agreement between the algorithms for valid data and clear-sky conditions. In most cases they detect the same features, however sometimes at different altitudes and of different intensity (see also Supplement S8.2). Under clear-sky conditions, the RMSDs between individual participants and the MAX-DOAS median results for AOTs, trace gas VCDs and trace gas surface concentrations range between $0.01 - 0.1$, $(1.5 - 15) \times 10^{14}\,\mathrm{molec\,cm^{-2}}$ and $(0.3 - 8) \times 10^{10}\,\mathrm{molec\,cm^{-3}}$, respectively. For the comparison against supporting obser-

vations, these values increase to $0.02 - 0.2$, $(11 - 55) \times 10^{14}\,\mathrm{molec\,cm^{-2}}$ and $(0.8 - 9) \times 10^{10}\,\mathrm{molec\,cm^{-3}}$, most likely due to (systematic) errors and imperfect spatio-temporal overlap of all observations. The consistency of Aerosol Vis and $NO_2$ Vis products (in particular the agreement among the participants) is typically worse in comparison to their UV counterparts by up to several ten percent. Only the agreement with the sun photometer AOT improves when going from the UV to the Vis spectral range. This might also be related to the reliability of the sun photometer AOTs $\tau_s$: while in the Vis the MAX-DOAS retrieval





wavelength ($477\,\mathrm{nm}$) is close to the lowest sun photometer wavelength channel ($440\,\mathrm{nm}$), in the UV extrapolation of $\tau_s$ down to $360\,\mathrm{nm}$ is required (see Sect. 2.2.1).

The presence of clouds strongly affects the agreement of aerosol retrieval results particularly in the visible spectral range. For AOTs in the UV (Vis) the increase in average RMSD against the median is around $30\,\%$ ($80\,\%$) while RMSDs against the

sun photometer are degraded by $10\,\%$ ($130\,\%$). This is expected as i) high aerosol optical thicknesses at altitudes of low MAX-DOAS sensitivity make the results extremely susceptible to even small changes in the retrieval strategy and ii) the few sun photometer observations under cloudy conditions are likely recorded through local cloud holes and therefore not representative for MAX-DOAS measurements integrating horizontally over several kilometres. In contrast, the impact of clouds on average RMSDs for trace gas VCDs is $< 15\,\%$. Surface concentration RMSDs against the median are degraded by around $25\,\%$, whereas

average RMSDs to supporting observations even decrease.

It could be shown that in the case of CINDI-2, the average impact of smoothing effects on the surface concentration is negligible (Supplement S9). In contrast to that, smoothing has a strong impact on the agreement of MAX-DOAS observations with AOTs and probably HCHO VCDs from supporting observations (Section 2.3.2). In particular, the low sensitivity at higher altitudes has the effect that MAX-DOAS integrated aerosol extinction and sun photometer total AOTs are not necessarily

comparable quantities (Section 3.4 and Supplement S2). Such comparisons can lead to doubtful conclusions if no additional information on the real aerosol distribution is available to perform the necessary corrections.

For CINDI-2 data, there is no clear indication that an $O_4$ dSCD scaling is necessary. On the one hand for OEM algorithms the MAX-DOAS AOT is in good agreement with the sun photometer partial AOT and in contrast to Beirle et al. (2019), we find that a scaling factor of 0.8 is too small (Supplement S3). On the other hand a less extreme scaling ($0.8 < SF < 1.0$) improves

the agreement between forward model and reality (see Fig. S5). $O_4$ scaling and PAC were found to have similar impact on the MAX-DOAS AOT results. Scaling might therefore be used to at least partly replace the PAC in the case of retrieval approaches that do not quantify their sensitivity or the assimilated *a priori* information. At last we think for this study the prescribed $SF = 1.0$ is justified. Even though it might not be ideal, it is the most straightforward approach and yields reasonable and consistent results within the uncertainties introduced by other factors. To draw more concise conclusions, further studies as

performed e.g. by Wagner et al. (2019) are necessary.

In most comparisons, RMSDs of individual participants against the MAX-DOAS median results (even when using the same algorithm) was of the order or larger than the uncertainties specified by the algorithms themselves (up to a factor of 3 for $NO_2$ Vis surface concentrations), indicating that the choice of the retrieval algorithm has severe impact on the results. It shows further, that the specified uncertainties (which typically take propagated measurement noise and smoothing errors into

account but neglect model errors) might be too optimistic as a measure for the MAX-DOAS retrieval accuracy and have to be regarded with care. The discrepancies between the results of the participants using the same algorithm indicate that the retrieval settings that were not prescribed within this study (e.g. number of applied iteration steps in the optimisation process, RTM accuracy options, ...) leave a lot of room for variations. However, technical reasons cannot be fully excluded as the source of the discrepancies. An example appearing in this study are the differences between IUPHD and UTOR (both using HEIPRO)



that were found to mainly be caused by differences in the number of applied iteration steps in the optimisation process of the aerosol inversions.

If the profiles are retrieved from the participant's individually measured dSCDs instead of using a common median dSCD dataset, the agreement of MAX-DOAS results with supporting observations (average RMSD) is degraded by very different
amounts, depending on species and data product. Low impact ($\leq 10\,\%$) was found for Aerosol UV AOTs and $NO_2$ data products. A particularly large impact was observed for HCHO VCDs ($65\,\%$).

Finally, investigations on the spatio-temporal variability (see Supplement S7) indicate that a significant fraction of the RMSD observed between MAX-DOAS and supporting observations is caused by imperfect spatio-temporal overlap. Thus for future campaigns we suggest putting enhanced focus on the coordinated operation of all (not only MAX-DOAS) instruments and to
incorporate techniques with more appropriate spatial kernels, e.g. limb DOAS observations from unmanned aerial vehicles.

*Author contributions.* JLT performed the comparison and the associated investigations as described in the paper and wrote the first draft. UF was involved in the planning of the campaign and the profiling activities, operated the IUPHD instrument, evaluated its data, supervised the comparison activities and contributed in scientific discussions and the manuscript revision. FH was involved in the planning of the campaign and the profiling activities, retrieved profiles for BIRA and contributed in scientific discussions and the manuscript revision. FH,
GP, MVR, AA, AP, AR, TW, KK, UF, JL designed, planned and organized the CINDI-2 campaign. AL/JX/PX, AP, CF/CH/AM/FT/GP/MVR, CZ/KLC/NH/ZW, EP/FW/TB, ES, IB, JJ/JM, KB/XZ, KLC, MY/OPu, SD and TD/AB prepared and operated the MAX-DOAS instrument(s) of AIOFM, KNMI, BIRA, USTC, IUPB, Pandora, BSU, CMA, UTOR, DLR, INTA, MPIC and AUTH, respectively. AL/JX/PX, AP/TV, CA, CF/MVR, CG/FH, CX/HL/KLC, EP/TB/ FW/ AR, ES, IB, JJ/JM, KB/XZ, KLC, LGM/MY/OPu, MMF, SBei, SD, TD/AB, YW and ZW evaluated the MAX-DOAS data for AIOFM, KNMI/MARK, LMU, BIRA, BIRA/bePRO, USTC, IUPB, NASA/Realtime, BSU, CMA,
UTOR, DLR/M3, INTA, BIRA/MMF, MPIC/MAPA, MPIC, AUTH, MPIC/PriAM and DLR/bePRO. AB, AR, CL, KS, MWe, NH and TW supervised the activities of AUTH, Bremen, USTC, UTOR, LMU, DLR and MPIC, respectively. KK as the campaign referee was involved in the actual running of the campaign and the data evaluation up to dSCDs. TW and JK planned and performed the common MAX-DOAS pointing calibration. NH coordinated the cooperation between DLR and USTC. Installation, operation and data evaluation of in-situ $NO_x$ instrumentation was performed by AF/AH (in-situ profile instrumentation in the tower), AM/FT (CAPS) and JL (ICAD/CE-DOAS). BH
calibrated and operated the CIMEL sun photometer that is part of AERONET. DSw, LGa, RVH and SBer operated the $NO_2$ lidar and processed its data into $NO_2$ profiles. SS installed, operated and evaluated the data of the LP-DOAS instrument. DSZ, MA and MDH operated and evaluated the data of the $NO_2$ radiosondes. AC and MT provided and installed the Pandora instruments from which $NO_2$ direct-sun and NASA/Realtime profiling data were deduced. AA, AF, AH, AR, ES, JH, KB, KLC, MMF, MWi, SBei, SS, TB, TW, UP and YW contributed to the scientific discussion and interpretation. AM, AR, CF, CH, FT, GP, JH, JV, MMF, MVR, MWi, SBei, SS, TB, TW, UP and YW revised
and contributed to the manuscript. All authors read and approved the submitted version.

*Competing interests.* The authors declare that they have no conflict of interest.





*Acknowledgements.* We gratefully acknowledge the KNMI staff at Cabauw for their excellent technical and infrastructural support during the campaign. Further we acknowledge EARLINET, CESAR and AERONET for providing data for this study. We acknowledge the authors of the QDOAS package (Caroline Fayt, Michel van Roozendael, Thomas Dankaert). Pandora instrument deployment was supported by Luftblick through ESA Pandonia Project and NASA Pandora Project at Goddard Space Flight Center under NASA Headquarters' Tropospheric Composition Program. We like to thank Airyx GmbH and Dr. Denis Pöhler for supporting measurements with the Airyx GmbH / EnviMeS MAX-DOAS and in-situ instruments. We kindly acknowledge further CINDI-2 participants, who indirectly contributed to the median dSCD dataset and a successful campaign: Abishek Mishra Kumar, Alexander Borovski, Alfonso Saiz-Lopez, Andre Seyler, Andrea Pazmino, Anja Schönhardt, Ermioni Dimitropoulou, Fahim Khokhar, Henning Finkenzeller, Hitoshi Irie, Jeron van Gent, Junaid Khayyam Butt, Manuel Pinharanda, Mareike Ostendorf, Martin Tiefengraber, Mihalis Vrekoussis, Monica Anguas, Monica Navarro-Comas, Moritz Müller, Nader Abuhassan, Nuria Benavent, Paul Johnston, Rainer Volkamer, Richard Querel, Shanshan Wang, Stefan F. Schreier, Syedul Hoque, Theodore K. Koenig, Vinayak Sinha, Vinod Kumar, Xin Tian.

Funding for this study was provided by ESA through the CINDI-2 (ESA Contract No. 4000118533/16/I-Sbo) and FRM4DOAS (ESA Contract No. 4000118181/16/I-EF) projects and partly within the EU 7th Framework Programme QA4ECV project (Grant Agreement no. 607405). The AIOFM group acknowledges the support by the NSFC under project No. 41530644. The participation of the University of Toronto team was supported by the Canadian Space Agency (through the AVATARS project) and the Natural Sciences and Engineering Research Council of Canada (through the PAHA project). The instrument was funded by the Canada Foundation for Innovation and is usually operated at the Polar Environment Atmospheric Research Laboratory (PEARL) by the Canadian Network for the Detection of Atmospheric Change (CANDAC). The activities of the IUP Heidelberg were supported by the DFG project RAPSODI (grant No. PL 193/17-1). INTA acknowledges support from the National funding projects HELADO (CTM2013-41311-P) and AVATAR (CGL2014-55230-R). CMA group acknowledges the support by the NSFC under project Nos. 41805027. The participation of the LMU team was made possible by the DFG Major Research Instrumentation Programme (INST 86/1499-1 FUGG). KLC has received funding from the Marie Curie Initial Training Network of the European 7th Framework Programme (Grant No. 607905) and the European Union's Horizon 2020 research and innovation programme (Grant No. 654109). Support was received from ACTRIS-2 H2020 Grant Agreement Nr. 654109. The CINDI-2 campaign received funding from the Dutch Space Office (NSO).



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







**Figure 23.** Summary of RMSDs from the comparisons in Sect. 3 for clear-sky conditions. The RMSD values of AOT, VCD and surface concentration are calculated with respect to the corresponding supporting observations. Average RMSD values define the colour scale of each column (see colourbar on the top right). White spaces indicate no data. Average observed values (bottom row) are rounded campaign averages of the supporting observations. The column on the far right indicates which fraction of the maximum number (170) of available profiles has been used. Participants who submitted flags are represented by two rows: one considering all data and one using only those flagged as valid ("valid only").