# Peer review of "Intercomparison of MAX-DOAS vertical profile retrieval algorithms: studies on field data from the CINDI-2 campaign"

_Atmospheric Measurement Techniques, 2019_

## Referee Comment (RC1) · Anonymous Referee #1 · 18 Feb 2020

Tirpitz et al presents trace gas concentration (NO2 & HCHO) and aerosol extinction profiles of 15 participating groups derived from MAX-DOAS measurements and implementing different retrieval algorithms during the CINDI-2 campaign. The authors attempt to validate profiles/partial columns using collocated observations. This is an important effort since there are several retrieval approaches using MAX-DOAS measurements, and even though MAX-DOAS measurements started a while ago still there are not harmonized approaches to retrieve gases and aerosols. Hence, this is an important work and likely suitable for the journal. However, I have major comments and foremost revisions are warranted before publication. In my opinion, the quality of the paper needs to be improved before publication.

[Figure]

Major Comments

- According with the manuscript the main goal "is to assess their consistency with respect to different conditions and to review strengths and weaknesses of the individual algorithms and techniques" and they use supporting collocated measurements to "validate" the retrieval algorithms. However, authors include primarily results of retrievals using "median dSCDs" obtained in a separate study (Kreher et al., 2019). I do completely understand the value of using the "median dSCDs" but I also see an extreme value in including detailed results using each participant's dSCDs. The current approach seems quite unusual in a validation point of view. So far, section 3.8 describes briefly results using dSCDs of individual participant but needs to be expanded in the main body, abstract, and conclusions.

- The algorithms are assessed primarily with the root mean square difference. Authors focus primarily on this quantity, which is always positive, and definitively help to understand the comparisons, especially among instruments. However, I highly suggest to include a bias estimator to know the under-or overestimation with respect to the independent measurements. Figure 23 is key in the paper, and I highly suggest to include a similar figure but using bias in percent.

- I find very useful to include the three different type of algorithm approaches (OEM, PAR, and ANA). However, a thorough analysis of what technique yields the best results is missing, especially in the abstract. According with the results, OEM seems to be most appropriate/reliable, but ANA approaches might be ideal for near-real time analysis. I would include a section with main finding regarding the comparison of these methods.

- For the groups using OEM, they use same dSCDs and main retrieval parameters are prescribed, still there are extremely large differences among the groups using OEM. A thorough analysis of the reason is missing. Additionally, If I understand correctly, the recommended altitude grid for all participants was from the surface to 4km (20 layers
of 200 m). This is quite unusual in transfer models, how is the atmosphere represented above 4km? If this is fact true I highly recommend having realistic information above 4km. Furthermore, I am surprise that for the retrieval settings all participants use average values of pressure, temperature, and O3 vertical profiles obtained in 2013-2015. However, the campaign was held in 2016. I believe pressure, temperature, water vapor, etc, might have an important effect in the forward model and foremost in the retrieval of aerosol extinction using O4. I do not understand why radiosondes (or even re-analysis) data obtained during the campaign are not used. If the goal is to validate profiles I highly suggest using the real atmospheric conditions during the campaign.

- It is well-known that sensitivity needs to be considered when comparing different measurement techniques. However, after reading the manuscript it sounds like you introduce new findings, e.g., last short paragraph in the abstract. I do not think it is assumed that integrated extinction profiles from MAX-DOAS and the AOD from the sun photometer should be comparable. In my opinion, this is not a finding or result in this paper. I suggest to re-write your findings accordingly, e.g, include that after smoothing (applying the "AOD correction") comparisons yield better results… in fact, I think authors should describe that this correction (partial OAD correction) is related to the O4 scaling factor used in past studies (and here too for some groups). If I understand correctly, the "AOD correction" yields better results/comparisons because sensitivity in mainly in the lower troposphere, hence aerosol layers aloft are not captured with MAX-DOAS. In this context, after reading Ortega et al. (2016) this reference is not pointed out but offers some insights and should be included.

- It is mentioned that "The ceilometer aerosol extinction profiles should be consulted for qualitative comparison only" and I fully agree due that many assumptions are used to calculate extinction from backscatter measurements. In this context, the aerosol extinction derived from the ceilometer cannot be used to validate the profiles. However, I do believe they offer you additional information that can be further used, especially

for OEM. In the manuscript, a priori extinction profiles for both aerosol and trace gas retrievals were exponentially-decreasing and of course OEM will converge, i.e., it is an ill-posed problem. However, if you use the aerosol extinction profiles as an a priori at least you estimate a better profile shape and the OEM technique might give you a better result. I highly recommend to use the ceilometer extinction profiles as a priori profiles and compare with the exponentially decrease profile. Several questions might arise: do sensitivity increase at higher layers? do AKs change? is the partial AOD correction still the same?

- Lastly, I do not agree that retrievals of NO2 in the UV and vis should give you same results, unless you proof homogeneity around the line of sight.

Specific Comments

P2, L1-6. This paragraph does not belong here, I suggest to move it to the introduction and expand the abstract based on major comments.

P2, L2. Change "boundary layer and the lower troposphere" with "lower troposphere"

P2, L3. Change "radiation" with "absorption"

P2, L5. I would explicitly say that you retrieve aerosol extinction concentration for profiles.

P2, L10. Include all the supporting observations and remove others in the parenthesis.

P2, L15. Do you mean magnitude instead of intensity?

P2, L15-20. Results are shown in root mean square, however, in order to have a more quantitative description please also include the bias in percentage, or the rmsd in percent. Otherwise, it is hard to interpret the magnitude of the differences.

P2, L21-23. It is well-known that different sensitivity needs to be considered when comparing different measurement techniques. I do not think it is assumed that integrated extinction profiles from MAX-DOAS and the AOD from the sun photometer should be

comparable. In my opinion, this is not a finding or result in this paper. There is noth-
ing new on this short paragraph. I suggest to remove this paragraph or re-write your
findings accordingly, e.g, include that after smoothing (applying the AOD correction)
comparisons yield better results due that similar air masses are compared.

P2, L26-28. Transport is missing in your description of chemical composition in the
PBL.

P3, L5. I agree that MAX-DOAS is a well-established technique with information of
absorption signature of trace gases. However, it is misleading because the whole point
of these type of studies is that MAX-DOAS is NOT a well-established technique to
measure accurately gas concentration.

P3, L6, It is mentioned that MAX-DOAS infers information in the boundary layer and
free troposphere. Please include some references for both cases.

P3, L8. I would remove "from the top of the atmosphere (TOA) to the instrument"

P3, L10. Change "Detectable gases are nitrogen dioxide (NO2), formaldehyde
(HCHO). . ." with "Gases that have been analyzed in the UV and visible spectral range
are nitrogen dioxide (NO2), formaldehyde (HCHO). . ."

P3, L18. Change "radiative transport models" with "radiative transfer models".

P3, L19. Change "such" with "of"

P3, L23. What do you mean by different conditions?... Weather conditions, pollution
conditions?

P3, L30. Again, add all supporting instruments and remove "others ". Otherwise,
remove "others".

P5, L16. Mention shortly what other effects, otherwise remove this.

P5, L27. I do not see see how Apituley et al fits in this study.

[Figure]

P5, L28 – P6, L9. As mentioned above, I see the value of using the "median dSCDs", but I strongly suggest to include in detail (and not in the supplement) the retrieval results using their own dSCDs. In fact, I recommend the "median dSCDs" to be included in the supplement if authors believe the manuscript will be lengthily.

P6, L22. How is water vapor profile included in the forward model? is it important? Also, remove the dots after aerosol microphysical properties.

P6, L25. What is p?. Also, I'm surprise to see 4 DOF, for what gas?, is there a referene?

P7, L3. The short OEM description seems awkward. Remove "filling". In general, you have an ill-posed problem and the solution is constrained by an a priori state vector.

P7, L7. It is mentioned that PAR require more memory, and the sentence sounds like this is a limitation. How much memory is needed for such a short campaign? Satellites use look up tables.

P7, L13. "The M3 algorithm by LMU appears as an additional algorithm in our study" looks awkward. What do you mean? Re-write this sentence. Why its description is included in the supplement?

P7, L25. As mentioned in the general comments. I highly suggest using real atmospheric conditions instead of average PTW from other years.

P7, L27. See my comment above regarding the altitude grid, it is not clear what was used above 4km.

P7, L33. My understanding is that the AERONET angstrom exponent (440-675 nm) derived from a single day (14 Sep) is used to extrapolate to 360nm for all days during CINDI-2, is this correct? If this is correct, please explain why you use a single day and not coincident measurements. I expect the angstrom exponent changing unless you have similar aerosol composition.

P8, L25. Remove the "..." in the sentence in parenthesis. Check many other sentences

like this along the manuscript.

P9, L11. Change "true aerosol extinction" with "aerosol extinction". Many assumptions are carried out for the creation of extinction profiles and might not be the true aerosol extinction.

P9, L23. What mean error does the 0.03 RMSD represent?

P9, L25. At the end of section 2.2.2 it is pointed out that "the ceilometer aerosol extinction profiles should be consulted for qualitative comparison only", which I fully agree since many assumptions are carried out to derive extinction profiles. In this case, the retrieval of extinction profiles cannot be fully validated during CINDI-2.

P9, L25. It is mentioned that NO2 profiles from sondes and lidars were carried out sporadically, but include a description of how often. How many sondes were launched?

P12, L15. For the "different observations" do you mean MAX-DOAS and supporting measurements?, or different groups using MAX-DOAS?. Please clarify.

P12, L18. IS xref,t measurement from a reference measurement?, i.e., collocated supporting observation?. Clarify.

P12. While the root mean square difference is useful, this is always positive. I highly recommend to include a bias to see the sign of bias with respect to collocated observations. Simply, use something like this: bias = median(max-doas-reference)/reference when comparing to collocated supporting observations.

P13, L12. It is mentioned that UV and Vis dSCDs should be the same. I disagree, light path in the UV and Vis might be different. Hence, different dSCDs.

P14, Section 2.3.3. I believe you can quantify the spatial mismatch between sonde-MAXDOAS by using the sonde gps information. It might be interesting to see the actual spatial difference.

Section 3.1.

P15, L12. "Figure 2 visualizes the average AVK matrices"... what do you mean by average AVK?. Are these averages of a single group using OE, or average of all groups?

P15, L13. I agree with this "Note, that the AVKs do not necessarily represent the real/ total sensitivity and information content of MAX-DOAS observations as they only consider the gain of information with respect to the a priori knowledge" and I think some literature is missing, e.g., Friess et al (2006) showed that aerosol extinction above 3km can be retrieved using O4 dSCDs measured at different wavelengths. Ortega el al (2016) showed that elevated aerosol layers modify O4 dSCDs, hence some sensitivity of aerosols aloft. In my opinion, this is a clear effect of an ill-posed issue, where an appropriate a priori information is important. In this case, I do not agree with authors claiming that there is not sensitivity of layers aloft, but it is difficult to retrieve layers aloft due to assumptions and less-ideal a priori information.

P15, L27. It is mentioned that "the presence of clouds can increase the sensitivity to higher layers due to multiple scattering and thus light path enhancement in the clouds". If clouds can enhance the sensitivity at higher altitudes, aerosols might have a similar effect, correct?

P28, L3, I would add if a priori information is not reliable at the end of this sentence: "high-altitude abundances of trace gases and aerosol typically cannot be reliably de-tected by ground- based MAX-DOAS observations "

P28, L11. If I understand correctly, in addition to the description provided, the ratio from equation 11 provides you the fraction of the aerosol retrieved by OEM. So, a factor of 0.8 means that about 20% extinction should be aloft, is my interpretation correct? If so, I think this is a very important result and should be further explained. Furthermore, could this fraction be related with the correction factor?

P29, L3. It is mentioned that "a scaling of the measured O4 dSCDs prior to the retrieval with SF $\approx$ f$\tau$ might be used to at least partly account for the PAC for MAPA and probably other PAR and ANA algorithms (see Supplement S3), even though the physical reason for PAC and SF are different.", please explain further and provide the physical differences between PAC and SF. Would it be possible that past correction factors are used due that they miss aerosols aloft, which if I understand correctly might be in agreement with findings in Ortega et el. (2016)?.

P29, L9. "underprivileged" sounds weird, please change it.

P30. figure 14. Please add bias in % (negative/positive) as mentioned above. Additionally, light vs opaque are not distinguishable, maybe using other colors might help? Furthermore, symbols on the two right column plots are not shown in the legend, maybe you meant to use the same symbols?

P30, L10. As suggested above, please include the bias in percent here, in addition to the rmsd.

P31, L1-2. In the text, it would be handy to describe the group (as in Figure 14) and in parenthesis the approach/name) in order to avoid going to table 2 every time. For example, PRIAM is mentioned in line 2 but this is not in figure 14 and table 2 needs to be checked.

P31, L7. KNMI/ MARK and NASA/ Realtime are mentioned as high rmsd, but I also see MPIC being high but not included in the text. So, all parameterization approaches show high rmsd.

P31, L9-12. It seems like the correction factor improves the agreement, but further description is missing. According with your "partial AOT correction" this might be due that PAR approaches miss layers aloft?. I consider this an important finding but is not described.

P35, Section 3.7. I do not agree that NO2 Vis and UV should yield similar results, unless you show with independent measurements that there is homogeneity in the sensitivity range (vertical/horizontal). Rather than an "intrinsic consistency check" I would

use this section to actually assess inhomogeneity. On the other hand, the manuscript is long enough and I would consider removing this section.

P37, Section 3.8. This section is important and deserves more description. A bunch of figures have been thrown in in Supplement S10 but not a complete description. In my opinion, this is a key section to show how reliable are the MAX-DOAS products, hence I also recommend a thorough description of the bias per participant, and not only rmsd.

P38, L11. Please include the approaches. Some people only read conclusions. Profiles are not really assessed, especially for trace gases. I recommend to explicitly describe that lower tropospheric columns are assessed. Figure 23. It is difficult to track what algorithm is used for each group. I suggest to include the algorithm next to the group name, maybe in parenthesis. I suggest to include another figure, similar as Fig. 23, but for the bias in percent.

P40, L20. It is mentioned that "O4 scaling and PAC were found to have similar impact on the MAX-DOAS AOT results." In my opinion, this is a major finding. It is shown that sensitivity needs to be considered when comparing two different remote sensing techniques, and here you have shown that the lower tropospheric column of extinction agrees well with Total column of AERONET when "corrected". This "PAC" is the same as the O4 scaling factor and by reading Ortega et al. (2016) might be due that aerosol layers aloft are normally neglected. I highly recommend to further describe this.

U. Friess, P.S. Monks, J.J. Remedios, A. Rozanov, R. Sinreich, T. Wagner, U. PlattMAX-DOAS O4 measurements: a new technique to derive information on atmospheric aerosols: 2. Modeling studies, J Geophys Res (Atmos), 111 (2006), p. D14203, 10.1029/2005JD006618

---

## Referee Comment (RC2) · Anonymous Referee #2 · 19 Feb 2020

Tirpitz et al. present a thorough assessment of MAX-DOAS profile retrieval algorithms using data collected during the CINDI-2 intercomparison exercise. The work is to this reviewer's knowledge the most comprehensive and up-to-date assessment of MAX-DOAS inversion using field data. As such, the work is worthy of publication. However, the scale of the work presents certain challenges in understanding. Including the supplemental materials, the total work is 106 pages of text figures and references in length. As such it is likely that many readers will not consume it in its entirety. Several seemingly minor or technical conventions adopted for communication are at risk of creating misunderstanding if the work is read only in part. Of critical importance, several possible reasons of discrepancies between MAX-DOAS and other techniques, and among MAX-DOAS inversions are identified and discussed at length yet the assessment of the relative relevance and importance of these is left unclear to the reader. A concise summary of findings should be included in the abstract and

Specific major comments:

1) The authors make use of a number outside measurements (sometimes in combination) for the purposes of "validation". However, a statistical assessment of the validation is not transparent and digested. A summary of the form and source of discrepancies is distinctly lacking. The RMSD approach is adopted by the authors to capture both systemic differences and statistical noise, yet as the authors discuss RMSD sometimes reflects random variations and other times systemic differences. However, this discussion is scattered and not collected and summarized. Some systematic summary is needed. Comparisons to the validation products similar to Figs. 8 – 12 or 21 and 22 would suffice, although ideally the comparison would be more concise.
   a. Supplement 5 gives some indication of the comparison of the differences between different measurement methods. Tables S4 and S5 give some indication of the relative magnitude of RMSD with the specified uncertainties ($\sigma$). However, it is not fully transparent which measurements contribute most to $\sigma$, nor whether the reported RMSD is primarily random or systematic. Systematic differences should be summarized, preferably the remaining residuals after correcting for systematic differences also.
   b. In Sect. 3.8 and Supplement 10 instrument specific dSCDs are used for inversion rather than the median dSCDs. This most closely matches how the inversions would typically be applied. The authors show an impact on RMSD, including for some data products a decrease. However, it is unclear whether the error contribution from the dSCDs or from the inversion is greater or even whether they are similar in magnitude. Quantitative comparison presents several challenges, however, the authors should at least address this question.
2) The authors state that species more than ≈1 km above the MAX-DOAS detectors cannot be reliably detect, but then discuss at length the impacts of signals originating at these altitudes on the retrievals. As such these signals are by demonstrably detected. Rather, the limitation the authors refer to is in determining the magnitude, shape, and location of the relevant signals. The language should be edited to reflect this.
3) Related to points 1 and 2, some of the limitations of inversions are reported as fundamental, when, in fact, they are the result of design decisions. For instance that OEM retrievals tend toward the a priori is not surprising and is a reflection of the

construction of the a priori as well as the covariance matrix. Similarly, that parameterization retrievals fail to capture cases which cannot be described by their limited set of parameters is not surprising either.

Importantly, these examples point to specific improvements which should be made, namely a priori profiles and parameterizations need to be designed to better reflect reality. For OEM retrievals the specification of covariance must also be critically assessed. Statements to this effect are found in the supplement, however, they are fundamental to the findings and should be prominently featured in the main text.

4) The authors report root-mean-square differences, for aerosol optical thickness, trace-gas columns, aerosol extinction, and trace-gas concentrations as absolute errors. The relative magnitude of different errors are also compared as percentages. However, a comparison of root-mean-square differences with the relevant reported median/mean value is lacking. This makes the comparisons difficult to assess outside the particular community of experts.

5) The authors often use parentheses to communicate pairs of results with one value named followed by the second in parentheses followed later by the value of the first and the value of second in parentheses. While this can often be understood it sometimes conflicts with grammatical use of parentheses and in general creates confusion.

Specific Comments

P2 L3 "different atmospheric parameters" is rather vague here, this work deals with "absorbers" and "scatterers" along the light path.

P2 L15 "intensity" here can be misleading in the context of radiation measurements "magnitude" is unambiguous

P2 L22 "… were found to not necessarily being comparable quantities," this is not grammatical, nor is it fully clear what the authors wish to communicate here. The authors compare these quantities and find they must use the PAC. The final paragraph of the abstract should be reworded and expanded, particularly to reflect point 2 above.

P3 L12 "oxygen collision complex" should instead be "oxygen collision induced absorption", a formal complex is unnecessary to explain the absorption and has not been demonstrated to exist in the atmosphere.

P3 L15-16 consultation of the values reported in Kreher et al., suggests that the average full aperture is closer to 20 mrad than 10 mrad.

P3 L26 I assume that "Arnoud et al., 2019 in prep." here and elsewhere is the same work as Apituley et al., 2019 in prep. referred to in Kreher et al., this reference should be updated or eliminated.

P3 L32 Same as previous comment, Wang et al., 2019 in prep. is either no longer in preparation or is not from 2019. This should be updated

P4 Fig1 The map on the right appears to be oriented with North on top, however, this should be marked for clarity. Notably, based on the position of the river in the photo on the left the orientation of the panels is rotated by ≈180° rotation of the map would improve clarity.

P5 L10 see comment above, based on Kreher et al., the FOV is smaller than the elevation angle resolution, but hardly negligible

P5 Eq1 The use of $\lambda$ to denote wavelength is not introduced here or previously

P5 Eq1 This equation is not valid unless the contributions $\sigma_{i,\lambda}S_i(\alpha)$ are summed over the set of contributing absorbers indexed i.

P5 Eqs2-3 $\tau_\lambda$ in Eq 2 is not the same quantity as $\tau_\lambda$ in Eq 1 and this fact is critical to the validity of Eq 3. This should be reflected by a consistent system of symbols.

P6 L14 DSCDs are reported for five data products, however the UV and Vis retrievals of $O_4$ and $NO_2$ retrieve the same chemical species

P6 L24-25 Algorithmically the retrievals are minimizing a cost function as stated at the end of the sentence, this is what the "model parameters are optimized to obtain", "maximum agreement" is not strictly the same as "minimum difference" and should be substituted.

P7 L2 The solutions obtained for the underconstrained problem are not unambiguous. In the case of OEM they are a maximum likelihood estimator predicated on the *a priori* information. Even if *a priori* information is perfect the obtained solution is not unambiguous simply the most likely. The authors should use a different word.

P7 L2-7 *a priori* information is more extensive than the *a priori* profile proper, it also includes the covariance matrix for OEM. This does more than "fill" the lack of information it also defines a portion of the cost function and forms the basis by which likelihood is assessed. This is critical background to understanding the path-dependent results the authors find and should be expanded upon.

P7 L33 the aerosol profiles are "extrapolated" not "interpolated"

P8 L8-9 The definition of the *a priori* covariance as defined here is a predicate to the later findings and should be discussed as such in relevant locations.

P11 L18-20 If I understand correctly, this method of processing gives a large weight to the uppermost one or two measurements available as these measurements define a majority of the relevant layer. Can the authors comment or elaborate?

P12 L8 temperature and pressure should be spelled out here.

P12 L9 Wagner et al., (2019) find effects of up to 7% on the modeled $O_4$ profile when using a standard atmosphere. This could be a significant contributor or the retrieved RMSD, can the authors comment?

P12 L20-25 Is the least-squares regression a minimization of vertical distance or orthogonal distance?

P12 Eq7 $1/N_p$ here should be in parentheses for clarity

P14 L24 replace "not given" with "inaccurate"

P15 L1-2  "$A_{ij}$ describes the sensitivity of the **measured** concentration in the $i^{th}$ layer to **small** changes in the real concentration in the $j^{th}$ layer.,"

P15 Eq11 The coefficient of 12 in this equation seems to be the result of summing over the lowest 12 layers, corresponding to 2.5 km. However, this is not stated.

P15 L16-18 The increase in information content reflects the an increase in the **differential** light path specifically. While this follows from the longer light paths overall, it is the increased differential path which is the source of the information.

P16 Fig 2. The symmetric boxes illustrating are misleading. As the AVK traces demonstrate, the information content moves as well as being "smoothed". The boxes should be centered in a more rational way or else eliminated.

P17 Table 2 Most groups are listed by city, however, Anhui is listed by province, should this not be Hefei?

Figs. 3-7. The red triangles are not readily seen against the color scale.

Figs. 6-7 In the bottom row when only surface measurement are available these are almost imperceptible.

P24 L6 what precisely do the authors mean by "update interval of the jacobians"?

P24 L6-7 Are the larger discrepancies not simply a reflection of the greater DOFS?

P24 11-13 In this section while using the same set of dSCDs how can the authors speak to horizontal inhomogeneity? How would such an inhomogeneity be detected?

P24 L28 Can the authors clarify what they mean by "technical problems" do they think there was some error in the implementation of the protocol?

Figs. 8-12 If there are uncertainties in these graphs as indicated by the legend for Fig 8, they cannot be seen.

P28 L3 As stated above, per the results presented signals aloft can be reliably detected, but not reliably located and/or quantified. Language should be edited to reflect this.

P28 L13-15 On first reading the finding that adjusting MAX-DOAS AOT by the ratio to the sun photometer improves the agreement seems obvious, even tautological. The actual processing as described in the supplement needs to be better reflected in the main text.

P29 L3-4 The authors state "even though the physical reason for PAC and SF are different." This is surprising as it suggests that the authors posit a specific physical reason for SF which is not that for PAC, what is this?

Fig. 13 and other Figs following same format. In the top row, why are the scatters plotted on an inverted axis? Cannot the scatter exceed one? Even quite significantly? Here and elsewhere the hashed and solid shading are not readily distinguishable.

Fig. 14 and other Figs following same format. While I can appreciate what the authors are trying to communicate with the pie chart symbols, the clear and cloudy data are drawn from the same total and the symbols repeat within a given column. This should be simplified in some way.

P31 L9-12 This paragraph in particular demonstrates that aerosol aloft are detectable.

P31 14 The first sentence should be reworded, the VCDs are compared to different standards or "assessed", but the $NO_2$ VCDs are not compared to the HCHO VCDs

Fig. 15 where is the outlier referred to on P31 L21?

P33 L13-14 the LP-DOAS data are described as "very accurate, representative, and complete" while these are likely well supported assessments, such strong statements should be demonstrated or else backed up by a citation.

Fig 19. Sondes are not listed in the legend. Here and elsewhere the color of the lidar and sondes is very challenging to distinguish.

P34 L3 The language here should be more precise. The surface concentration does reflect the ability of MAX-DOAS retrieval to solate the surface layer specifically. However, the isolation and resolution of the surface layer does not imply in and of itself the resolution of the vertical profile above it.

P35 L5-7 How the consistency of the surface concentrations point to a problem in the direct sun data? Is it not equally possible that the MAX-DOAS VCD apart from the lowermost layer are flawed?

P35 L10-11 I believe this final sentence refers to the comparisons in Tables S4 and S5, however, that is not clear in the text.

P36 L1-4 Can this thinking be made more quantitative by reference to the $f_\tau$ for the Vis and UV products?

In the supplement:

P2 L18 the shift to lower altitudes is a simple reflection of the construction of the covariance. This is hinted at on L21, but should be spelt out. As constructed the retrieval does not have uncertainty into which to place the information at higher altitudes, but the information is present in the measurements and is placed at an altitude which is accessible within the constraints of the prescribed covariance.

P4 L12-14 Clear-sky $O_4$ dSCD are not the largest possible, if there is small but non-zero aerosol scattering concentrated at altitudes below the median altitude of photon scattering for a relevant geometry this leads to brightening. Hence why aerosol can appear as increased albedo for satellites.

Fig S11 The color scheme makes this figure very difficult to read.

Fig S12 The distance scale in this figure seems somewhat misleading in light of Fig. S13. The provided exponential curves appear to imply a radical difference in ranging between the Vis and UV, whereas Fig. S13 makes clear that changes in atmospheric conditions are responsible for most of the difference.

Fig. S34 If I understand this figure correctly virtually all data are within two standard deviations, is this not as expected. P33 L6-7 seems to imply something unexpected.

---

## Author Comment (AC3) · 7 May 2020

Please find the replies to the referee's comments in the appended file "ANSWERS_REVIEWER_1.pdf"

Please also note the supplement to this comment:
https://www.atmos-meas-tech-discuss.net/amt-2019-456/amt-2019-456-AC3-supplement.zip
* * *

---

## Author Comment (AC4) · 7 May 2020

Please find the replies to the referee's comments in the appended file "AN-SWERS_REVIEWER_2.pdf"

Please also note the supplement to this comment:
https://www.atmos-meas-tech-discuss.net/amt-2019-456/amt-2019-456-AC4-supplement.zip

———————————————————————

---

## Author Response (AR1)

Please note that the required information has already been uploaded as author comments to the public discussion forum at https://www.atmos-meas-tech-discuss.net/amt-2019-456/#discussion in 4 separate files:

- **Answers to referee 1**
  To be found in: "Reply to anonymous referee #1" → supplement → ANSWERS_REVIEWER_1.pdf

- **Answers to referee 2**
  To be found in: "Reply to anonymous referee #2" → supplement → ANSWERS_REVIEWER_2.pdf

- **Marked-up version of manuscript**
  Either "Reply to anonymous referee #1" or "Reply to anonymous referee #2" → supplement → Latexdiff_Manuscript.pdf

- **Marked-up version of supplements**
  Either "Reply to anonymous referee #1" or "Reply to anonymous referee #2" → supplement → Latexdiff_Supplement.pdf

This document is a merged version of the files listed above.

**Answers to anonymous referee 1**

**General information**

First of all, we would like to gratefully acknowledge the efforts taken by the reviewers to read and revise this extensive manuscript. We are convinced that their comments helped to significantly improve the manuscript regarding comprehensibility and completeness, particularly in the conclusions.

**Document formatting**

- The reviewer's comments are reprinted here in bold face.
- Our answers are given in regular font
- Explicit changes made in the manuscript are in italic font
- Page-, Line-, Section-, etc. numbers refer to the initially submitted (unrevised) manuscript unless stated otherwise.

**Summary on the changes**

Major changes on the manuscript were made regarding abstract, conclusions and section 2.3.1 (on the description of the statistical approaches; most changes were made in the course of the introduction of the "Bias" as described below). Further, Section 3.7 (the comparison of NO2 UV and NO2 Vis results) was completely eliminated and Supplement S2 (on the partial AOT correction) is now embedded into Section 3.4 in the main text (on the comparison of AOTs).

Answering some of the comments required minor revisions throughout the manuscript, of which not all are explicitly mentioned here. For an overview on all the changes taken, please refer to the Latexdiff_Manuscript.pdf and Latexdiff_Supplements.pdf files.

**Answers**

**Tirpitz et al presents trace gas concentration (NO2 & HCHO) and aerosol extinction profiles of 15 participating groups derived from MAX-DOAS measurements and implementing different retrieval algorithms during the CINDI-2 campaign. The authors attempt to validate profiles/partial columns using collocated observations. This is an important effort since there are several retrieval approaches using MAX-DOAS measurements, and even though MAX-DOAS measurements started a while ago still there are not harmonized approaches to retrieve gases and aerosols. Hence, this is an important work and likely suitable for the journal. However, I have major comments and foremost revisions are warranted before publication. In my opinion, the quality of the paper needs to be improved before publication.**

**- According with the manuscript the main goal "is to assess their consistency with respect to different conditions and to review strengths and weaknesses of the individual algorithms and techniques" and they use supporting collocated measurements to "validate" the retrieval algorithms. However, authors include primarily results of retrievals using "median dSCDs" obtained in a separate study (Kreher et al., 2019). I do completely understand the value of using the "median dSCDs" but I also see an extreme value in including detailed results using each participant's dSCDs. The current approach seems quite unusual in a validation point of view. So far, section 3.8 describes briefly results using dSCDs of individual participant but needs to be expanded in the main body, abstract, and conclusions.**

Response:
We fully agree with the reviewer's statement, that the retrieval results from the own dSCDs are of importance. But as mentioned by the reviewer in the specific comments below, discussing both in detail in a single paper goes beyond its scope, so the focus should be on one of the two. As our focus

was on the comparison exclusively of the retrieval algorithms, we consider the median dSCDs to be the better choice.

Nevertheless, we extended the information on the own dSCD comparison in the following ways:
1. A summarising figure similar to Fig. 23 was created also for the own dSCD comparison and is contained in the supplementary material
2. In the corresponding Section (3.8) in the main text, "Bias" values (description below) were added in Table 5. Further, we now directly compare the impact of the use of own dSCDs and the impact of the use of different retrieval algorithms on the consistency among MAX-DOAS participants.
3. Corresponding discussions in the conclusions were extended.
4. Major results of the own dSCD comparison are mentioned in the abstract now

**- The algorithms are assessed primarily with the root mean square difference. Authors focus primarily on this quantity, which is always positive, and definitely help to understand the comparisons, especially among instruments. However, I highly suggest to include a bias estimator to know the under-or overestimation with respect to the independent measurements. Figure 23 is key in the paper, and I highly suggest to include a similar figure but using bias in percent.**

Response:
The "Bias" was introduced as an additional statistical parameter (see section 2.3.1) to capture systematic discrepancies between the individual evaluations (see also response to reviewer #2). It is simply defined as the weighted average of the difference between a pair of compared observations:

$$\sigma_{bias,p} = \frac{1}{N_T} \cdot \frac{1}{\sum_t w_t} \cdot \sum_t w_t \left( x_{p,t} - x_{ref,t} \right)$$

It appears as an additional parameter in the correlation analysis plots (Fig. 14, 17 and 20) and is discussed there. Further, the summarizing figure (Fig.23) was extended by a panel for the bias:

[Figure]

**- I find very useful to include the three different type of algorithm approaches (OEM,**

**PAR, and ANA). However, a thorough analysis of what technique yields the best results is missing, especially in the abstract. According with the results, OEM seems to be most appropriate/reliable, but ANA approaches might be ideal for near-real time analysis. I would include a section with main finding regarding the comparison of these methods.**

Response:
While we agree with the reviewer that it would be desirable to come to a conclusion which technique is "best" we feel that a quantitative statistical investigation on the results grouped by algorithm techniques is not very meaningful, because

1.) PAR and ANA approaches are heavily underrepresented compared to OEM.
2.) A single ANA and two PAR algorithms are not reasonably representative for the general technique. This becomes apparent for instance by looking at the two parameterised approaches which perform extremely different.

However, the advantages and disadvantages of the different techniques are qualitatively discussed in the conclusions (and have slightly been extended in the course of the revision).
Note that also among the authors there is not yet a consensus on the "best" approach, since this strongly depends on the assessment criteria. For the abstract we consider this topic as too complex to be discussed in an understandable and balanced way without going beyond the scope of the manuscript.

**- For the groups using OEM, they use same dSCDs and main retrieval parameters are prescribed, still there are extremely large differences among the groups using OEM. A thorough analysis of the reason is missing.**

Response:
Note that some of the reasons (e. g. in the case of the two HEIPRO participants) were identified, others not. As described in the paper, in detail OEM approaches can actually be implemented in very different ways. We agree that it would be extremely helpful to investigate the reasons for any of the deviations, however, we believe that this is not affordable at this point and out of the scope of a comparison paper, particularly of the given extent.

**Additionally, if I understand correctly, the recommended altitude grid for all participants was from the surface to 4 km (20 layers of 200 m). This is quite unusual in transfer models, how is the atmosphere represented above 4 km?**
**If this is fact true I highly recommend having realistic information above 4 km.**

Response:
We apologize for the misunderstanding. Some aspects here were not well communicated in the manuscript:

One must clearly distinguish between the "RTM grid" and the "retrieval grid". The RTM grid describes how the atmosphere is represented within the radiative transport forward model while the retrieval grid defines at what vertical resolution the actual inversion (e.g. the OEM formalism) is applied. In most retrieval algorithms, the RTM grid is inherently predefined by the developer and cannot be changed offhand (in particular in the case of look-up table approaches). In contrast to the retrieval grid, it typically features a higher resolution (25 m to 100 m layers close to the surface, increasing with altitude) and extends up to 40 to 90 km altitude. Radiosonde profiles of temperature, pressure and ozone were provided from 0 to 90 km altitude and implemented within the constraints of the RTM grid of the individual algorithms.

To make things clearer to the reader we changed the text:

From: *"Pressure, temperature, total air density, and O3 vertical profiles were averaged from O3 sonde measurements performed in De Bilt by KNMI during September months of the years 2013-2015. […] A fixed altitude grid was used for the retrieval, consisting of 20 layers between 0 and 4 km altitude, each with a height Δh = 200 m. The results of the parametrized approaches and OEM algorithms where the exact grid could not be directly implemented, were interpolated/averaged to this grid to simplify the comparison."*

To: *"Pressure, temperature, total air density, and O3 vertical profiles between 0 and 90 km altitude were averaged from O3 sonde measurements performed in De Bilt by KNMI during September months of the years 2013-2015. […] A fixed altitude grid was used for the inversion, consisting of 20 layers between 0 and 4 km altitude, each with a height of Δh = 200 m. The results of the parametrized approaches and OEM algorithms where the exact grid could not readily be applied during inversion, were interpolated/averaged accordingly afterwards. Note that, for radiative transport simulations, the atmosphere was represented by finer (25 m to 100 m layers close to the surface, increasing with altitude) and farther extending (up to 40 to 90 km altitude) grid, inherently (and differently) defined by the individual retrieval algorithms."*

**Furthermore, I am surprise that for the retrieval settings all participants use average values of pressure, temperature, and $O_3$ vertical profiles obtained in 2013-2015. However, the campaign was held in 2016. I believe pressure, temperature, water vapor, etc, might have an important effect in the forward model and foremost in the retrieval of aerosol extinction using O4. I do not understand why radiosondes (or even re-analysis) data obtained during the campaign are not used. If the goal is to validate profiles I highly suggest using the real atmospheric conditions during the campaign.**

Response:
This comment is addressed in our response to the following comment.

**- It is mentioned that "The ceilometer aerosol extinction profiles should be consulted for qualitative comparison only" and I fully agree due that many assumptions are used to calculate extinction from backscatter measurements. In this context, the aerosol extinction derived from the ceilometer cannot be used to validate the profiles. However, I do believe they offer you additional information that can be further used, especially for OEM. In the manuscript, a priori extinction profiles for both aerosol and trace gas retrievals were exponentially-decreasing and of course OEM will converge, i.e., it is an ill-posed problem. However, if you use the aerosol extinction profiles as an a priori at least you estimate a better profile shape and the OEM technique might give you a better result. I highly recommend to use the ceilometer extinction profiles as a priori profiles and compare with the exponentially decrease profile. Several questions might arise: do sensitivity increase at higher layers? do AKs change? is the partial AOD correction still the same?**

Response:
We fully agree that the settings are not optimal and in particular for scientific studies (rather than methodological studies, as presented here), all available information should be used and the suggestions by the reviewer are exactly the way to go.

Yet, it must be considered:

1. The paper aims at the comparison and validation of MAX-DOAS profiles retrieved under typical measurement conditions. This includes using prior information as they are typically available for an arbitrary measurement location and season. Having daily radiosondes, ceilometer data and collocated sun photometer measurements at hand is not a very usual

scenario. In fact, most MAX-DOAS studies have to resort to climatologies for their prior assumptions.

2. Since the MAX-DOAS results are validated by the supported observations (at least qualitatively, in the case of the ceilometer profiles), they need to be kept independent, which is not the case if one observation serves as a priori for the other.

From this point of view, it is not obvious at which point to "stop" the adaption of prior information. Our settings are similarly carefully chosen as for other MAX-DOAS studies and therefore we think they are justified, as long as they are clearly communicated.

The reviewer's questions at the end of the comment can be answered qualitatively:

**Do sensitivity increase at higher layers? do AKs change?**
This depends on the a priori covariance. Since the uncertainty of ceilometer data is surely smaller than that of an exponential profile, the sensitivity and DOFs will decrease.

**Is the partial AOD correction still the same?**
No. Depending on the a priori covariance, aerosol profiles will remain close to the ceilometer profiles in particular at higher altitudes. Since the PAC is based on exactly these ceilometer profile, f_tau will be close to one and the PAC will not have any effect.

**It is well-known that sensitivity needs to be considered when comparing different measurement techniques. However, after reading the manuscript it sounds like you introduce new findings, e.g., last short paragraph in the abstract. I do not think it is assumed that integrated extinction profiles from MAX-DOAS and the AOD from the sun photometer should be comparable. In my opinion, this is not a finding or result in this paper. I suggest to re-write your findings accordingly, e.g, include that after smoothing (applying the "AOD correction") comparisons yield better results.**

Response:
We agree that it is not a "finding" or result that sensitivity needs to be considered. Generally, this is well known and applied. After reading again through former publications, we also found that the low sensitivity at higher altitudes was already suggested e.g. by Irie (2008) and Frieß (2016) to explain the discrepancies between sun photometers and MAX-DOAS observations, but it has not been proven. This information was added now in the beginning of Section 3.4.:

*"In former publications (e.g. Irie et al., 2008; Clémer et al., 2010; Frieß et al., 2016; Bösch et al., 2018) and also during this comparison study, it was found that MAX-DOAS vertically integrated aerosol profiles systematically underestimate AOTs. It has already been proposed by Irie et 5 al. (2008), Frieß et al. (2016) and Bösch et al. (2018) but not proven that this is related to smoothing effects, namely the reduced sensitivity of MAX-DOAS observations to higher altitudes and associated a priori assumptions."*

In any case the last paragraph in the abstract as submitted is pretentious and misleading. We therefore reformulated it in a similar manner:

*"In former publications and also during this comparison study, it was found that MAX-DOAS vertically integrated aerosol extinction coefficient profiles systematically underestimate the AOT observed by the sun photometer. For the first time it is quantitatively shown that for optimal estimation algorithms this can be largely explained and compensated by considering smoothing effects, namely biases arising from the reduced sensitivity of MAX-DOAS observations to higher altitudes and associated a priori assumptions."*

Related statements in the main text were adapted correspondingly.

**In fact, I think authors should describe that this correction (partial AOD correction) is related to the O4 scaling factor used in past studies (and here too for some groups). If I understand correctly, the "AOD correction" yields better results/comparisons because sensitivity in mainly in the lower troposphere, hence aerosol layers aloft are not captured with MAXDOAS. In this context, after reading Ortega et al. (2016) this reference is not pointed out but offers some insights and should be included.**

Response:
Also to us a direct relation between the O4 scaling factor and the PAC seemed obvious in the beginning. However, after reading different publications on this issue (Wagner (2009), Clémer (2010), Ortega (2016) and Wagner (2019)) we believe that the relation is weak for several reasons:

1. The motivations are very different: the application of the PAC is necessary solely for mathematical reasons related to the concept of optimal estimation and prior constraints applied therein. In contrast, all publications listed above compare forward modelled O4 dSCDs (using an atmosphere derived from supporting observations to reproduce the real conditions as good as possible) to measured O4 dSCDs. They do not make use of optimal estimation or a priori profiles similar to those used in our study. Thus their findings are independent from any kind of PAC.
2. The PAC correction factors are dependent on the a priori profile and covariance. In principle, by changing the a priori constraints, any arbitrary correction factors can be generated. The agreement of the CINDI-2 PAC correction factors with typically applied scaling factors (≈0.8) must therefore be considered to be coincidence.
3. Not all discrepancies between MAX-DOAS and sun photometer are explained by the PAC. As shown in our study, biases remain (Figure 14) in the UV, that can indeed be removed by additionally applying a weaker (campaign averaged) O4 dSCD scaling factor of approx. 0.9 (Supplement, Figure S4). It is well possible, that stronger scaling is necessary for individual days.
4. Applying a scaling factor improves the agreement of modelled and measured O4 dSCDs (Supplement, Figure S5). However, we admit that the discrepancies might also be induced by a priori assumptions limiting the scope of the forward model.

This issue is discussed in the paper main text and also in the conclusions.

Regarding point 1, we added further explanations on P29L4:

*"[…]even though the motivation for the application of the PAC and the SF are different: the application of the PAC is necessary solely for mathematical reasons related to the concept of OEM and prior constraints applied therein. In contrast, publications that suggest or discuss the application of an SF (e.g. Wagner et al., 2009; Clémer et al., 2010; Ortega et al., 2016; Wagner et al., 2019) directly compare forward modelled O4 dSCDs (using an atmosphere derived from supporting observations to reproduce the real conditions to best knowledge) to measured O4 dSCDs. They do not make use of optimal estimation or prior constraints similar to those used in our study. Thus their findings can be considered independent from any kind of PAC."*

**- Lastly, I do not agree that retrievals of NO2 in the UV and vis should give you same results, unless you proof homogeneity around the line of sight.**

Response:
We agree. At least the potential inhomogeneity complicates the interpretation. We therefore removed the section according to the reviewer's suggestion below.

**Specific Comments**

**P2, L1-6. This paragraph does not belong here, I suggest to move it to the introduction and expand the abstract based on major comments.**

The paragraph was removed. The introduction already contains a very similar paragraph.

**P2, L2. Change "boundary layer and the lower troposphere" with "lower troposphere"**

The phrase was removed with the above paragraph. The introduction contains a similar statement, there it was corrected.

**P2, L3. Change "radiation" with "absorption"**

Is obsolete, since the corresponding paragraph was removed. A similar sentence in the introduction was corrected.

**P2, L5. I would explicitly say that you retrieve aerosol extinction concentration for profiles.**

We assume that the reviewer meant "aerosol extinction coefficient profiles"(?). Comment is obsolete since the line was removed. However, we adapted corresponding statements in the main text.

**P2, L10. Include all the supporting observations and remove others in the parenthesis.**

Done.

**P2, L15. Do you mean magnitude instead of intensity?**

Yes, changed.

**P2, L15-20. Results are shown in root mean square, however, in order to have a more quantitative description please also include the bias in percentage, or the rmsd in percent. Otherwise, it is hard to interpret the magnitude of the differences.**

Since many different RMSD values are given in the abstract (different species, different observations) we decided to simply add the average observed AOTs, VCDs and surface to simplify the interpretation of all RMSDs. As stated above the bias was introduced, but to obtain a concise abstract we decided to only show RMSDs which reflect both, systematic and random discrepancies at once.

**P2, L21-23. It is well-known that different sensitivity needs to be considered when comparing different measurement techniques. I do not think it is assumed that integrated extinction profiles from MAX-DOAS and the AOD from the sun photometer should be comparable. In my opinion, this is not a finding or result in this paper. There is nothing new on this short paragraph. I suggest to remove this paragraph or re-write your findings accordingly, e.g, include that after smoothing (applying the AOD correction) comparisons yield better results due that similar air masses are compared.**

See our answer in the major comments above.

**P2, L26-28. Transport is missing in your description of chemical composition in the PBL.**

We agree and changed the text from: *"Its chemical composition and aerosol load is determined by gas and particulate matter exchange with the surface and also driven by homogeneous and heterogeneous chemical reactions."*

To: *"Its chemical composition and aerosol load is driven by the exchange with the surface, transport processes and homogeneous and heterogeneous chemical reactions."*

**P3, L5. I agree that MAX-DOAS is a well-established technique with information of absorption signature of trace gases. However, it is misleading because the whole point of these type of studies is that MAX-DOAS is NOT a well-established technique to measure accurately gas concentration.**

We only partly agree. Intercoparison studies are still valuable and necessary, also for well established techniques. On the other hand, such a differentiation is probably too detailed for the first sentence on MAX-DOAS. We replaced "*well-established*" by "*widely used*", which is a weaker statement.

**P3, L6, It is mentioned that MAX-DOAS infers information in the boundary layer and free troposphere. Please include some references for both cases.**

Note that this sentence has been changed by addressing a comment above. Now we state that MAX-DOAS infers information "*on the lower troposphere*". Corresponding references are listed in the manuscript in the three lines directly above (P3, L3-5).

**P3, L8. I would remove "from the top of the atmosphere (TOA) to the instrument"**

Done.

**P3, L10. Change "Detectable gases are nitrogen dioxide (NO2), formaldehyde (HCHO): …" with "Gases that have been analyzed in the UV and visible spectral range are nitrogen dioxide (NO2), formaldehyde (HCHO): …"**

Done.

**P3, L18. Change "radiative transport models" with "radiative transfer models".**

Changed. Also in further occurrences.

**P3, L19. Change "such" with "of"**

We do not understand. "of" does not make sense here (grammatically). We replaced "*numerous such algorithms*" by "*numerous retrieval algorithms*" instead.

**P3, L23. What do you mean by different conditions?... Weather conditions, pollution conditions?**

The major differentiations made during the comparison are w.r.t. cloud conditions and whether flagging of profiles is allowed or not. However, it is not necessary to spell it out at this point of the manuscript. We therefore deleted the phrase "different conditions":

We changed: *"The main objective of this study is to assess their consistency with respect to different conditions and to review strengths and weaknesses […]"*

To: *"The main objective of this study is to assess their consistency and to review strengths and weaknesses [...]"*

**P3, L30. Again, add all supporting instruments and remove "others ". Otherwise, remove "others".**

Done.

**P5, L16. Mention shortly what other effects, otherwise remove this.**

The comment refers to equation (1):
$$\tau_\lambda(\alpha) = \log\left(\frac{I_{\lambda,TOA}}{I_\lambda(\alpha)}\right) = \sum_i \sigma_{i,\lambda}\, S_i(\alpha) + C$$

The variable "C" is a placeholder for a potentially long list of physical and instrumental effects (linear as well as non-linear), that are not of immediate relevance for the actual comparison study. Listing them here might not be very helpful. We think the "C" should still be mentioned to give consideration to them. As a compromise we add one prominent example in brackets.

We changed: *"C represents further terms accounting for other effects than trace gas absorption that will not be further discussed in this context."*

To: *"C represents terms accounting for other instrumental and physical effects than trace gas absorption (for instance scattering on molecules and aerosols)"*

**P5, L27. I do not see see how Apituley et al fits in this study.**

Thanks, we changed that to Apituley, 2020.

**P5, L28 – P6, L9. As mentioned above, I see the value of using the "median dSCDs", but I strongly suggest to include in detail (and not in the supplement) the retrieval results using their own dSCDs. In fact, I recommend the "median dSCDs" to be included in the supplement if authors believe the manuscript will be lengthily.**

As mentioned by the reviewer, discussing both types of data in detail in a single paper is problematic, so the focus should be on one of them. Whether the "own" or the "median dSCDs" are favoured depends on the aim of the paper. As our focus was on the comparison of the retrieval algorithms, the median dSCDs are the right choice. This is motivated in more detail in the manuscript P6, L1-6 (initially submitted version). However, as stated in our answer on the first major comment above, we added some additional information on the own dSCD results.

**P6, L22. How is water vapor profile included in the forward model? is it important? Also, remove the dots after aerosol microphysical properties.**

Most forward models allow to include water vapour. Therefore, we added it to the list. In the UV/Vis, there are a few $H_2O$ absorption bands and the presence of $H_2O$ changes the average Rayleigh scattering cross-section in the atmosphere but the total effect on the dSCDs (and thus the retrieved profiles) is very small. Assuming typical $H_2O$ concentrations encountered during the CINDI-2 campaign, dSCD simulation results with and without $H_2O$ differed by about 0.1 %. It was therefore considered negligible and was not prescribed in the retrieval settings.

We changed: *"(aerosol extinction, trace gas amounts, temperature, pressure, aerosol microphysical properties, ...)"*

To: *"(aerosol extinction, trace gas amounts, temperature, pressure, water vapour and aerosol microphysical properties)"*

**P6, L25. What is p? Also, I'm surprise to see 4 DOF, for what gas? is there a referene?**

p is implicitly defined here to be the DOFS. We made this clearer:
We changed: *"Typically only p = 2 to 4 degrees of freedom for signal (DOFS) […]"*
To: *"Typically only two to four degrees of freedom for signal (DOFS or p) […]"*
DOFS of 4 were actually achieved for NO2 Vis within this study for distinct profiles.

**P7, L3. The short OEM description seems awkward. Remove "filling". In general, you have an ill-posed problem and the solution is constrained by an a priori state vector.**

We revised the description. It is now:
*"Regarding profiles, typically only two to four degrees of freedom for signal (DOFS or p) can be retrieved from MAX-DOAS observations, such that general profile retrieval problems with more than p independent retrieved parameters are ill-posed and prior information has to be assimilated to achieve convergence. For OEM algorithms, this is provided in the form of an a priori profile and associated a priori covariance (Rodgers, 2000), defining the most likely profile and constraining the space of possible solutions according to prior experience. They constitute a portion of the OEM cost function such that with decreasing information contained in the measurements, layer concentrations are drawn towards their a priori values."*

**P7, L7. It is mentioned that PAR require more memory, and the sentence sounds like this is a limitation. How much memory is needed for such a short campaign? Satellites use look up tables.**

The campaign duration is irrelevant. The look up tables are calculated once over the parameter space of interest (realistic atmospheric/measurement scenarios) and can then be applied to any campaign dataset. For the PAR algorithms presented in this study, a look up table for ground-based aerosol and trace gas retrievals at multiple wavelengths requires about 1 GB of memory.

**P7, L13. "The M3 algorithm by LMU appears as an additional algorithm in our study" looks awkward. What do you mean? Re-write this sentence. Why its description is included in the supplement?**

We changed: *"The $M^3$ algorithm by LMU appears as an additional algorithm in our study"*
To: *"Besides the algorithms described therein, our study includes results from the $M^3$ (OEM) algorithm by LMU."*

We first included the description in the main text, however there it appeared out of place and rather distractive, this is why we moved it to the Supplements.

**P7, L25. As mentioned in the general comments. I highly suggest using real atmospheric conditions instead of average PTW from other years.**

See our answer to the corresponding general comments.

**P7, L27. See my comment above regarding the altitude grid, it is not clear what was used above 4km.**

See our answer to the corresponding comment above.

**P7, L33. My understanding is that the AERONET angstrom exponent (440-675 nm) derived from a single day (14 Sep) is used to extrapolate to 360nm for all days during CINDI-2, is this correct? If this is correct, please explain why you use a single day and not coincident measurements. I expect the angstrom exponent changing unless you have similar aerosol composition.**

Yes, this is correct. See our comment on the choice of prior information in the general comments above.

**P8, L25. Remove the "…" in the sentence in parenthesis. Check many other sentences like this along the manuscript.**

Done.

**P9, L11. Change "true aerosol extinction" with "aerosol extinction". Many assumptions are carried out for the creation of extinction profiles and might not be the true aerosol extinction.**

Done.

**P9, L23. What mean error does the 0.03 RMSD represent?**

We forgot the unit here (it's extinction coefficient in $km^{-1}$) and also over which altitude interval this value was calculated.

We changed: "*The average RMSD between scaled ceilometer and Raman lidar profiles is ≈ 0.03.*"

To: "*The average RMSD between scaled ceilometer and Raman lidar profiles up to 4 km altitude is ≈ 0.03 $km^{-1}$.*"

**P9, L25. At the end of section 2.2.2 it is pointed out that "the ceilometer aerosol extinction profiles should be consulted for qualitative comparison only", which I fully agree since many assumptions are carried out to derive extinction profiles. In this case, the retrieval of extinction profiles cannot be fully validated during CINDI-2.**

Yes, we agree with the reviewer's conclusion, this is why we stated that the aerosol extinction profiles should be consulted for qualitative comparison only. To emphasize that the focus is on AOTs, VCDs and surface concentrations, we added corresponding statements in abstract and conclusion:

"*In the presented study, the retrieved CINDI-2 MAX-DOAS trace gas ($NO_2$, HCHO) and aerosol vertical profiles of 15 participating groups using different inversion algorithms are compared and validated against the colocated supporting observations, with the focus on aerosol optical thicknesses (AOTs), trace gas vertical column densities (VCDs) and trace gas surface concentrations.*"

**P9, L25. It is mentioned that NO2 profiles from sondes and lidars were carried out sporadically, but include a description of how often. How many sondes were launched?**

For the radiosondes this is given and referred to: a few lines further down, we reference Supplement S5.2, which includes a list with the details on each radiosonde flight.

For the Lidar we added a sentence: "*This resulted into 25 suitable Lidar profiles recorded on six different days during the campaign.*"

Note, that the exact timing of both observations can also be inferred from the comparison plots of the actual comparison (e.g. Fig. 16 and 19)

**P12, L15. For the "different observations" do you mean MAX-DOAS and supporting measurements?, or different groups using MAX-DOAS?. Please clarify.**

At this point "different observations" refers to any observation. This comprises multiple cases which are subsequently discussed in the same paragraph. The paragraph was revised in the course of the introduction of the "Bias" and should be clearer now.

**P12, L18. IS xref,t measurement from a reference measurement?, i.e., collocated supporting observation?. Clarify.**

It's either the MAX-DOAS median results or a supporting observation. This was clarified in the course of the revision of the paragraph.

**P12. While the root mean square difference is useful, this is always positive. I highly recommend to include a bias to see the sign of bias with respect to collocated observations. Simply, use something like this: bias = median(max-doas-reference)/reference when comparing to collocated supporting observations.**

As stated above, the "Bias" was introduced as an additional statistical quantity.

**P13, L12. It is mentioned that UV and Vis dSCDs should be the same. I disagree, light path in the UV and Vis might be different. Hence, different dSCDs.**

As suggested below, this comparison has been eliminated.

**P14, Section 2.3.3. I believe you can quantify the spatial mismatch between sonde-MAXDOAS by using the sonde gps information. It might be interesting to see the actual spatial difference. Section 3.1.**

We agree that this is useful: we added a table (S6 in the new manuscript version) with the average temporal and spatial mismatches between MAX-DOAS observations and all supporting observations in Supplement S7:

**Table S6.** Estimates for the average spatio-temporal mismatch of different supporting observations w.r.t. to the MAX-DOAS measurements. For the location of the MAX-DOAS observations the centers of mass of the horizontal sensitivity curves from section S6 were used. For the location of sun photometer and DS-DOAS observations, the center of the lines of sight towards the sun up to $2\,km$ atitude were considered.

| Observation | Spatial mismatch [km] | Temporal mismatch [min] |
|---|---|---|
| Sun photometer | 13 | 8 |
| Ceilometer | 11 | 0 |
| DS-DOAS | 13 | 23 |
| $NO_2$-Lidar | 10 | 9 |
| Radiosonde | 6 | 13 |
| LP-DOAS | 10 | 6 |
| In-situ in tower | 11 | 0 |

We further refined our discussion in Supplement S7 according to these numbers and now also present a rough estimate of the impact of spatio-temporal variability on the comparison of NO2 surface concentrations in Sect. 2.3.3.:

*"Table S6 summarizes the spatial and temporal mismatches between MAX-DOAS and supporting observations. Spatial mismatches are of the order of 10 km, temporal mismatches vary between 0 and 20 minutes. Consequently, strong spatio-temporal variations of the observed quantities are expected to induce large discrepancies among the observations, independent of the data quality. Quantitative estimates of the impact on the comparison could only be derived for $NO_2$ surface concentrations and under strong simplifications (for details see Supplement S6) yielding an RMSD of $3.5x10^{10}$ molec $cm^{-3}$. This is indeed of similar magnitude as the average RMSD observed during the comparison (approx. $5x10^{10}$ molec $cm^{-3}$)."*

**P15, L12. "Figure 2 visualizes the average AVK matrices"… what do you mean by average AVK?. Are these averages of a single group using OE, or average of all groups?**

It's the median over participants and the mean over time. This is described in the figure's caption but we also added it to the main text in brackets.

The text reads now: "*Figure 2 visualizes the average AVK matrices (median over participants and mean over time) […]*"

**P15, L13. I agree with this "Note, that the AVKs do not necessarily represent the real/ total sensitivity and information content of MAX-DOAS observations as they only consider the gain of information with respect to the a priori knowledge" and I think some literature is missing, e.g., Friess et al (2006) showed that aerosol extinction above 3km can be retrieved using O4 dSCDs measured at different wavelengths. Ortega el al. (2016) showed that elevated aerosol layers modify O4 dSCDs, hence some sensitivity of aerosols aloft. In my opinion, this is a clear effect of an ill-posed issue, where an appropriate a priori information is important. In this case, I do not agree with authors claiming that there is not sensitivity of layers aloft, but it is difficult to retrieve layers aloft due to assumptions and less-ideal a priori information.**

We reworded several statements on this issue throughout the manuscript to clarify that the low/no sensitivity to higher altitudes is not fundamental. E.g. on P15, L16, where we changed the text:
*"For all species, the sensitivity is limited to about the lowest 1.5 km of the atmosphere."*
To: *"With the a priori profiles and covariances used within this study, the sensitivity is limited to about the lowest 1.5 km of the atmosphere for all species."*

We further added a paragraph to section 2.3.2. (on smoothing effects):

*"It shall be pointed out however, that the sensitivity and spatial resolution is strongly affected by the exact approach that is chosen to solve the ill-posed inversion problem. Frieß (2006) for instance demonstrates, that the sensitivity to higher altitudes can be enhanced by relaxing the prior constraints and by retrieving profiles at several wavelengths simultaneously. Also the sensitivity depends on the atmospheric state: the presence of clouds and aerosols at higher altitudes for instance change the radiative transport and can increase sensitivity particularly to the layers where they reside."*

**P15, L27. It is mentioned that "the presence of clouds can increase the sensitivity to higher layers due to multiple scattering and thus light path enhancement in the clouds". If clouds can enhance the sensitivity at higher altitudes, aerosols might have a similar effect, correct?**

Yes, correct. This is now also mentioned in the text (see answer to the comment before).

**P28, L3, I would add if a priori information is not reliable at the end of this sentence: "high-altitude abundances of trace gases and aerosol typically cannot be reliably detected by ground- based MAX-DOAS observations "**

Note, that according to suggestions by reviewer 2 the wording was changed from:
*"high-altitude abundances of trace gases and aerosol typically cannot be reliably detected […]"*
To: *"high-altitude abundances of trace gases and aerosol typically cannot be reliably located and quantified […]"*

We only partly agree with the statement of the reviewer. It is only right if by "reliable a priori information" the reviewer means "the state of the atmosphere is known before the inversion". Otherwise, there will always be biases, also if a priori profile and covariance perfectly reflect the prior knowledge.

**P28, L11. If I understand correctly, in addition to the description provided, the ratio from equation 11 provides you the fraction of the aerosol retrieved by OEM. So, a factor of 0.8 means that about 20% extinction should be aloft, is my interpretation correct? If so, I think this is a very important result and should be further explained. Furthermore, could this fraction be related with the correction factor?**

Yes, the reviewer's interpretation appears to be correct. This should become clearer now since we embedded Supplement S2 (detailed results of the PAC factors) into the main text Section 3.4. Regarding the relation to the scaling factor please refer to our answer in the general comments.

**P29, L3. It is mentioned that "a scaling of the measured O4 dSCDs prior to the retrieval with SF _ f_ might be used to at least partly account for the PAC for MAPA and probably other PAR and ANA algorithms (see Supplement S3), even though the physical reason for PAC and SF are different.", please explain further and provide the physical differences between PAC and SF. Would it be possible that past correction factors are used due that they miss aerosols aloft, which if I understand correctly might be in agreement with findings in Ortega et el. (2016)?.**

We have to correct this statement regarding the "physical reason", as it is not well-founded. We replaced the sentence and extended the paragraph by a further explanation:

*"[…] even though the motivation for the application of the PAC and the SF are different: the application of the PAC is necessary solely for mathematical reasons related to the concept of OEM and prior constraints applied therein. In contrast, publications that suggest or discuss the application of an SF (e.g. Wagner, 2009; Clémer, 2010; Ortega, 2016; Wagner, 2019) directly compare forward modelled $O_4$ dSCDs (using an atmosphere derived from supporting observations to reproduce the real conditions to best knowledge) to measured $O_4$ dSCDs. They do not make use of optimal estimation or prior constraints similar to those used in our study. Thus their findings can be considered independent from any kind of PAC."*

Regarding the reviewers question **"Would it be possible that past correction factors are used due that they miss aerosols aloft":**
To our knowledge the typically observed disagreement between total AOT observations and MAX-DOAS integrated aerosol profiles has indeed been regarded as another evidence for the need of a scaling factor in some publications but it never was the primary argument.

**P29, L9. "underprivileged" sounds weird, please change it.**

We changed: "*Keep in mind that the non-OEM approaches (NASA, KNMI and MPIC/ MAPA) are correlated against τ_s and might therefore be underprivileged*"

To: "*Keep in mind that the non-OEM approaches (NASA, KNMI and MPIC/ MAPA) are correlated against τ_s and are therefore expected to generally achieve worse agreement*"

**P30. figure 14. Please add bias in % (negative/positive) as mentioned above. Additionally, light vs opaque are not distinguishable, maybe using other colors might help? Furthermore, symbols on the two right column plots are not shown in the legend, maybe you meant to use the same symbols?**

Figure 14 (and similar figures afterwards) were revised accordingly, also considering comments by reviewer 2.

Fig. 14 in the submitted version of our manuscript:

[Figure]

Now:

[Figure]

**P30, L10. As suggested above, please include the bias in percent here, in addition to the rmsd.**

Biases are discussed now. In this particular case:
"All participants except MPIC-0.8/ MAPA underestimate the AOT (Bias < 0.03) in the UV, despite the PAC has been applied for the OEM algorithms. Note that the slopes and intercepts vary significantly among the participants, however, in an anti-correlated manner, finally resulting into similar Bias values. The average Bias values are -0.06 in the UV and 0.02 in the Vis."

**P31, L1-2. In the text, it would be handy to describe the group (as in Figure 14) and in parenthesis the approach/name) in order to avoid going to table 2 every time. For example, PRIAM is mentioned in line 2 but this is not in figure 14 and table 2 needs to be checked.**

We changed this: we now use the "participant/ algorithm"–notation where applicable and otherwise "algorithm (list of participants)" notation.

**P31, L7. KNMI/ MARK and NASA/ Realtime are mentioned as high rmsd, but I also see MPIC being high but not included in the text. So, all parameterization approaches show high rmsd.**

We added MPIC/ MAPA to the list.

**P31, L9-12. It seems like the correction factor improves the agreement, but further description is missing. According with your "partial AOT correction" this might be due that PAR approaches miss layers aloft?. I consider this an important finding but is not described.**

This should be solved since we embedded Supplement S2 (detailed results of the PAC factors) into the main text Section 3.4.

**P35, Section 3.7. I do not agree that NO2 Vis and UV should yield similar results, unless you show with independent measurements that there is homogeneity in the sensitivity range (vertical/horizontal). Rather than an "intrinsic consistency check" I would use this section to**

**actually assess inhomogeneity. On the other hand, the manuscript is long enough and I would consider removing this section.**

Section and corresponding references were removed as suggested.

**P37, Section 3.8. This section is important and deserves more description. A bunch of figures have been thrown in in Supplement S10 but not a complete description. In my opinion, this is a key section to show how reliable are the MAX-DOAS products, hence I also recommend a thorough description of the bias per participant, and not only rmsd.**

See the answer on this issue in the general comments above.

**P38, L11. Please include the approaches. Some people only read conclusions. I recommend to explicitly describe that lower tropospheric columns are assessed. I suggest to include the algorithm next to the group name, maybe in parenthesis. I suggest to include another figure, similar as Fig. 23, but for the bias in percent.**

The whole conclusions were revised, also considering these suggestions.

**Profiles are not really assessed, especially for trace gases.**
Note that the agreement of profiles is assessed (however not discussed) in the Supplements (Fig. S21 to S25). In the main text we focus on those quantities that we have supporting observations for, to not further extend the manuscript.

**Figure 23. It is difficult to track what algorithm is used for each group.**
In the course of the revision of Fig. 23 (see also general comments) we added a list of the algorithm names in the corresponding colours.

**P40, L20. It is mentioned that "O4 scaling and PAC were found to have similar impact on the MAX-DOAS AOT results." In my opinion, this is a major finding. It is shown that sensitivity needs to be considered when comparing two different remote sensing techniques, and here you have shown that the lower tropospheric column of extinction agrees well with Total column of AERONET when "corrected". This "PAC" is the same as the O4 scaling factor and by reading Ortega et al. (2016) might be due that aerosol layers aloft are normally neglected. I highly recommend to further describe this.**

We only partly agree. Particularly, we disagree with the reviewer's statement "This "PAC" is the same as the O4 scaling factor". See our answer in the general comments.

**Answers to anonymous referee 2**

**General information**

First of all, we would like to gratefully acknowledge the efforts taken by the reviewers to read and revise this extensive manuscript. We are convinced that their comments helped to significantly improve the manuscript regarding comprehensibility and completeness, particularly in the conclusions.

**Document formatting**

- The reviewer's comments are reprinted here in bold face.
- Our answers are given in regular font
- Explicit changes made in the manuscript are in italic font
- Page-, Line-, Section-, etc. numbers apply for the initially submitted (unrevised) manuscript unless stated otherwise.

**Summary on the changes**

Major changes on the manuscript were made regarding abstract, conclusions and section 2.3.1 (on the description of the statistical approaches; most changes were made in the course of the introduction of the "Bias" as described below). Further, Section 3.7 (the comparison of NO2 UV and NO2 Vis results) was completely eliminated and Supplement S2 (on the partial AOT correction) has been embedded into Section 3.4 in the main text (on the comparison of AOTs).

Some comments required minor revisions throughout the manuscript, of which not all are explicitly mentioned here. For an overview on all the changes taken, please refer to the Latexdiff_Manuscript.pdf and Latexdiff_Supplements.pdf files.

**Answers**

**Tirpitz et al. present a thorough assessment of MAX-DOAS profile retrieval algorithms using data collected during the CINDI-2 intercomparison exercise. The work is to this reviewer's knowledge the most comprehensive and up-to-date assessment of MAX-DOAS inversion using field data. As such, the work is worthy of publication.**

**However, the scale of the work presents certain challenges in understanding. Including the supplemental materials, the total work is 106 pages of text figures and references in length. As such it is likely that many readers will not consume it in its entirety. Several seemingly minor or technical conventions adopted for communication are at risk of creating misunderstanding if the work is read only in part.**

Response:
We like to thank the reviewer for the commending words. Having addressed the reviewers' comments below and after revision particularly of abstract and conclusions, we are confident that this has improved in the new version of the manuscript.

**Of critical importance, several possible reasons of discrepancies between MAX-DOAS and other techniques, and among MAX-DOAS inversions are identified and discussed at length yet the assessment of the relative relevance and importance of these is left unclear to the reader.**

Response:
The study is meant to be a comparison, in the first instance quantifying the (in-)consistency of the different observations during CINDI-2. Further, likely reasons for the discrepancies were identified.

Of course it is highly desirable to even quantify all these effects, however, we believe that this is not affordable and out of the scope for a comparison paper, particularly of the given extent

Anyway we made corresponding efforts using available data and resources, but not all yielded simple quantitative results. Still we decided to publish them within the supplementary material, since they provide qualitative information which we hold to be of value.

Finally, we agree, that particularly the conclusions lacked quantitative results that are actually assessed during the study. In this regard we revised the conclusions considering the specific comments from both reviewers.

**A concise summary of findings should be included in the abstract.**

In this regard we also revised the abstract, considering the specific comments of both reviewers.

**Specific major comments:**

**1) The authors make use of a number outside measurements (sometimes in combination) for the purposes of "validation". However, a statistical assessment of the validation is not transparent and digested. A summary of the form and source of discrepancies is distinctly lacking. The RMSD approach is adopted by the authors to capture both systemic differences and statistical noise, yet as the authors discuss RMSD sometimes reflects random variations and other times systemic differences. However, this discussion is scattered and not collected and summarized. Some systematic summary is needed. Comparisons to the validation products similar to Figs. 8 – 12 or 21 and 22 would suffice, although ideally the comparison would be more concise.**

Response:
The conclusions were revised as stated above and according to the specific comments below (see also response to reviewer #1).

The "bias" was introduced as an additional statistical parameter (see section 2.3.1) to capture systematic discrepancies:

$$\sigma_{bias,p} = \frac{1}{N_T} \cdot \frac{1}{\sum_t w_t} \cdot \sum_t w_t \left( x_{p,t} - x_{ref,t} \right)$$

It appears now in the correlation analysis plots (Fig. 14, 17 and 20) and is discussed at relevant locations in the manuscript.

The new, summarizing figure (Fig.23) at the very end of the document was extended, amongst others by a panel for the bias:

[Figure]

Comparisons to the validation products similar to Figs. 8 – 12 or 21 and 22 exist and are included in the supplement. In the main text the regression results of these scatter plots are summarised in Figures 14, 17 and 20 for compactness. This way of visualisation was adapted from Frieß (2019) and Kreher (2020).

**a. Supplement 5 gives some indication of the comparison of the differences between different measurement methods. Tables S4 and S5 give some indication of the relative magnitude of RMSD with the specified uncertainties (σ). However, it is not fully transparent which measurements contribute most to σ, nor whether the reported RMSD is primarily random or systematic. Systematic differences should be summarized, preferably the remaining residuals after correcting for systematic differences also.**

Response:
We added the specified uncertainties of each observation in the tables (in brackets behind the corresponding labels). These values now also appear in the conclusions of the main to assess their contribution to the overall RMSD observed between MAX-DOAS and supporting observations.

We decided to not further extend the tables in the Supplement by bias or residual values, since these would then only be assessed w.r.t. to other supporting observations (not w.r.t. the truth). This is however not of major relevance for the main comparison, where particularly in the statistical analysis only single supporting observations are compared to the MAX-DOAS data. We hold it to be sufficient that the reader can draw the systematic and random discrepancies among the supporting observations qualitatively from the scatter plots in the figure above (Fig. S10).

The updated tables are now:

**Table S4.** Comparison of redundant measurements of the NO$_2$ surface concentration (in $10^{11}$ molec cm$^{-3}$). For each pair of observations, the observed scatter (RMS) is compared to the specified uncertainty ($\sigma$).

| | Tower in-situ (0.56) | | Radiosonde (0.50) | | NO$_2$-Lidar (0.13) | |
|---|---|---|---|---|---|---|
| | RMSD | $\sigma$ | RMSD | $\sigma$ | RMSD | $\sigma$ |
| LP-DOAS (0.06) | 0.32 | 0.56 | 1.01 | 0.51 | 0.57 | 0.13 |
| NO$_2$-Lidar (0.13) | 0.72 | 0.57 | 0.40 | 0.52 | - | - |
| Radiosonde (0.50) | 0.99 | 0.78 | - | - | - | - |

**Table S5.** Comparison of redundant measurements of the NO$_2$ total columns (in $10^{16}$ molec cm$^{-2}$). For each pair of observation, the observed scatter (RMS) is compared to the specified uncertainty ($\sigma$).

| | Radiosonde (0.44) | | NO$_2$-Lidar (0.15) | |
|---|---|---|---|---|
| | RMSD | $\sigma$ | RMSD | $\sigma$ |
| DS-DOAS (0.23) | 0.24 | 0.51 | 0.40 | 0.26 |
| NO$_2$-Lidar (0.15) | 0.34 | 0.48 | - | - |

**b. In Sect. 3.8 and Supplement 10 instrument specific dSCDs are used for inversion rather than the median dSCDs. This most closely matches how the inversions would typically be applied. The authors show an impact on RMSD, including for some data products a decrease. However, it is unclear whether the error contribution from the dSCDs or from the inversion is greater or even whether they are similar in magnitude. Quantitative comparison presents several challenges, however, the authors should at least address this question.**

Response:
We agree with the reviewer that this is important information. The most reliable way to determine it would be to evaluate the own dSCD datasets of all participants with the same algorithm (and ideally repeat this with each participating algorithm). However, this would be a large effort compared to the benefit. Note also that the result's general validity is limited: in the case of CINDI-2, the experience with the MAX-DOAS technique varied strongly among the participants. The quality of the own dSCDs might therefore not be representative for MAX-DOAS observations performed by experienced groups.

We therefore chose a simpler approach to obtain corresponding estimates. We explain it in the the following paragraph added to Section 3.8.:

*"It is also of interest to explicitly estimate which fractions of the total observed discrepancies among the different MAX-DOAS profiling results are caused either by the use of different retrieval algorithms or by inconsistencies in the dSCD acquisition. Note that the RMSD values from the median dSCD comparison represent the error arising solely from using different algorithms while the RMSD values from the own dSCD comparison represent the combined effects of both aspects. For simplicity, we assume that the contributions of both aspects are random and independent so that the effect of using own dSCDs can be isolated by simple RMSD error calculations. In this way, its contribution to the total variance observed among the participants under clear sky conditions can be estimated to 40 % (for AOTs), 85 % (HCHO VCDs), 70 % (HCHO surface concentrations), 50 % (NO$_2$ VCDs), 40 % (NO$_2$ UV surface concentrations) and 20 % (NO$_2$ Vis surface concentrations), respectively. The residual variance can be attributed to the choice and setup of the retrieval algorithm."*

We also added a corresponding discussion in the conclusions.

**2) The authors state that species more than ≈1 km above the MAX-DOAS detectors cannot be reliably detect, but then discuss at length the impacts of signals originating at these altitudes on**

the retrievals. As such these signals are by demonstrably detected. Rather, the limitation the authors refer to is in determining the magnitude, shape, and location of the relevant signals. The language should be edited to reflect this.

Response:
Corresponding statements were adapted throughout the manuscript.

**3) Related to points 1 and 2, some of the limitations of inversions are reported as fundamental, when, in fact, they are the result of design decisions. For instance that OEM retrievals tend toward the a priori is not surprising and is a reflection of the construction of the a priori as well as the covariance matrix. Similarly, that parameterization retrievals fail to capture cases which cannot be described by their limited set of parameters is not surprising either. Importantly, these examples point to specific improvements which should be made, namely a priori profiles and parameterizations need to be designed to better reflect reality. For OEM retrievals the specification of covariance must also be critically assessed. Statements to this effect are found in the supplement, however, they are fundamental to the findings and should be prominently featured in the main text.**

Response:
Corresponding statements in the main text were adapted and extended to better describe the role of a priori profiles and covariance and to emphasize, that the limitations of inversions depend on their choice (see also specific comments below). Further, we moved Supplement 2 (describing the PAC results) to the main text, providing additional insight on these aspects.

**4) The authors report root-mean-square differences, for aerosol optical thickness, trace-gas columns, aerosol extinction, and trace-gas concentrations as absolute errors. The relative magnitude of different errors are also compared as percentages. However, a comparison of root-mean-square differences with the relevant reported median/mean value is lacking. This makes the comparisons difficult to assess outside the particular community of experts.**

Response:
Note, that all these information is included in the summarising Figure 23. However, we also added a corresponding sentence to the abstract as well as to the conclusions:

*"These values compare to approximate average optical thicknesses of 0.3, trace gas vertical columns of $90 \times 10^{14}$ molec cm$^{-2}$ and trace gas surface concentrations of $11 \times 10^{10}$ molec cm$^{-3}$ observed over the campaign period."*

**5) The authors often use parentheses to communicate pairs of results with one value named followed by the second in parentheses followed later by the value of the first and the value of second in parentheses. While this can often be understood it sometimes conflicts with grammatical use of parentheses and in general creates confusion.**

Response:
We revised corresponding passages.

**Specific Comments**
**P2 L3 "different atmospheric parameters" is rather vague here, this work deals with "absorbers" and "scatterers" along the light path.**

We appreciate the reviewer's comment, however it is now obsolete for the abstract, since reviewer 1 suggested to completely remove the paragraph. A similar sentence appears in the introduction. There, it was corrected.

**P2 L15 "intensity" here can be misleading in the context of radiation measurements "magnitude" is unambiguous**

Done

**P2 L22 "… were found to not necessarily being comparable quantities," this is not grammatical, nor is it fully clear what the authors wish to communicate here. The authors compare these quantities and find they must use the PAC. The final paragraph of the abstract should be reworded and expanded, particularly to reflect point 2 above.**

The whole paragraph was revised, also on request of reviewer 1. It now reads:

*"In former publications and also during this comparison study, it was found that MAX-DOAS vertically integrated aerosol extinction coefficient profiles systematically underestimate the AOT observed by the sun photometer. For the first time it is quantitatively shown that for optimal estimation algorithms this can be largely explained and compensated by considering smoothing effects, namely biases arising from the reduced sensitivity of MAX-DOAS observations to higher altitudes and associated a priori assumptions."*

Related statements in the main text were adapted accordingly.

**P3 L12 "oxygen collision complex" should instead be "oxygen collision induced absorption", a formal complex is unnecessary to explain the absorption and has not been demonstrated to exist in the atmosphere.**

Done

**P3 L15-16 consultation of the values reported in Kreher et al., suggests that the average full aperture is closer to 20 mrad than 10 mrad.**

This is true regarding the instruments participating in the CINDI-2 campaign. Yet, for MAX-DOAS profiling applications typically a smaller FOV of <= 10 mrad is desired. As a compromise we wrote "*10-20 mrad*".

**P3 L26 I assume that "Arnoud et al., 2019 in prep." here and elsewhere is the same work as Apituley et al., 2019 in prep. referred to in Kreher et al., this reference should be updated or eliminated.**

We like to thank the reviewers for pointing this out and updated the reference to "*Apituley et al. 2020 in prep.*"

**P3 L32 Same as previous comment, Wang et al., 2019 in prep. is either no longer in preparation or is not from 2019. This should be updated**

Meanwhile Wang et al. is under review at AMTD. The reference was updated accordingly.

**P4 Fig1 The map on the right appears to be oriented with North on top, however, this should be marked for clarity. Notably, based on the position of the river in the photo on the left the orientation of the panels is rotated by ≈180° rotation of the map would improve clarity.**

A mark for indicating north direction was added to the map.

**P5 L10 see comment above, based on Kreher et al., the FOV is smaller than the elevation angle resolution, but hardly negligible.**

Changed from "*the telescope's FOV is usually negligible compared to the elevation angle resolution*" to "*ideally the telescope's FOV is negligible compared to the elevation angle resolution*"

**P5 Eq1 The use of λ to denote wavelength is not introduced here or previously**

We changed the text from: "*The very initial data in the MAX-DOAS processing chain are spectra of scattered skylight $I_\lambda(\alpha)$ [...]*"

To: "*The very initial data in the MAX-DOAS processing chain are intensities of scattered skylight $I_\lambda(\alpha)$ at different wavelengths λ [...]*"

**P5 Eq1 This equation is not valid unless the contributions $\sigma_{i,\lambda}*S_i(\alpha)$ are summed over the set of contributing absorbers indexed i.**

We agree with the reviewer, the sum was inserted.

Instead of:
$$\tau_\lambda(\alpha) = \log\left(\frac{I_{\lambda,TOA}}{I_\lambda(\alpha)}\right) = \sigma_{i,\lambda} S_i(\alpha) + C$$

We now have:
$$\tau_\lambda(\alpha) = \log\left(\frac{I_{\lambda,TOA}}{I_\lambda(\alpha)}\right) = \sum_i \sigma_{i,\lambda} S_i(\alpha) + C$$

**P5 Eqs2-3 $\tau_\lambda$ in Eq 2 is not the same quantity as $\tau_\lambda$ in Eq 1 and this fact is critical to the validity of Eq 3. This should be reflected by a consistent system of symbols.**

We changed "$\tau_\lambda$" to "$\Delta\tau_\lambda$"

**P6 L14 DSCDs are reported for five data products, however the UV and Vis retrievals of O4 and NO2 retrieve the same chemical species.**

We made this clearer by changing the text from:

"*DSCDs were provided for five species, namely $O_4$ UV, $O_4$ Vis, HCHO, $NO_2$ UV and $NO_2$ Vis, where "UV" and "Vis" indicate different DOAS spectral fitting ranges in the ultraviolet and the visible spectral region, respectively (see Table 1)*"

To:

"*DSCDs were provided for three chemical species, namely $O_4$, $NO_2$ and HCHO. $O_4$ and $NO_2$ were each provided for two different spectral fitting ranges, in the ultra-violet (UV) and the visible (Vis) spectral region, resulting in five data products (see Table 1)*".

**P6 L24-25 Algorithmically the retrievals are minimizing a cost function as stated at the end of the sentence, this is what the "model parameters are optimized to obtain", "maximum agreement" is not strictly the same as "minimum difference" and should be substituted.**

We changed the text from: "*To retrieve a profile from the measured dSCDs, the model parameters are optimized to obtain maximum agreement between the simulated and measured dSCDs by minimising a pre-defined cost function.*"

To: "*To retrieve a profile from the measured dSCDs, the model parameters are optimized to minimise the difference between the simulated and measured dSCDs based on a pre-defined cost function.*"

**P7 L2 The solutions obtained for the underconstrained problem are not unambiguous. In the case of OEM they are a maximum likelihood estimator predicated on the *a priori* information. Even if *a priori* information is perfect the obtained solution is not unambiguous simply the most likely. The authors should use a different word.**

We changed the wording, see our answer on the comment below.

**P7 L2-7 *a priori* information is more extensive than the *a priori* profile proper, it also includes the covariance matrix for OEM. This does more than "fill" the lack of information it also defines a portion of the cost function and forms the basis by which likelihood is assessed. This is critical background to understanding the path-dependent results the authors find and should be expanded upon.**

The corresponding paragraph was revised, also considering the comments by reviewer #1. It now reads:

"*Regarding profiles, typically only two to four degrees of freedom for signal (DOFS or p) can be retrieved from MAX-DOAS observations, such that general profile retrieval problems with more than p independent retrieved parameters are ill-posed and prior information has to be assimilated to achieve convergence. For OEM algorithms, this is provided in the form of an a priori profile and associated a priori covariance (Rodgers, 2000), defining the most likely profile and constraining the space of possible solutions according to prior experience. They constitute a portion of the OEM cost function such that with decreasing information contained in the measurements, layer concentrations are drawn towards their a priori values.*"

Also we extended some formulations throughout the manuscript, e.g. P13L29: "*At higher altitudes, OEM retrieval results are drawn towards the a priori profile (according to the definition of the cost-function, see Rodgers [2000])*"

For the very details of OEM the reader is encouraged to refer to the corresponding literature.

**P7 L33 the aerosol profiles are "extrapolated" not "interpolated"**

Done.

**P8 L8-9 The definition of the *a priori* covariance as defined here is a predicate to the later findings and should be discussed as such in relevant locations.**

Corresponding passages were revised. The importance of the choice of the a priori covariance is emphasized at relevant locations and the definition in P8 L8-9 is referenced.

**P11 L18-20 If I understand correctly, this method of processing gives a large weight to the uppermost one or two measurements available as these measurements define a majority of the relevant layer. Can the authors comment or elaborate?**

We agree with the reviewer. To make this point clearer we added a sentence very similar to the reviewer's comment: "*Note, that this approach gives a large weight to the uppermost measurements, as they are representative for the majority of the relevant layer.*"

**P12 L8 temperature and pressure should be spelled out here.**

Done.

**P12 L9 Wagner et al., (2019) find effects of up to 7% on the modeled O4 profile when using a standard atmosphere. This could be a significant contributor or the retrieved RMSD, can the authors comment?**

This is an aspect that we omitted so far. We did further investigation on this, with the results being summarised in the Supplementary material as follows:

**S7 Impact of the choice of pressure and temperature profiles for the RTMs**

Pressure ($p$) and temperature ($T$) profiles used for the RTMs within this study are averaged sonde measurements performed in De Bilt by KNMI during September months of the years 2013-2015 (see main text Sect. 2.1.3). To estimate the effect of this approximation on the results, IUPHD/ HEIPRO retrieved an additional set of profiles, using $p$ and $T$ information from radiosondes launched at KNMI (De Bilt) during the campaign. Between one and three sondes were launched every day except on 16 September. For each profile inversion, the temporally closest sonde observation was used. Table S7 shows the difference in RMSD and Bias magnitude between these results and the "standard" results of IUPHD/ HEIPRO (that used the prescribed averaged $p$ and $T$ profiles from years before) relative to the average RMSDs and average Bias magnitude for all participants.

The impact on the dSCD comparison is less than 5% for both, RMSDs and Bias magnitudes. For AOTs, VCDs and surface concentrations, significant improvement ($> 10\,\%$ in RMSD) is only observed for HCHO surface concentrations (17%) that contrasts with a deterioration for UV AOTs by 13%. The average improvement in RMSD for AOTs, VCDs and surface concentrations is 3.2%. The overall consistency between MAX-DOAS and supporting observations can thus be considered to remain similar, despite larger changes in some Bias magnitudes are observed (up to $51\,\%$ improvement for NO$_2$ Vis surface concentrations and up to $20\,\%$ deterioration for UV AOTs).

**Table S7.** The differences in RMSDs and Bias magnitudes for the IUPHD/ HEIPRO results arising from using daily $p$ and $T$ profiles, relative to the average RMSDs and Bias magnitudes assessed within the main study. Values are given for the comparisons of modelled and measured dSCDs ("dSCDs") and the comparisons against the supporting observations of AOTs, VCDs and surface concentrations as described in the main text. Minus signs indicate improvement. Only clear sky conditions were considered.

|  | dSCDs | | AOT/VCD | | Surface | |
| --- | --- | --- | --- | --- | --- | --- |
|  | ΔRMSD [%] | ΔBias [%] | ΔRMSD [%] | ΔBias [%] | ΔRMSD [%] | ΔBias [%] |
| HCHO | 2.7 | 3.5 | 6.8 | 10.5 | -17.4 | -22.0 |
| NO$_2$ UV | -0.7 | -1.1 | -2.7 | -2.6 | -3.5 | 8.7 |
| NO$_2$ Vis | -0.7 | -3.3 | -0.8 | -1.0 | -2.8 | -50.9 |
| Aerosol UV | -0.7 | 0.7 | 12.5 | 20.2 | - | - |
| Aerosol Vis | -0.2 | 2.1 | -8.7 | -40.1 | - | - |

These findings are also briefly discussed in the conclusions now.

**P12 L20-25 Is the least-squares regression a minimization of vertical distance or orthogonal distance?**

The vertical distance is minimised. This information was added during the course of the revision of Sect. 2.3.1.: *"For the linear regression analysis, the vertical distance between the model and the data points is minimised […]"*

**P12 Eq7 1/Np here should be in parentheses for clarity**

Instead of adding parentheses we changed the formatting to achieve a similar effect.

$$\sigma_{arms,p} = 1/N_P \sum_p \sigma_{rms,p}$$

We changed:

$$\sigma_{arms,p} = \frac{1}{N_P} \cdot \sum_p \sigma_{rms,p}$$

To:

**P14 L24 replace "not given" with "inaccurate"**

Done.

**P15 L1-2 "Aij describes the sensitivity of the measured concentration in the ith layer to small changes in the real concentration in the jth layer.,"**

Done.

**P15 Eq11 The coefficient of 12 in this equation seems to be the result of summing over the lowest 12 layers, corresponding to 2.5 km. However, this is not stated.**

The spread is calculated considering the cross sensitivity to each layer. The coefficient of 12 is a normalisation factor which is part of the original definition of the "spread" (see Rodgers, 2000, as cited in connection with Eq. 11 in the manuscript). Initially we thought it might be helpful to find some simple measure for the retrieval's spatial resolution and show it in the plots. However, as the spread does not provide any substantially new information to the reader and might rather be misleading than helpful (see also the reviewer 's comment on Fig. 2 below) we decided to completely remove it from the text and the plots.

**P15 L16-18 The increase in information content reflects an increase in the differential light path specifically. While this follows from the longer light paths overall, it is the increased differential path which is the source of the information.**

We replaced "*light path*" by "*differential light path*"

**P16 Fig 2. The symmetric boxes illustrating are misleading. As the AVK traces demonstrate, the information content moves as well as being "smoothed". The boxes should be centered in a more rational way or else eliminated.**

As explained above (comment on P15, Eq11), the boxes in the plots and corresponding paragraphs on the "spread" in the main text were eliminated.

**P17 Table 2 Most groups are listed by city, however, Anhui is listed by province, should this not be Hefei?**

We changed this to "Hefei". Further similar issues in the same table were also fixed:

"Department of Physics, University of Toronto, Toronto, Canada" → "Department of Physics, University of Toronto, Canada"

"NASA-Goddard, Greenbelt, Maryland" → "NASA-Goddard, Greenbelt, United States"

**Figs. 3-7. The red triangles are not readily seen against the color scale.**

We changed the colour of the triangles to pink, which is not ideal either but was the colour we consider best distinguishable from the colour scale in the background:

Submitted version of the mansucript:

[Figure]

Now:

[Figure]

**Figs. 6-7 In the bottom row when only surface measurement are available these are almost imperceptible.**

We agree. However, we do not see how to change this without introducing potentially confusing features. Please note, that the figures the reviewer refers to are meant to provide an at best complete overview of the available datasets for qualitative comparison and that data of this extent and inhomogeneity are challenging to visualise. Further note that the same data appears again in the following sections in more detailed plots which are easier to read. This is why we finally decided to leave them as they are.

**P24 L6 what precisely do the authors mean by "update interval of the jacobians"?**

In optimal estimation algorithms (where the model parameters are iteratively adapted), one of the computationally most expensive steps is to derive the jacobians of the simulated dSCDs w.r.t. the model parameters. Typically, inversion problems of the kind discussed in the manuscript are moderately linear and do not require a recalculation of these jacobians in each iteration to achieve convergence. This is used by some algorithms to save computing time. The impact of this "shortcut" on the final results depends on the atmospheric scenario, on the exact implementation and the settings defined by the user.

We replaced the text in brackets *"(e.g. number of iteration in the inversion, accuracy criteria for the RTMs, update interval of the jacobians, ...)"* by:

*"The latter are for instance the accuracy criteria for the RTMs, the number of iterations in the inversion, the convergence criteria or the decision at which points of the iteration process the forward model jacobians are (re-)calculated."*

**P24 L6-7 Are the larger discrepancies not simply a reflection of the greater DOFS?**

This is well possible and also stated in just the following sentence: *"In the case of OEM algorithms, a reason might be that there is lower information content in the UV, meaning that the retrievals are drawn closer to the collectively used a priori profile"*.

**P24 11-13 In this section while using the same set of dSCDs how can the authors speak to horizontal inhomogeneity? How would such an inhomogeneity be detected?**

The idea was, that inhomogeneity leads to less stable solutions, making the algorithms more sensitive to differences in the inversion settings. But this might indeed be too far fetched to be mentioned here. We therefore removed the sentence: *"Horizontal inhomogeneities are an unlikely reason because the worse performance in the Vis was also apparent in the study by Frieß et al. (2019) with synthetic data, where horizontal gradients were non-existent."*

**P24 L28 Can the authors clarify what they mean by "technical problems" do they think there was some error in the implementation of the protocol?**

Yes this could have been the case. Or that improper/different retrieval settings were applied as it was the case for Heipro, where discrepancies between IUPHD and UTOR could be explained by different numbers of applied iteration steps. The paragraph was rearranged and revised. Amongst others we removed the statement with the "technical problems" and now "suspect similar reasons" as for the IUPHD <-> UTOR discrepancies.

Before:

*"An example for large discrepancies between participants using the same algorithm is AUTH aerosol in the UV, where in contrast to other bePRO users oscillations seem to appear. We suspect this to originate from technical problems which could not yet been identified. The discrepancies between IUPHD and UTOR (both using HEIPRO) were found to mainly be caused by differences in the number of applied iteration steps in the Levenberg-Marquardt optimization scheme during aerosol retrieval. IUPHD (UTOR) applied 20 (5) iterations. The consequences are evident throughout the comparison."*

Now:

*"An example are the discrepancies between UTOR/ HEIPRO and IUPHD/ HEIPRO. In this case the number of applied iteration steps in the aerosol inversion was identified as the main reason: UTOR and IUPHD used 20 and 5 iterations here, respectively. The consequences are evident throughout the comparison. Another example is the aerosol UV retrieval of AUTH/ bePro, where in contrast to other bePRO users oscillations seem to appear. We suspect this to originate from similar reasons, which could not yet been identified."*

**Figs. 8-12 If there are uncertainties in these graphs as indicated by the legend for Fig 8, they cannot be seen.**

We agree. We reduced the edge width of the markers to improve this. Still they are only visible when looking very closely at data points lying apart from the main point cloud. Anyway we decided to keep them as they at least give an impression of the uncertainties' order of magnitude.

**P28 L3 As stated above, per the results presented signals aloft can be reliably detected, but not reliably located and/or quantified. Language should be edited to reflect this.**

We changed: *"[…] cannot be reliably detected […]"*

To: *"[…] cannot be reliably located and quantified […]"*

Similar statements were adapted throughout the manuscript.

**P28 L13-15 On first reading the finding that adjusting MAX-DOAS AOT by the ratio to the sun photometer improves the agreement seems obvious, even tautological. The actual processing as described in the supplement needs to be better reflected in the main text.**

We agree that it is strange to emphasize the PAC all over the manuscript to finally show the results in the Supplement. Therefore, we embedded Supplement S2 into the main text Section 3.4.

**P29 L3-4 The authors state "even though the physical reason for PAC and SF are different." This is surprising as it suggests that the authors posit a specific physical reason for SF which is not that for PAC, what is this?**

We agree with the reviewer corrected this statement regarding the "physical reason", as it is not well-founded. We replace the sentence by:

"[…] even though the motivation for the application of the PAC and the SF are different."

The motivations are in fact very different: the application of the PAC is necessary solely for mathematical reasons related to the concept of optimal estimation and prior constraints applied therein. In contrast, the prominent publications motivating/discussing the application of an O4 scaling factor (Wagner (2009), Clémer (2010), Ortega (2016) and Wagner (2019)) forward modelled O4 dSCDs (using an atmosphere derived from supporting observations like Lidars) to measured O4 dSCDs. They do not make use of optimal estimation or a priori profiles similar to those used in our study. Thus their findings are independent from any kind of PAC.

We added a corresponding explanation to the same paragraph:

*"[…] even though the motivation for the application of the PAC and the SF are different: the application of the PAC is necessary solely for mathematical reasons related to the concept of OEM and prior constraints applied therein. In contrast, publications that suggest or discuss the application of an SF (Wagner, 2009; Clémer, 2010; Ortega, 2016; Wagner, 2019) directly compare forward modelled $O_4$ dSCDs (using an atmosphere derived from supporting observations to reproduce the real conditions to best knowledge) to measured $O_4$ dSCDs. They do not make use of optimal estimation or prior constraints similar to those used in our study. Thus their findings can be considered independent from any kind of PAC."*

And to the paragraph above:

*"It shall be pointed out that for OEM algorithms the necessity for the PAC can generally be reduced by using improved a priori profiles and covariances (e.g. from climatologies, supporting observations and/ or model data). Also the values for $f_\tau$ will differ, when other a priori profiles and covariances than the ones prescribed for this study (see Sect. 2.1.3) are used."*

**Fig. 13 and other Figs following same format. In the top row, why are the scatters plotted on an inverted axis? Cannot the scatter exceed one? Even quite significantly? Here and elsewhere the hashed and solid shading are not readily distinguishable.**

We agree, that this was not a good solution. We inverted the axis back to the normal direction. Further we adapted the figure to make a distinction between hashed and solid areas unnecessary.

Example of the updated plot:

[Figure]

**Fig. 14 and other Figs following same format. While I can appreciate what the authors are trying to communicate with the pie chart symbols, the clear and cloudy data are drawn from the same total and the symbols repeat within a given column. This should be simplified in some way.**

We thank the reviewer for this suggestion, which makes the figures much easier and more comfortable to read. We discarded the pie chart symbol and added another thin row of plots indicating the number of used profiles for each of the columns.

Submitted:

[Figure]

Now:

[Figure]

**P31 L9-12 This paragraph in particular demonstrates that aerosol aloft are detectable.**

We partly agree. The detection of aerosol aloft is at least limited. However, as stated above, we revised text passages stating that aerosol aloft are undetectable.

**P31 14 The first sentence should be reworded, the VCDs are compared to different standards or "assessed", but the NO2 VCDs are not compared to the HCHO VCDs**

We changed: *"This section compares the VCDs of HCHO and NO₂."*

To: *"This section assesses the consistency of the VCDs for each of the trace gases HCHO and NO₂".*

**Fig. 15 where is the outlier referred to on P31 L21?**

By "outlier" we refer to a radiosonde profile here, these are not shown in Fig. 15. In the case of this "outlying" profile, the NO₂ concentrations were close to the radiosonde detection limit and instrumental offsets made it unsuitable for the corresponding study, which was to show whether a correction similar to the PAC might be necessary also for NO₂ VCDs. However, "outlier" is probably not the right word to use here.

To make things clearer we changed the text: "*Ignoring an outlier on 09-27 07:00:00, where NO₂ concentration was close to the radiosonde detection limit, […]*"

To: "*Ignoring one problematic radiosonde profile on 09-27 07:00:00 (where NO₂ concentration was close to the radiosonde detection limit and thus instrumental offsets became particularly apparent), […]*"

**P33 L13-14 the LP-DOAS data are described as "very accurate, representative, and complete" while these are likely well supported assessments, such strong statements should be demonstrated or else backed up by a citation.**

This statement is already justified in Section 2.2.5, where the LP-DOAS setup at CINDI-2 is introduced. We added a cross reference to this section.

"*Very accurate*" is supported by multiple references there: Pöhler et al., 2010; Merten et al., 2011; Nasse et al., 2019. We added Pikelnaya et al., 2007 to further support this statement.

"*Representative*", since its light path covers the lowest MAX-DOAS retrieval layer fully and exclusively.

"*Complete*" since it provides a near-continuous dataset over the campaign period.

**Fig 19. Sondes are not listed in the legend. Here and elsewhere the color of the lidar and sondes is very challenging to distinguish.**

We like to thank the reviewer for pointing out this omission, we added radiosondes to the legend. Further, we brightened the orange color and darkened the red color which are used to visualize $NO_2$-Lidar and radiosonde data throughout the paper.

**P34 L3 The language here should be more precise. The surface concentration does reflect the ability of MAX-DOAS retrieval to isolate the surface layer specifically. However, the isolation and resolution of the surface layer does not imply in and of itself the resolution of the vertical profile above it.**

We agree that this could be misleading.

We changed: *"[…] the surface concentration comparison also reflects the MAX-DOAS' ability to actually resolve vertical profiles, as it requires an isolation of the surface layer from the layers above."*

To: *"[…] the surface concentration comparison requires an isolation of the surface layer from the layers above and therefore reflects the MAX-DOAS' ability to actually resolve vertical profiles at least close to the surface."*

**P35 L5-7 How the consistency of the surface concentrations point to a problem in the direct sun data? Is it not equally possible that the MAX-DOAS VCD apart from the lowermost layer are flawed?**

Yes, we agree with the reviewer. We changed the text: *"The good agreement of the surface concentrations with the supporting observations during the first days is opposite to the VCD comparison, which at least for $NO_2$ points to a problem with the direct-sun data."*

To: *"The good agreement of the surface concentrations with the supporting observations during the first days is opposite to the VCD comparison, which at least for $NO_2$ points to a problem with the retrieval results in higher layers or the direct-sun data"*

**P35 L10-11 I believe this final sentence refers to the comparisons in Tables S4 and S5, however, that is not clear in the text.**

The sentence refers to Fig.18 (HCHO time series) and Fig. 19 ($NO_2$ time series), where in the top row the scatter among the participants and in the two lower rows the specified uncertainties of the MAX-DOAS observations are indicated by the faint areas.

To clarify this point, we changed the text: "*Again the scatter to the MAX-DOAS median even for clear-sky conditions are similar or larger than the specified errors (factors of about 1, 2 and 3 for HCHO, $NO_2$ UV, $NO_2$ Vis, respectively).*"

To: *"Again, as for AOTs and VCDs, the scatter among the participants is similar or larger than the specified errors even for clear-sky conditions (factors of about one for HCHO, two for $NO_2$ UV and three for NO2 Vis, see Fig. 19 and Fig. 20)"*

Further, we added a sentence to the caption of Fig. 19: "*Note, that the mean specified uncertainties in the two lower rows of the figure are very small and thus barely visible.*"

**P36 L1-4 Can this thinking be made more quantitative by reference to the fτ for the Vis and UV products?**

This point became obsolete, since the whole section was removed as suggested by reviewer 1.

**In the supplement:**

**P2 L18 the shift to lower altitudes is a simple reflection of the construction of the covariance. This is hinted at on L21, but should be spelt out. As constructed the retrieval does not have uncertainty into which to place the information at higher altitudes, but the information is present in the measurements and is placed at an altitude which is accessible within the constraints of the prescribed covariance.**

This comment explains the issue accurately and concise. We adopted the reviewer's wording:

We changed: "However, a part of the high-altitude aerosol appears to be shifted to lower altitudes here by the retrieval."

To: "However, corresponding information actually seems to be present in the measurements, since part of the high-altitude aerosol appears to be shifted to lower altitudes which are accessible within the constraints of the a priori covariance."

**P4 L12-14 Clear-sky O4 dSCD are not the largest possible, if there is small but non-zero aerosol scattering concentrated at altitudes below the median altitude of photon scattering for a relevant geometry this leads to brightening. Hence why aerosol can appear as increased albedo for satellites.**

The reviewer is correct here, our statement is wrong. Note, however, that the sentence refers to low aerosol clear sky scenarios, where this assumption is nearly fulfilled.

We therefore changed the text: "*Finally, Wagner et al. (2009) reported, that under low aerosol conditions, measured dSCDs sometimes even exceed dSCDs modelled within an aerosol free atmosphere, where O4 dSCDs are expected to be the largest possible (regarding clear-sky scenarios only).*"

To: "Also, *Wagner et al. (2009) reported that, under low aerosol conditions, measured dSCDs sometimes even significantly exceed dSCDs modelled within an aerosol free atmosphere, where O4 dSCDs are close to the largest possible (regarding clear-sky scenarios only).*"

**Fig S11 The color scheme makes this figure very difficult to read.**

This problem was solved by changing the colors for radiosondes and NO2-Lidar throughout the paper.

**Fig S12 The distance scale in this figure seems somewhat misleading in light of Fig. S13. The provided exponential curves appear to imply a radical difference in ranging between the Vis and UV, whereas Fig. S13 makes clear that changes in atmospheric conditions are responsible for most of the difference.**

We changed the figure by showing the average, minimum and maximum sensitivity range for UV and Vis, respectively:

Submitted version of the manuscript:

[Figure]

Now:

[Figure]

**Fig. S34 If I understand this figure correctly virtually all data are within two standard deviations, is this not as expected. P33 L6-7 seems to imply something unexpected.**

The word "indeed" is misleading here. Further, a short conclusion on the actual meaning of this study is missing.

We changed the text: "*Figure S34 shows histograms of the calculated differences. An estimate of the impact of smoothing on the retrieval results is actually provided by the OEM retrievals themselves as the "smoothing error". The specified smoothing errors are also indicated in Fig. S34 and indeed slightly larger than the standard deviation observed in in this test.*"

To: "*Figure S34 shows histograms of the calculated differences. The standard deviation is about $5x10^9$ molec. $cm^{-3}$ which is only about 10 % of the total average RMSD between MAX-DOAS and LP-DOAS observations. An estimate of the impact of smoothing on the retrieval results is actually provided by the OEM retrievals themselves as the "smoothing error". The specified smoothing errors are also indicated in Fig. S34 and are similar to the standard deviation observed in in this test, meaning that for the surface layer they are well representative for the real impact of smoothing.*"

[revised manuscript text omitted]
  average ( solid), the minimum (dashed) and maximum (dashed) viewing distances encountered during the campaign  in the UV (blue) and Vis (green).

[Figure]

**Figure S12.** Viewing distance (HSR) of MAX-DOAS instruments during CINDI-2. It was calculated for different  elevation angles (1, 2, 3, 4, 5, 6 and 8° with increasing transparency of the curves) and the average value for UV and Vis (thick lines).

**S6**

**S6 Spatio-temporal mismatch and variability**

Given the MAX-DOAS horizontal sensitivity ranges determined in Sect. S5, approximate values for the spatio-temporal
5  mismatch of MAX-DOAS and different supporting observations can be derived. They are given in Table S6. The potential impact of these mismatches can be demonstrated by means of the $NO_2$ surface concentration. The left panel of Fig. S13 shows observations of the $NO_2$ surface concentrations at their original temporal resolution $\Delta t$ and integration time $t_{int}$ . The CE-DOAS as a point measurement with $\Delta t = t_{int} = 1\,min$ shows very strong variability on short timescales. However, for the tower measurements (all in situ instru-
10 ments in the tower vertically integrated as described in main text Sect. 2.2.5 at $\Delta t = 20\,min$ and $t_{int} \approx 5\,min$), the LP-DOAS

**Table S6.** Estimates for the average spatio-temporal mismatch of different supporting observations w.r.t. to the MAX-DOAS measurements. For the location of the MAX-DOAS observations the centers of mass of the horizontal sensitivity curves from Sect. S5 were used. For the location of sun photometer and direct-sun DOAS observations, the center of the lines of sight towards the sun up to 2 km atitude were considered.

| Observation | Spatial mismatch [km] | Temporal mismatch [min] |
|---|---|---|
| Sun photometer | 13 | 8 |
| Ceilometer | 11 | 0 |
| Direct-sun DOAS | 13 | 23 |
| $NO_2$-Lidar | 10 | 9 |
| Radiosonde | 6 | 13 |
| LP-DOAS | 10 | 6 |
| In-situ in tower | 11 | 0 |

($\Delta t = 32 \, \text{min}$, $t_{int} \approx 100 \, \text{s}$) there is already significant smoothing. The 1D-MAX-DOAS data was recorded by DLR (see Supplement S10), who retrieved profiles in the nominal azimuth direction ($287°$) more or less continuously ($\Delta t = 15 \, \text{min}$, $t_{int} \approx 10 \, \text{min}$). In all measurements there is significant variation on the sub-hour timescale. Further, spatial variability might be observed in the form of disagreement between UV and Vis observations of the 1D-MAX-DOAS as viewing distance and

5   thus the sampled air volume changes between the two spectral ranges (see Supplement S5). To estimate the order of magnitude, the right panel of Fig. S13 shows a kind of autocorrelation of the total campaign time series of each observation. The RMSD between the original and a temporally shifted signal is calculated.  1D MAX-DOAS Vis data is not  shown, as multiple

10   gaps in the data complicated the autocorrelation. Comparing this figure with values from Table S6 yields that spatio-temporal variability causes RMSD values of around $3.5 \times 10^{10} \, \text{molec cm}^{-3}$  in the $NO_2$ surface concentration, which is indeed of the order of the observed RMSD values in the $NO_2$ surface concentration comparisons within this study (approx. $5 \times 10^{10} \, \text{molec cm}^{-3}$, compare to main text Fig. 22).

For another demonstration of the spatial variability, we refer to data from the IMPACT instrument (Peters et al., 2019),

15   an imaging MAX-DOAS operated by IUP-Bremen (IUPB) which allows to perform elevation "scans" in different azimuth viewing directions in quick succession. During CINDI-2, the IMPACT performed full-azimuthal scans in $10°$ steps every 15 minutes. Figure S14 exemplarily shows the observed $NO_2$ Vis dSCDs at $4°$ elevation on the 20 September 2016 together with dSCDs measured by the IUPB standard MAX-DOAS instrument in the nominal azimuth direction ($287°$, compare main text Sect. 2.1). The red shaded area depicts the variation of the dSCD with azimuth viewing direction. In particular around local

20   noon this variation is tremendous, exceeding a factor of five. Further investigation on this issue can be found in Peters et al. (2019).

[Figure]

**Figure S13.** Left: Different observations of the $NO_2$ surface concentrations on 14 September 2016, each at its original temporal resolution to reveal short-term variations. Coloured areas behind the lines indicate the specified uncertainties. Right: RMSD values obtained from a kind of autocorrelation analysis over the whole campaign (night times excluded). For each observation, the RMSD between the original and a temporally shifted signal is calculated. The temporal shift (bottom horizontal axis) was varied between 0 and 4 hours. The temporal shift was roughly converted to its spatial equivalent by multiplication with the average observed wind speed in the surface layer ($\approx 5\,\mathrm{m/s}$), yielding the top horizontal axis.

[Figure]

**Figure S14.** Variation in the $NO_2$ Vis dSCDs with different azimuth viewing directions at $4°$ elevation, as observed by the IMPACT imaging MAX-DOAS (Peters et al., 2019). Around local noon this variation is largest, exceeding a factor of 5. The time is UTC.

**S7   Impact of the choice of pressure and temperature profiles for the RTMs**

Pressure ($p$) and temperature ($T$) profiles used for the RTMs within this study are averaged sonde measurements performed in De Bilt by KNMI during September months of the years 2013-2015 (see main text Sect. 2.1.3). To estimate the effect of this approximation on the results, IUPHD/ HEIPRO retrieved an additional set of profiles, using $p$ and $T$ information from radiosondes launched at KNMI (De Bilt) during the campaign. Between one and three sondes were launched every day except on 16 September. For each profile inversion, the temporally closest sonde observation was used. Table S7 shows the difference in RMSD and Bias magnitude between these results and the "standard" results of IUPHD/ HEIPRO (that used the prescribed averaged $p$ and $T$ profiles from years before) relative to the average RMSDs and average Bias magnitude for all participants.

The impact on the dSCD comparison is less than 5% for both, RMSDs and Bias magnitudes. For AOTs, VCDs and surface concentrations, significant improvement ($> 10\%$ in RMSD) is only observed for HCHO surface concentrations (17%) that contrasts with a deterioration for UV AOTs by 13%. The average improvement in RMSD for AOTs, VCDs and surface concentrations is 3.2%. The overall consistency between MAX-DOAS and supporting observations can thus be considered to remain similar, despite larger changes in some Bias magnitudes are observed (up to 51% improvement for $NO_2$ Vis surface concentrations and up to 20% deterioration for UV AOTs).

**Table S7.** The differences in RMSDs and Bias magnitudes for the IUPHD/ HEIPRO results arising from using daily $p$ and $T$ profiles, relative to the average RMSDs and Bias magnitudes assessed within the main study. Values are given for the comparisons of modelled and measured dSCDs ("dSCDs") and the comparisons against the supporting observations of AOTs, VCDs and surface concentrations as described in the main text. Minus signs indicate improvement. Only clear sky conditions were considered.

| | dSCDs | | AOT/VCD | | Surface | |
|---|---|---|---|---|---|---|
| | $\Delta$RMSD [%] | $\Delta$Bias [%] | $\Delta$RMSD [%] | $\Delta$Bias [%] | $\Delta$RMSD [%] | $\Delta$Bias [%] |
| HCHO | 2.7 | 3.5 | 6.8 | 10.5 | -17.4 | -22.0 |
| $NO_2$ UV | -0.7 | -1.1 | -2.7 | -2.6 | -3.5 | 8.7 |
| $NO_2$ Vis | -0.7 | -3.3 | -0.8 | -1.0 | -2.8 | -50.9 |
| Aerosol UV | -0.7 | 0.7 | 12.5 | 20.2 | - | - |
| Aerosol Vis | -0.2 | 2.1 | -8.7 | -40.1 | - | - |

**S8 Further details on the comparison results**

**S8.1 AVKs of individual participants**

Figures S35 to S39 show the averaging kernels (AVKs) and retrieved degrees of freedom of signal (DOFS) of each participant for aerosol UV. For explanation of colours and symbols please refer to main text Sect. 3.1. The DOFS values in brackets were calculated considering valid data only.

[Figure]

**Figure S15.** Mean averaging kernels for Aerosol UV for each participant. Coloured values at AVK peaks show the amount of retrieved information on the respective layer in percent. "DOFS" numbers are given for clear-sky (green) and cloudy (red) conditions. Values in brackets are DOFS including flagging.

[Figure]

**Figure S16.** Mean averaging kernels for Aerosol Vis for each participant. Description of Fig. S35 applies.

[Figure]

**Figure S17.** Mean averaging kernels for HCHO for each participant. Description of Fig. S35 applies.

[Figure]

**Figure S18.** Mean averaging kernels for $NO_2$ UV for each participant. Description of Fig. S35 applies.

[Figure]

**Figure S19.** Mean averaging kernels for NO₂ Vis for each participant. Description of Fig. S35 applies.

**S8.2  Profile deviation statistics**

Figures S20 to S24 show statistics on the observed differences in the retrieved profiles for all five species. The plots on the left compare the retrieved profiles of individual participants $x$ to the median MAX-DOAS profiles $\bar{x}$. While the vertical axes represent altitude, the horizontal axes depicts the difference $x - \bar{x}$. The coloured boxes indicate the $25\% - 75\%$ percentile, whiskers are $5\% - 95\%$. Black dots indicate the mean value. For each layer there are boxplots for clear-sky (green) and cloudy conditions (red). Note that for aerosol there are two different horizontal axes defined for the two cloud conditions: the green scale at the bottom and the red scale on the top of each plot. Only valid data (flagged) was considered. For aerosol and NO$_2$ a plot on the very right shows statistics of the difference of supporting measurements $x_{anc}$ (lidar/ radiosonde for NO$_2$, sun photometer scaled ceilometer for aerosol) to the median $\bar{x}$, hence $x_{anc} - \bar{x}$. The numbers in all the plots show RMSD deviation of the three lowest (most sensitive) layers. Dashed lines indicate the median retrieval uncertainty as specified by the participants.

[Figure]

**Figure S20.** Left:  Deviations of Aerosol UV profiles (valid only) of individual participants from the MAX-DOAS median profiles. Dots show the mean, boxes indicate the (25%-75%) percentile, and whiskers show (25%-75%) percentile. Green (red) box-whiskers represent clear (cloudy) conditions. Note, that there are different x-scales (on top and bottom of the plot) for different cloud conditions. The average standard deviations specified by the participants are indicated by the dashed lines. Right: Deviation of the AOT scaled ceilometer backscatter signal to the MAX-DOAS median profiles. The numbers in the plots indicate RMSD values for clear sky (green) and cloudy (red) conditions. Further details are given in the related text in Sect. S8.2

[Figure]

**Figure S21.** Flagged deviations for Deviations of Aerosol Vis profiles (valid only) of individual participants from the MAX-DOAS median profiles. Description of Fig. S20 applies.

[Figure]

**Figure S22.** Flagged deviations for Deviations of HCHO profiles (valid only) of individual participants from the MAX-DOAS median profiles. The description of Fig. S20 applies but for HCHO, there is no independent reference profile available.

[Figure]

**Figure S23.**  Deviations of NO₂ UV profiles (valid only) of individual participants from the MAX-DOAS median profiles. The description of Fig. S20 applies. On the right, deviation of the median retrieved profiles from the few available NO₂ lidar and sonde profiles are shown.

[Figure]

**Figure S24.**  Deviations of NO₂  Vis profiles (valid only) of individual participants from the MAX-DOAS median profiles. The description of Fig. S23 applies.

**S8.3    Correlation plots for AOTs, VCDs and surface concentrations**

[revised manuscript text omitted]

**S10.2 Overview plots**

This section is equivalent to Sect. 3.2 in the main text.

[Figure]

**Figure S40.** Aerosol UV extinction profiles retrieved from the participant's own dSCDs. The lowest row shows AOT scaled ceilometer backscatter profiles, calculated as described in Sect. S4.1. Backscatter profiles, which were scaled from MAX-DOAS AOTs (and which are therefore not fully independent) are marked by red-pink triangles. Maximum extinction values reach $20\,\mathrm{km}^{-1}$, exceeding the colour scale.

[Figure]

**Figure S41.** Aerosol Vis extinction profiles retrieved from the participant's own dSCDs. The lowest row shows AOT scaled ceilometer backscatter profiles, calculated as described in Sect. S4.1. Backscatter profiles, which were scaled from MAX-DOAS AOTs (and which are therefore not fully independent) are marked by red-pink triangles. Maximum extinction values reach 20 km$^{-1}$, exceeding the colour scale.

[Figure]

**Figure S42.** HCHO concentration profiles retrieved from the participant's own dSCDs. The "Surf"-row shows LP-DOAS surface concentrations.

[Figure]

**Figure S43.** NO$_2$ UV concentration profiles retrieved from the participant's own dSCDs. The lowest row shows a combined dataset of NO$_2$ lidar, radiosonde, LP-DOAS and tower in-situ data. Redundant surface concentration measurements were averaged.

[Figure]

**Figure S44.** NO$_2$ Vis concentration profiles retrieved from the participant's own dSCDs. The lowest row shows a combined dataset of NO$_2$ lidar, radiosonde, LP-DOAS and tower in-situ data. Redundant surface concentration measurements were averaged.

**S10.3    Modelled and measured dSCDs**

This section is equivalent to Sect. S10.3 in the main text.

[Figure]

**Figure S45.** $O_4$ UV dSCD correlation when profiles are retrieved from the participant's own dSCDs. Marker colours and marker shapes indicate the cloud conditions and viewing elevation angles, respectively. Numbers represent the measurement error weighted RMSD between measured and modelled dSCDs for clear sky (green) and cloudy (red) conditions. Values in brackets were calculated only considering valid data.

[Figure]

**Figure S46.** O$_4$ Vis dSCD correlation. Legends of Fig. S45 apply.

[Figure]

**Figure S47.** HCHO dSCD correlation. Legends of Fig. S45 apply.

[Figure]

**Figure S48.** NO$_2$ UV dSCD correlation. Legends of Fig. S45 apply.

[revised manuscript text omitted]

---

## Author Response (AR2)

**General**

We thank the reviewer for the constructive comments on our revised manuscript. Based on these comments we applied minor changes, as described below.

Further changes applied in the course of the revision:

1. We corrected the $NO_2$ spectral fitting window in Table 1 from "336.5 – 359" to "338 – 370".
2. We added another reference (Chan, 2020) for the $M^3$ algorithm in Table 2.

No changes have been applied to the supplementary material.

Notation: **Reviewer's comments are printed in bold**, author responses in regular font. *Explicit changes in the manuscript are indicated by italic font.*

**Responses**

**Reviewer comment: I have read the responses from my initial comments and check the revised manuscript. I appreciate that the author(s) reply to all my comments. With that being said, I still have minor (maybe major) comments below:**

**- It is very concerning that differences of individual participants vary by a factor of 10 for VCDs, AOT and surface concentration, considering that identical dSCDs are used and key retrieval parameters are prescribed. For the comparison against supporting/independent observations the differences are even larger. Note that if participants used their own dSCDs discrepancies degrade even more.**

Answer: First of all, we would like to point out that we are not talking about the range of the profiling results themselves, but of the range of their differences (RMSDs) here. We agree, that the range of RMSDs is large but it is not surprising. The following aspects need to be considered:

1. Former publications report similar (albeit smaller) variations (e.g. Frieß 2016, Frieß 2019).
2. In contrast to Frieß (2016), algorithms with different approaches and priorities participated. An Example is the NASA Realtime algorithm: with its simplified radiative transport assumptions, it achieves outstanding computational performance at the cost of accuracy.
3. In contrast to former studies, more groups of different levels of experience with MAX-DOAS profiling participated in our study.

Finally, all RMSDs should be regarded in relation to the typical mean values of AOT, VCD and surface concentrations observed over the campaign period (also given in the abstract). Mean AOTs and VCDs for instance are significantly larger than the largest corresponding RMSDs, and thus even the least accurate algorithms can be considered to provide useful information.

The abstract is already extensive, but the aspects above are discussed in the conclusions (P41, L3-L20). Here we added one more sentence in the discussion:

"*Note also that the compared algorithms have different priorities: the NASA/ Realtime algorithm for instance is optimised for computational performance rather than accuracy.*"

**Typically, MAX-DOAS (and other ground-based remote sensing instruments) are viewed as key measurements for validation of satellite products, however, after reading the revised manuscript it is not clear what to make of this work, again from a validation point of view.**

As pointed out in the answer to the previous reviewer's comment all data provide useful information, even considering those with a relatively large uncertainty. Therefore, we don't think that the comment of the reviewer is appropriate. We consider the major outcomes from a validation point of view to be:

1. The quality of MAX-DOAS retrieval results depends strongly on the choice of the algorithm and the careful choice of the settings.
2. The MAX-DOAS technique allows to retrieve AOTs, VCDs and surface concentrations with good accuracy (see the best RMSDs and Biases achieved), if algorithms, settings and quality filters are chosen carefully, ideally by more experienced users. E.g. the minimum achieved RMSDs of 0.02 to 0.03 (IUPB/BOREAS, BIRA/BePRO (flagged), INTA/BePRO (flagged), IUPHD/HEIPRO) against the sun photometer AOT approach the accuracy of sun photometers (estimated to 0.02 by Smirnov et al., 2000).

Both aspects are discussed in the conclusions already, but we found that it might be useful to point them out in a summarising paragraph at the very end of the paper. We added:

*"We summarize our major findings as follows: besides the quality of the spectral data, the applied inversion strategy has significant impact on the accuracy of MAX-DOAS retrieval results. Nevertheless, partial AOTs, VCDs and surface concentrations can be retrieved with good accuracy, if algorithm, settings and quality filters are chosen carefully and ideally by experienced users. For the future, we therefore suggest to put focus on further harmonisation of MAX-DOAS retrievals, in particular with regard to their application by the broader scientific community."*

**The work presented here is important and findings are key as well, but I highly suggest making the point of what is needed to improve these differences.**

As stated in our comments from May, 2020, we fully agree, that this would be of great value but is out of the scope for a (already quiet long) comparison study as presented here. This study is in first line meant to quantify the discrepancies among the algorithms/participants and to identify possible reasons, which might be further investigated. However, regarding the complexity and diversity of retrieval algorithms, the latter is rather a task for the future and not to be solved here.

**- At the end of the abstract and in the manuscript it is mentioned that for OE the underestimation of AOT compared to sun photometer is due to lack of sensitivity to higher altitudes and associated to smoothing effects and a priori assumptions. On my first review I recommended to use the ceilometer extinction profiles as a priori and compare retrievals with the exponentially decreasing profile to further improve this statement along with drawing better findings regarding the scaling factor (and PAC). However, authors decided not to perform and replied with the following two arguments:**

**1. The paper aims at the comparison and validation of MAX-DOAS profiles retrieved under typical measurement conditions. This includes using prior information as they are typically available for an arbitrary measurement location and season. Having daily radiosondes, ceilometer data and collocated sun photometer measurements at hand is not a very usual scenario. In fact, most MAX-DOAS studies have to resort to climatologies for their prior assumptions.**

**2. Since the MAX-DOAS results are validated by the supported observations (at least qualitatively, in the case of the ceilometer profiles), they need to be kept independent, which is not the case if one observation serves as a priori for the other.**

We apologize, there was a misunderstanding from our side: we argued against the application of ceilometer profiles as a priori for the whole comparison. A side study to investigate the impact is another story (see comments below).

**Regarding 1, what is the point of having all supporting measurements if they are not used to improve retrievals and reduce errors in the forward models?, especially because also the aim of the study is to further compare with colocated supporting observations.**

In our study they are primarily meant to be used for validation of MAX-DOAS profile retrievals as they are typically performed today (in the absence of supporting observations or other sources of accurate a priori knowledge). We think that other activities (e.g. improving the forward models or combining all CINDI-2 observations to further confine the real atmospheric state) should be performed in separate studies.

**Regarding 2, I am not sure if I follow this, my understanding is that ceilometer observations are not used quantitatively, why do you need to keep them independent?**
The ceilometer profiles are used qualitatively, e.g. in the overview Figures 3 and 4 and quantitatively in the course of the PAC. Further, quantitative comparisons of the profile shapes are included in the Supplement (Fig. S.20, S.21). For this reason and the reasons given above, they should be kept independent at least for the main comparison.

**Furthermore, I asked if by using a ceilometer as a priori the sensitivity increases at higher layers? Do AKs change?**

**Author reply: This depends on the a priori covariance. Since the uncertainty of ceilometer data is surely smaller than that of an exponential profile, the sensitivity and DOFs will decrease.**

**I do not agree with this. Uncertainty of ceilometer data is not the same as the a priori covariance. A priori covariance should cover a range of expected variability.**

In our understanding "A Priori is the best estimate of the state before the measurement is made" (Rodgers, 2000). If a scaled ceilometer profile is used as a priori, it is therefore reasonable (but of course not mandatory, as pointed out by the reviewer) to also use its uncertainty as a priori covariance. Our answer was motivated by the fact, that by definition, DOFs directly depend on the a priori covariance but not on the a priori profile $x_a$. $x_a$ only has indirect impact, since it alters the posteriori profile and thus (due to non-linearity of the problem) the weighting functions involved in the DOF calculation. To further investigate this, we performed the side study described below.

**To me it is not clear how authors draw this conclusion (written in the abstract): "for optimal estimation algorithms this can be largely explained and compensated by considering smoothing effects, namely biases arising from the reduced sensitivity of MAX-DOAS observations to higher altitudes and associated a priori assumptions".**

This conclusion refers to the PAC, whose description and impact on the results is shown section 3.4 (particularly Figure 13) and related supplementary material (particularly Figure S25 and S26): Regarding the OEM formalism, there is a mathematical motivation for the PAC (see e.g. section "2.3.2 Smoothing effects", the beginning of section "3.4 Aerosol optical thickness" and Rodgers,

2000) and its application largely removes the typically observed underestimation of the AOT of MAX-DOAS observations (see right panel of Figure 13, and last two rows in figure S25 and S26).

In this context we noticed, that the term "smoothing effects" might be confusing and unnecessary here and we decided to remove it. The text reads now:

*"[…] for optimal estimation algorithms this can be largely explained and compensated by considering biases arising from the reduced sensitivity of MAX-DOAS observations to higher altitudes and associated a priori assumptions".*

**For my suggestion, participants do not need to repeat retrieval, but likely only one group performing OE can test the aerosol extinction profile obtained with the ceilometer and make an educated guess, probably based on the variability during the campaign, for the covariance matrix, is the PAC correction factor still needed?.**

Note, that the PAC correction is itself based on sun photometer scaled ceilometer profiles ($x_c$) and cannot be applied if $x_c$ is used as a priori. The correction factors become unity then (according to Eq. 9 and 10 in the manuscript). In fact, under simplifying assumptions (linearization of the inversion problem and ideal measurements), the application of the PAC as performed in the manuscript and the application of $x_c$ as apriori are expected to yield equal results for the AOT.

According to the reviewer's proposition and the comments above we performed a simplified study: We retrieved profiles with HEIPRO for the 14[th] of September (clear sky day with significant amounts of high altitude aerosol), using the daily average $x_c$ as a priori. The daily average was taken instead of the hourly profiles due to technical limitations in the current implementation of HEIPRO. The results are shown below. The figures were created from a run were the a priori covariance matrix was left unchanged compared to the one proposed in the paper.

[Figure]

[Figure]

Figure 1: Time series of UV AOT (360 nm) over the day (2016-09-14) for different retrieval approaches compared to the sun photometer (in black). Blue: "standard" retrieval with an exponential a priori profile. Green: same retrieval, but with the PAC applied. Orange: Retrieval with scaled ceilometer profile as a priori.

Figure 3: The average DOFs (AVK diagonal) for the aerosol UV extinction profile in the individual retrieved layers plotted over altitude. Colour coding is the same as in Fig. 1.

[Figure]

Figure 2: Example retrieved aerosol UV extinction profiles at 10h, together with the underlying a priori profiles. Colour coding is the same as in Fig. 1

Between 7h and 11h, the hourly ceilometer profiles are close to the daily average and rather stable. Indeed, we find good agreement here with the PAC corrected values. Interestingly, at least on this day, the DOAS observations still systematically draw the profiles away from the scaled ceilometer profile to smaller extinction values and thus smaller AOTs. As discussed in the paper, we suggest future investigation on this on the basis of forward simulated O4 dSCDs and improved assumptions on the aerosol properties.

Figure 2 shows example retrieved profiles. As before, profiles are drawn towards the a priori at high altitudes. The ceilometer a priori profile therefore leads to higher aerosol extinction at higher altitudes and thus to a better agreement of MAX-DOAS and sun-photometer AOTs

The DOFs slightly increase for the ceilometer a priori profile. Since the a priori covariance was not changed, the increase can be attributed to a real increase in sensitivity (the actual weighting functions) due to the altered light paths in the presence of aerosol. This is in agreement with findings in former studies (e.g. Frieß, 2006) and section 3.1. in our manuscript.

Decreasing the a priori covariance had two effects:

1. A decrease in the DOFs, since a priori knowledge is improved and the gain in information reduced.
2. Retrieved AOTs approach the sun photometer values, since deviations from the sun photometer scaled ceilometer profile become more "expensive".

We decided to not show these results in the already extensive paper, as this goes beyond its scope and should be investigated more thoroughly in the future. In addition, most of the mechanisms are already discussed qualitatively in the current version of the manuscript (e.g. the relation between a priori profiles and the necessity of the PAC on P14).

**Authors included this:**

**"[…]even though the motivation for the application of the PAC and the SF are different: the application of the PAC is necessary solely for mathematical reasons related to the concept of OEM and prior constraints applied therein. In contrast, publications that suggest or discuss the application of an SF (e.g. Wagner et al., 2009; Clémer et al., 2010; Ortega et al., 2016; Wagner et al., 2019).....They do not make use of optimal estimation or prior constraints similar to those used in our study. Thus their findings can be considered independent from any kind of PAC."**

**I might be wrong but I recall that Clemer et al. (2010) used OE in the retrieval.**

For the retrieval they do, but for the determination of the O4 scaling factor they do not perform any inversion. This is described in section 2.2. of their publication. To make things clearer, we added the corresponding section number behind the reference and changed P30, L26:

From: *"They do not make use of optimal estimation or prior constraints similar to those used in our study."*

To: *"For the determination of the SF, they do not make use of optimal estimation or prior constraints similar to those used in our study."*

**Note also that Ortega et al. (2016) suggested that aerosol extinction aloft might be important, which is related with your statement about the lack of sensitivity aloft due to assumptions in the a priori but it is not reflected or mentioned in the manuscript.**

We thank the reviewer for pointing this out and included a reference to Ortega on P29 L9:

[revised manuscript text omitted]

---

## Author Response (AR3)

**Response to the editor**

Remark: Editor comments are printed in bold, our responses in regular font, explicit changes made in the manuscript in italic font.

**After 2nd review, the author response file contains only responses to reviewer #1, but responses to the minor comments and suggestions made by reviewer #2 is missing; no changes appear to have been implemented in the revised paper. The editor considers this a simple oversight, but reviewer #2 should not be ignored.**

In fact, the missing responses to reviewer 2 were due to an oversight, we apologize. Please find the responses below.

**Both reviewers had raised the likelihood of a connection between PAC and SF < 1, and reviewer #1 had pointed to a missing paper (Ortega et al., 2016) that seems relevant here. Reviewer #2 had recommended to soften language that suggests SF and PAC "can be regarded as independent" (Section 3.4). The editor agrees that such an affirmative statement may be too strong, and can easily be misread; it further seems somewhat at odds with the authors own conclusion that "O4 scaling and PAC were found to have similar impact on the MAX-DOAS AOT results" (Conclusions). Consider to adopt the suggestion made by reviewer #2.**

We agree that the statement is too unspecific and strong and should be softened. However, it should clearly be stated that the PAC (the removal of an OEM mathematical artefact) cannot provide new insights on whether aerosol aloft is responsible for the SF (a finding drawn from forward simulations in Ortega 2016) or not. Further, we would like to emphasize that our statement in the conclusions "O4 scaling and PAC were found to have similar impact on the MAX-DOAS AOT results" is not generally valid, as the PAC correction factors (f_tau) are determined by the a priori assumptions. In fact, according to equation 9 and 10 in the manuscript, any f_tau between zero and infinity can be produced applying corresponding a priori assumptions. We therefore changed the corresponding sentences to:

*"Thus their findings can in general be regarded as independent from any kind of PAC, even though PAC and SF have similar impact on the MAX-DOAS AOT results with the a priori assumptions applied in this study. Particularly, it shall be pointed out that our findings regarding the PAC have no implications on whether elevated aerosol layers explain the necessity of the SF (as proposed by Ortega, 2016), or not."*

*"With the a priori settings applied in this study, O4 scaling and PAC were found to have similar impact on the MAX-DOAS AOT results."*

**Furthermore, as reviewer #1 points out, the Ortega et al. is not only a "source of bias" (as the current revisions state), but establishes that "lack of sensitivity aloft" (read "AOT-PAC = elevated layers aloft") and SF are plausibly connected. This is currently not said clearly enough in the revised paper. A simple solution may be to add a half sentence "..., consistent with earlier findings (Ortega et al., 2016)." behind the above sentence in the Conclusions.**

See our response above and our reasoning in the manuscript section 3.4. particularly P30, L14-28. Based on that, we believe that our findings regarding the PAC do not allow to doubt or support findings on the SF in former publications.

Ortega 2016 has been added as a reference in the course of the response above.

**Also consider to cite Ortega et al. 2016 together with Wagner et al., 2019 at the end of that same paragraph.**

We added Ortega as a second reference here.

**Belated responses to reviewer #2**

Reviewer comments are printed in bold, our responses in regular font.

**Reviewer #2: I have the two minor technical suggestions on wording:**
**While it is reasonably clear in the context of the response the referee, I think the following passage could be misinterpreted in the text: "In this way, its contribution to the total variance observed among the participants under clear sky conditions can be estimated to 40 % (for AOTs), 85 % (HCHO VCDs), 70 % (HCHO surface concentrations), 50 % (NO2 VCDs), 40 % (NO2 UV surface concentrations) and 20 % (NO2 Vis surface concentrations), respectively. The residual variance can be attributed to the choice and setup of the retrieval algorithm." Could the authors reword to make it more apparent that the reported variance contributions are those that arise from the measurements?**

Response: We reworded the sentence accordingly. It reads now:

"For clear-sky conditions, we find that the differences in the measured dSCDs are responsible for approximately 40% (for AOTs), 85% (HCHO VCDs), 70% (HCHO surface concentrations) and 50% (NO2) of the total variance observed among the participants. The residual variance can be attributed to the choice and setup of the retrieval algorithm."

**My comment regarding the PAC and SF comparison has been well addressed. Looking at the comments of Referee 1, however, there is a connection between these concepts documented in the literature, even if the the motivations for them are different. This study does not seem to clearly indicate one way or the other whether the concepts are related. Read in context the sentence near the top of page 32: "Thus their findings can be considered independent from any kind of PAC." is phrased as a conclusion and seems to imply that the concepts should be considered separately. While it is important that the comparison of PAC and SF is addressed think even this conditional conclusion is phrased to strongly. It is also possible that despite the different motivations, that the PAC and SF are related and considering them together might be fruitful, we do not know. I would recommend softening the language to something like:**

[revised manuscript text omitted]